# The same biophysical mechanism is involved in both temporal interference and direct kHz stimulation of peripheral nerves

Aleksandar Opančar [1,2], Petra Ondráčková[1,3], David Samuel Rose[1], Jan Trajlinek[1,4], Vedran Đerek[2] & Eric Daniel Głowacki [1,5] ✉

Temporal interference stimulation (TIS) is a promising noninvasive method for neurostimulation, yet its mechanism remains debated. TIS is often described as delivering low-frequency stimulation via the amplitude modulation (beat frequency) of interfering kHz carriers. However, this view overlooks known biophysical responses to kHz stimuli. Here, we test modulated (TIS) and unmodulated kHz waveforms on peripheral nerves in *Locusta migratoria* and in human sensory and motor pathways. We find that stimulation thresholds and strength-frequency relationships are governed by the kHz carrier itself, with minimal dependence on amplitude modulation. Across 0.5–12.5 kHz in humans and up to 100 kHz in locusts, all waveforms show overlapping excitation behavior, indicating a shared underlying mechanism. Our results support the hypothesis that suprathreshold TIS acts through kHz rectification, rather than envelope-specific effects. We further explore modulation frequency resonance, tonic vs. phasic effects, and suggest that two-electrode premodulated kHz may offer advantages over multielectrode TIS approaches.

Electrical neurostimulation always involves a tradeoff between invasiveness and specificity. Applying small electrodes to the exact area of interest is possible with high precision, however, at the cost of a complex, invasive, and to some degree damaging surgical procedure[1–3]. Completely noninvasive electrical stimulation from outside the body is much easier to perform; however, it is plagued by low spatial specificity and limited depth of effective penetration[4]. This means only superficial targets can be effectively stimulated, for instance, nerves located near the skin surface or brain structures directly below the skull. Obtaining suprathreshold neurostimulation, where neurons become sufficiently depolarized and directly fire action potentials, requires electric fields (> 1 V/m), which are difficult to obtain transcutaneously without causing discomfort from stimulation of sensory receptors and fibers in superficial tissues[5,6]. This limits the

diagnostic and therapeutic potential of noninvasive electrical stimulation, and implantable electrodes remain necessary for many applications where triggering of action potentials is desired (*i.e.* suprathreshold neurostimulation). The fundamental limitation behind noninvasive central or peripheral electrical stimulation is that the electric field decreases inversely with distance from the electrodes for any given conductive medium. The relatively high impedance of skin imposes practical limits to electric field penetration[6]. The challenge of how to effectively deliver electrical stimuli through the skin is a fundamental problem for noninvasive bioelectronic medicine. One proposed solution is to use high-frequency electric fields, which benefit from reduced skin impedance at higher frequencies[7]. The higher the carrier frequency, the lower the effective impedance is. One method to exploit this, which this paper interrogates, is the concept of mixing

[1]Bioelectronics Materials and Devices Lab, Central European Institute of Technology CEITEC, Brno University of Technology, Purkyňova 123, Brno, Czech Republic. [2]Department of Physics, Faculty of Science, University of Zagreb, Bijenička c. 32, Zagreb, Croatia. [3]Institute of Scientific Instruments of the CAS, v. v. i., Královopolská 147, Brno, Czech Republic. [4]International Clinical Research Center, St. Anne's University Hospital Brno, Brno, Czech Republic. [5]Department of Biomedical Engineering, Faculty of Electrical Engineering and Communication, Brno University of Technology, Technická 12, Brno, Czech Republic. ✉e-mail: glowacki@vutbr.cz

multiple high-frequency electrical signals to produce patterns of interference and amplitude-modulated (AM) beats in the tissue. This amplitude modulation frequency (AMF) is assumed to be the effective stimulation frequency recruiting neural firing. This interference method could solve the issue of depth penetration and potentially also spatial specificity. It was originally suggested by Nemec in the 1950s under the name interferential current stimulation (ICS) and first marketed by the company *Nemectron*. Since then, ICS has been widely adopted in the field of peripheral nerve stimulation, for instance, in functional electrical stimulation, physiotherapy, rehabilitation, and sports medicine[8–13]. The kHz interference method has regained increasing attention since its demonstration for brain stimulation in mice in the 2017 paper by Grossman et al.[14]. While in the older literature, the terms ICS or interferential current therapy (ICT) are more common, in recent studies this type of stimulation is termed temporal interference stimulation (TIS)[15–20]. TIS has shown efficacy in humans for stimulation of peripheral nerves[21], and promising examples of its use in brain stimulation have been reported[20,22]. In the literature, studies employing the term ICS typically involve suprathreshold stimulation of peripheral nerve targets, while TIS is the term more often associated with brain stimulation experiments, though in recent years, "TIS" often replaces "ICS" for peripheral nerve studies as well. The terms ICS and TIS have both been used since 2017. For clarity, we use the term TIS in the remainder of this manuscript.

The principle behind TIS (Fig. 1A) is that relatively high-frequency electrical stimulation signals, known as carriers, are applied in a range of over 1 kHz. In TIS, it is assumed that these > 1 kHz stimuli do not efficiently recruit electrophysiological activity on their own. Many papers describing TIS assume a priori that the carriers are of too high frequency and too low amplitude to efficiently elicit action potential

## A Temporal interference stimulation, TIS

$f_1$ = 3000 Hz      $f_1 + f_2$ = amplitude modulation

+

$f_2$ = 3001 Hz      =

$f_1 - f_2 = \Delta f = AMF$
1 Hz AM envelope

$f_2$   $f_1$   $f_2$   $f_1$

$\Delta f$ envelope

Stimulation

## B The kHz waveforms compared in this work

"TIS" 4-electrode          "Sine burst"          "Sine"
"AM Sine" 2-electrode      2-electrode           2-electrode

## C Peripheral nerve models

N5 nerve, locust

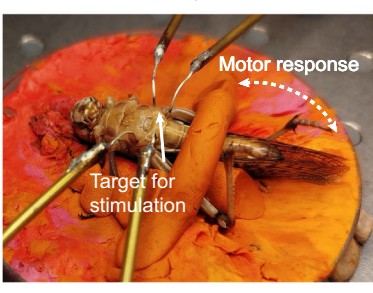

Motor response

Target for stimulation

Median nerve, human

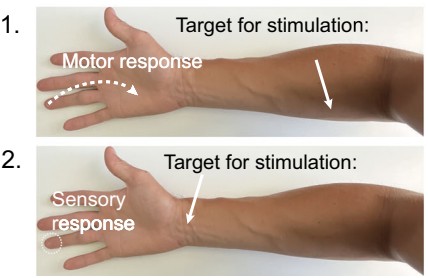

1.   Target for stimulation:
Motor response

2.   Target for stimulation:
Sensory response

**Fig. 1 | Testing the assumptions of TIS with respect to kHz electrical stimulation. A** The principle of TIS involves two kHz carriers at two different frequencies, $f_1$ and $f_2$, which interact in a medium via interference, creating areas of amplitude modulation (AM). Maximum AM depth is achieved at midpoints between stimulation electrodes. The AM occurs at a frequency which will be equal to the frequency difference between the two carriers, $\Delta f$. TIS can be used in vivo to deliver transcutaneous stimulation, where AM occurs at an area of interest to deliver phasic stimulation at a target region, for instance, to stimulate muscle activation. Note: the sine waves are illustrative only, they are not plotted to-scale. **B** We compare suprathreshold electrical stimulation of peripheral nerves using kHz carriers with three different waveforms and four different stimulation conditions: TIS, with 4 electrodes and AMF generated by interference in the tissue; AM sine, with 2-electrodes and AMF given by premodulation; Sine burst, where the kHz stimulus is turned on/off; and finally unmodulated sine. **C** The peripheral nerve stimulation models tested in this work. First, an insect model: stimulation of the N5 nerve in *Locusta migratoria*, giving evoked movement of the leg as the biomarker. Second, the median nerve in the human forearm. Evoked motor activity was tested by stimulating the median nerve branch to the *flexor digitorum superficialis* muscle (1), and evoked sensory response was tested by stimulating the median nerve at the carpal tunnel (2).

firing by themselves[14,16,19,20,23]. To accomplish TIS, multiple carriers of slightly different frequencies (*i.e.*, $f_1$ and $f_2$) are applied simultaneously, resulting in a frequency difference $\Delta f$ between them (Fig. 1A). The two carriers interact inside the tissue, coming into and out of phase with each other, a phenomenon known as interference. Interference causes amplitude modulation (AM) of the resultant kHz electric fields inside the tissue. The amplitude modulation frequency (AMF) is equal to $\Delta f$. The two frequencies $f_1$ and $f_2$ thus create an AM envelope frequency at $\Delta f$. A number of published TIS studies establish that suprathreshold neurostimulation synchronous to the AMF is occurring[14,21,24,25]. This has led to two competing interpretations: A) the AMF produced by the interference effectively causes phasic stimulation at the target site because of the intrinsic preference of neurons for lower frequencies, while the unmodulated high-frequency carrier has a negligible effect on tissues. In other words, the threshold for AM stimulation is significantly lower than the threshold of stimulation by the unmodulated carrier[14,26]; and neurons demodulate the AMF. B) TIS leads to phasic stimulation in regions of interference producing AM, while other regions are subject to equal or higher degrees of tonic stimulation by the carrier frequency, since biophysically the origin of the phasic and tonic stimulation is the same, and no selective demodulation of the AMF exists. Interpretation B was explained in a theory paper by Mirzakhalili et al.[17]. This paper motivated us to test these hypotheses in practice: we compare the stimulation thresholds of different modulated and unmodulated kHz waveforms (Fig. 1B). Namely, we test a 4-electrode TIS configuration versus a 2-electrode AM-sine (these two conditions produce equivalent modulated E-fields), an on/off modulated sine (Sine burst), and finally an unmodulated sine. We measure these four conditions as a function of carrier frequency, the strength-frequency, s-f, relation (0.5 kHz – 12.5 kHz) in insect and human peripheral nerve models (Fig. 1C). We find that all waveforms produce qualitatively similar stimulation, with essentially overlapping s-f curves for TIS and the other kHz waveforms. Taken together, our results support the conclusion that interpretation B is correct.

In both the older ICS and more recent TIS literature, the assumption that kHz carrier frequencies (i.e., $f_1$ and $f_2$) do not elicit stimulation has been rationalized in several ways. It is often assumed that opposing phases of the sinusoidal waveform cancel each other out, as their frequency exceeds typical neuronal firing rates. Another rationale for this assumption is based on the impedance/frequency characteristics of neural tissues. Because of the capacitive properties of cell membranes, higher-frequency currents pass through with less attenuation. This results in a lower voltage drop across the membrane and a reduced effective electric field as carrier frequency increases[27]. This argument was invoked by Grossman et al. as the low-pass filtering explanation[14]. Another aspect of the impedance/frequency argument can be seen in the ICS literature: The concept of higher frequencies encountering lower loss has been used to claim that high-frequency carriers can achieve more effective penetration than conventional pulsed current stimulation, allowing TIS to stimulate deeper targets than conventional (low-frequency) forms of transcutaneous stimulation[9,28]. While often taken as a given in papers, it has been pointed out that this depth argument is erroneous since the phase length of typical biphasic pulsed current stimulation is not different from kHz sine waves (hundreds of microseconds)[11,29]. Regardless of rationale, many TIS/ICS studies assume that the carrier sine is not the primary source of stimulation. Since neural activity typically occurs at frequencies below 100 Hz, TIS experiments often assume that effective stimulation arises from the AMF, which is also within this range. The central experimental goal of this paper is to quantify how stimulation thresholds differ between amplitude-modulated and unmodulated kHz signals. The key method we rely on is measuring strength-frequency (s–f) dependence. This is the frequency-domain analog to the well-known strength-duration (s–d) dependence of neural stimulation. To our knowledge, no prior study has compared s–f

dependence specifically in the context of TIS. Historically, s–f threshold characteristics have been measured for sinusoidal stimuli in the kHz range[30]. Such kHz signals can lead to stimulation, and the associated effect of nerve conduction block (kHz frequency nerve block)[31–36]. The inability of neurons to fire at high frequencies does not imply that they cannot be electrically stimulated by them. Experiments in the 1930s–1960s found that kHz frequencies 0.5–50 kHz would lead to evoked action potentials, and that the threshold current required for stimulation increased with frequency[30,37–40]. Electrodes in direct contact with the nerve can even produce suprathreshold stimulation with sinusoidal frequencies up to 1 MHz[41]. The s-f curve in the kHz range has been confirmed in different excitable tissues and cell types[42]. Over the frequency range of sinusoidal stimuli, it is apparent that the s–f relation (Fig. 2A) is governed by two exponential functions describing increasing thresholds at very low frequencies (<40 Hz, the so-called accommodation region), and at high frequencies (around 100 kHz and higher), producing a characteristic U-shaped curve[43,44]. This sine stimulation U-curve was first reported by Hill in 1936[45] and Katz[30] and described by the Hill equation. The empirical equation for the sine U-curve (plotted in Fig. 2A) was later refined by Reilly and colleagues[44]. The reason why a symmetric sine wave leads to net stimulation is that the depolarizing effect of the first phase is not fully counteracted by the subsequent opposite-polarity phase, due to nonlinear ionic conductivity across excitable cell membranes[35,42]. This effect can be illustrated by using established computational methods for model neurons, as first pointed out by Mirzakhalili et al.[17]. Figure 2B shows the calculated asymmetric membrane voltage change in response to a symmetric 1 kHz sinusoidal stimulus (*i.e.*, rectification effect). This net depolarization arises from the asymmetric conduction of sodium and potassium channels. Sodium channels respond faster and allow more sodium into the cell before inactivating, while the potassium channels are slower and less conductive, conducting relatively less potassium out of the cell. This results in more sodium in than potassium out, causing net depolarization with each successive sinus period[46]. This results in the summation of subthreshold depolarizations, as successive kHz half-cycles occur before the neuron can return to its resting potential. This kind of summation effect for kHz stimulation was first described by Gildemeister[37], to explain his observations that stimulation efficacy of kHz bursts was strongly dependent on burst duration (aka number of repetitions of the sinusoidal cycle). Using the model neuron, one can observe in practice how subthreshold and suprathreshold stimulation with unmodulated kHz versus TIS looks like (Fig. 2C). An unmodulated subthreshold kHz stimulus maintains depolarization in a way analogous to DC stimulation (blue trace in Fig. 2C). Gildemeister likened this effect to that of a DC stimulus, noting that a symmetric kHz signal causes both electrodes to act as effective cathodes, calling it "ambipolar stimulation"[38,39] since each electrode causes the same net depolarization (there is no anode or cathode with a kHz sine). He observed that cells below both electrodes were depolarized to the same level, as if both were acting like DC cathodes. As the current amplitude of the kHz sine increases, suprathreshold stimulation is reached (green trace in Fig. 2C). The summation effect builds up can be seen in the low-pass filtered membrane potential trace (dotted green line). The potential rises with successive kHz periods until the membrane potential threshold for AP firing is reached. By applying a $\Delta f$ = 5 Hz frequency offset between kHz sines, that is TIS conditions, a similar behavior is shown by the model: the kHz summation effect (red trace in Fig. 2C) causes the threshold to be reached every 200 ms. Importantly, the mechanism of stimulation originates from the summation of subthreshold depolarizations caused by the kHz carrier. The AMF can modulate how often the threshold is reached. While the AMF influences how frequently the threshold is crossed, it does not itself generate a low-frequency electric field. It is important to recall that in TIS, a frequency/power spectrum of two kHz sine carriers contains no energy at low frequencies. Two

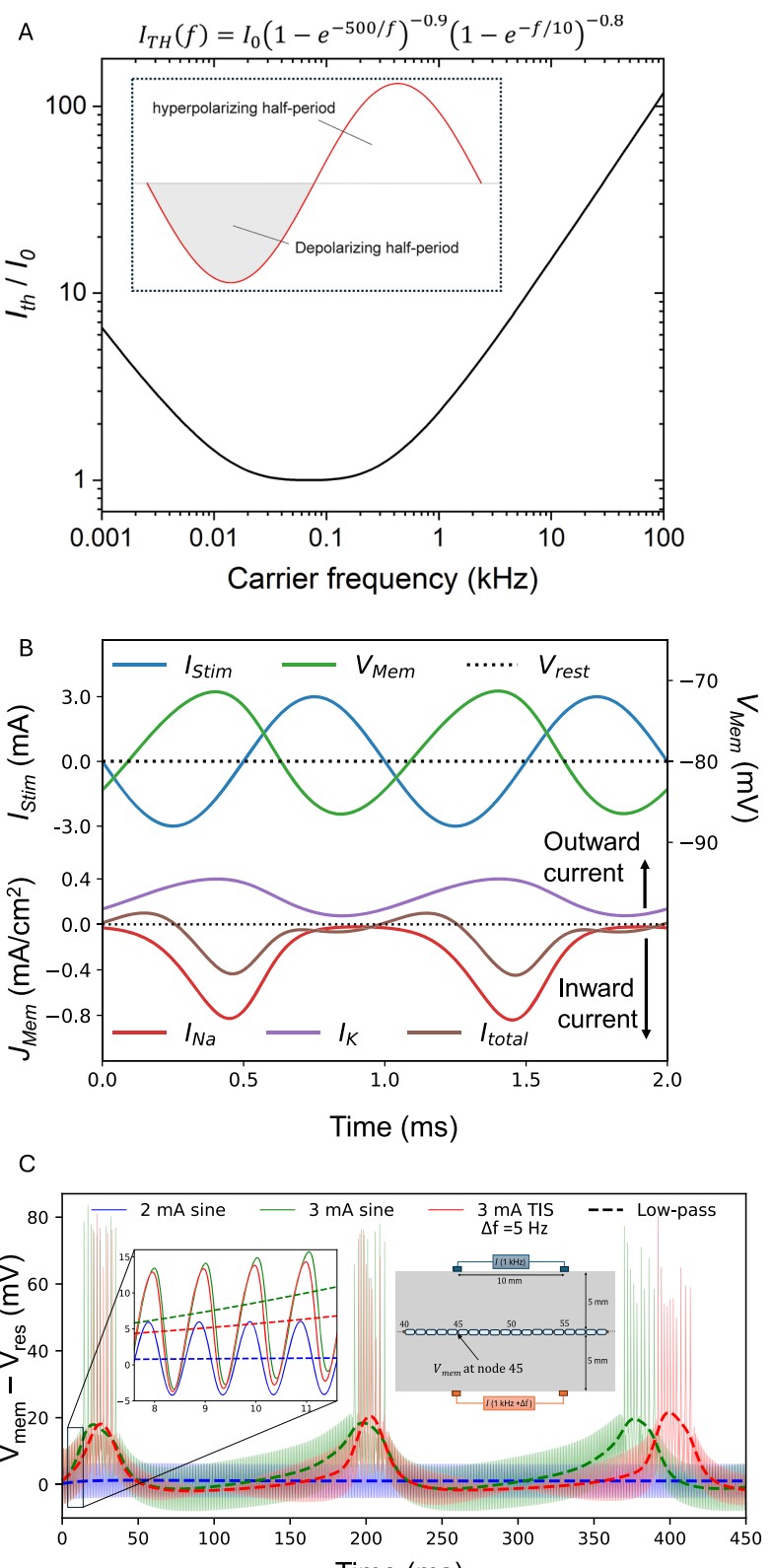

$$I_{TH}(f) = I_0\left(1 - e^{-500/f}\right)^{-0.9}\left(1 - e^{-f/10}\right)^{-0.8}$$

recent theoretical papers[17,47] maintain that the phasic stimulation effects observable in TIS experiments are driven by the rectification of kHz signals, and thus that off-target effects caused by tonic stimulation by the unmodulated carriers are unavoidable. These papers point out the importance of considering the effect of sustained application of kHz carriers on tissues superficial to the region being targeted by TIS, especially since tonic kHz stimulation can cause the phenomenon of

nerve conduction block[31,34,48]. However, it has been postulated theoretically[17] and reported experimentally[25] that due to resonant coupling of a kHz signal modulated by a preferred AMF, a given AMF can result in an optimally low stimulation threshold. Thus, it is possible to have a situation where a given unmodulated carrier is subthreshold, while the region of AM gives suprathreshold stimulation at the beat frequency. Therefore, in this paper, we set out to experimentally verify

**Fig. 2 | Biophysics of sinusoidal current stimulation. A** The strength-frequency, s-f, relation for the threshold current of sinusoidal stimulation is described by the Reilly equation[44] $I_{TH}(f) = I_0 \left(1 - e^{-500/f}\right)^{-0.9} \left(1 - e^{-f/10}\right)^{-0.8}$. The inset schematizes sinusoidal stimulation. It follows that if TIS stimulation is a function of carrier frequency according to this s-f equation, then TIS is driven by the carrier sine itself and not another mechanism. **B** According to the Mirzakhalili model[17], symmetric kHz stimulation waveforms cause net depolarization of cells, because of membrane rectification. Over one period of the kHz sine, there is more $Na^+$ influx than $K^+$ outflow, thus leading to net inward current and depolarization. **C** 1 kHz sinusoidal stimulation on a mammalian myelinated axon model[17], demonstrating the effect of rectification leading to depolarization. The model shows the calculated membrane potential during application of an extracellular stimulating current. Three test cases

are shown: unmodulated sine subthreshold stimulation (blue line), unmodulated sine suprathreshold stimulation (green line), and two sines with $\Delta f = $ Hz carrier frequency offset (TIS, condition, red line). All solid lines are unfiltered, showing the kHz oscillations, while the dashed traces are low-pass filtered to allow easier visualization of the slower membrane potential changes. Subthreshold stimulation gives a net membrane depolarization analogous to cathodic DC, however when the stimulation current becomes suprathreshold, one can see the summation effect at work as the membrane depolarizes more with each successive sinusoidal period, until threshold is reached and action potentials fire at the preferred resonance frequency of the neuron (5 Hz in this case). TIS, with a 5 Hz AMF, leads to a similar suprathreshold stimulation effect.

the stimulation thresholds for unmodulated kHz stimuli and compare them with AM kHz and AM delivered by TIS (as summarized in Fig. 1B,C). We found that TIS shows the same s-f relation as other modulated and unmodulated kHz waveforms, establishing that TIS is a form of kHz stimulation and there is no evidence for invoking another low-frequency demodulation mechanism. Nonetheless, we observed small differences in apparent thresholds between modulated and unmodulated kHz waveforms, consistent with theoretical predictions.

## Results

### TIS has the same strength-frequency dependence as 2-electrode kHz stimulation waveforms

To enable rapid prototyping of the TIS protocol against kHz stimulation, we propose using an invertebrate model such as the locust (*Locusta migratoria*). Despite structural and physiological differences from mammals, locusts provide a useful model for noninvasive electrical stimulation, as their cuticle has electrical properties comparable to human skin[49]. Moreover, the locust has a relatively simple and well-understood nervous system, including a pair of large symmetric motor/sensory nerves known as the N5 nerves, projecting from the metathoracic ganglion (Fig. 3A)[50]. The N5 nerves control the movement of the hind legs of the animal[50,51]. Stimulation of the N5 nerve leads to a highly reproducible biomarker in the form of evoked leg motion, which can be finely tracked with video analysis[52]. The reason for this reproducible evoked biomarker is that the largest muscle in the leg, the *extensor tibiae*, responsible for jumping, is controlled by only three neurons. Of the three, the fast extensor tibiae motoneuron (FETi) has a larger axonal cross-section than any other neuron inside the N5 nerve, thus it is preferentially stimulated, and the downstream actuation of the extensor tibiae muscle is a reliable biomarker. The stimulation of the FETi motor neuron was described by Zurita et al.[52,53] using a nerve-cuff electrode and tracking protocols for the N5 nerve in the locust, which inspired our method to stimulate the N5 target non-invasively. We applied two electrode pairs aligned with the N5 nerves so that opposite electrode pairs are oriented obliquely with respect to the axon direction. This way, the midway regions between the two electrode pairs where modulation index is maximal will be aligned over the N5. Using TIS in this configuration, we were able to evoke robust actuation of leg motion, using AMFs between 0.1–2 Hz. We next compared TIS using this optimized 4-electrode placement to a 2-electrode arrangement in which the active electrode was placed directly above the N5 (see Fig. 3A for configurations). The tested frequency range of the applied kHz signal, or carrier signal, was 500–12500 Hz. We used leg movement as the biomarker for stimulation, tracking the motion on video. We measured the carrier frequency-current threshold dependence (s-f curve) for TIS versus the two modulated sine waveforms and the unmodulated sine waveform (Fig. 3B). The AMF was set to 1 Hz for all modulated experiments. To ensure an approximately equal time for each trial, the maximum output current was changed from 1 mA, for frequencies below 5 kHz, to 10 mA for frequencies greater than or equal to 5 kHz. In all locust experiments the same ramping procedure was used across waveforms

and consisted of starting stimulation at approximately 5 μA and increasing stimulation in increments of 5 μA ( < 5 kHz) or 50 μA ( ≥ 5 kHz) every 0.5 s. Once the threshold was reached, the current was decreased until the evoked movement disappeared. At that point, the current was raised again to find the threshold, and this threshold current was recorded. The presentation of a given frequency was randomized for every trial. The modulated waveforms all produced a phasic kicking at 1 Hz, while the sine wave stimulation resulted in a tetanic contraction with the leg remaining in a fully extended position. All four stimulation types demonstrated a very similar s-f dependence (Fig. 3B). Statistical analysis was completed using a linear mixed effect model with fixed factors of frequency, waveform and their interaction term, as well as a random factor for 'participant'. A Box-Cox transformation was applied to the data ($\lambda = 0.02$). This analysis highlighted a significant difference for the main effect of frequency ($F_{(7,92)} = 940.73$, $p < 0.0001$) and waveform ($F_{(2, 92)} = 40.70$, $p < 0.0001$) with the interaction term not being significant ($p > 0.080$). A post-hoc pairwise comparison on the significant waveform effect using Tukey HSD highlighted that this effect was driven by a difference in AM sine threshold relative to sine burst (SE = 0.00058, 95% CI [0.00254, 0.00531], $t_{(92)} = 6.76$, $p < 0.0001$) and sine (SE = 0.00058, 95% CI [0.00358, 0.00635], $t_{(92)} = 8.55$, $p < 0.0001$). To interpret the practical difference in magnitude in threshold values between modulated and unmodulated sine, it is useful to consider the mean threshold ratio ± SEM. We find that unmodulated sine has, on average across frequencies, a lower threshold than modulated waveforms: AM/sine = 1.29 ± 0.04; Burst/sine = 1.06 ± 0.02. The 4-electrode TIS was excluded from these statistical comparisons above due to the inherent differences in electrode placements and the fact that ×2 total current is injected into the system, making a direct comparison of threshold value unreliable. For a detailed discussion of current injection/geometry effects, we refer the reader to Supplementary Note 1. The essentially identical qualitative features between all stimulation types, the overlapping s-f relations, and the finding that unmodulated sine has essentially the same threshold for stimulation indicate that TIS works by delivering kHz stimulation. The current threshold values we report are for one carrier only; that is, TIS is injecting in total twice as much current as the 2-electrode modalities.

We next extrapolated the experiment of comparing TIS with direct kHz stimulation to human subjects (Fig. 3C–F). As a convenient stimulation target, we chose the median nerve in the forearm. Both afferent and efferent fibers were targeted. Efferent fibers innervating the FDS in the mid forearm were the target for the motor stimulation experiment. To quantify stimulation, flexion and movement of the ring finger was used as a biomarker. Afferent fibers below the carpal tunnel at the wrist were also targeted which evoked a 'pins and needles' like sensation in the fingertips and palm (Fig. 3C). For efferent fibers, the threshold was defined as the current required to induce complete flexion, causing the finger to touch the palm (Fig. 3E). The threshold for the afferent fibers is defined as the onset of 'pins and needles' in the tip of the participant's ring finger. For TIS, the median nerve was optimally targeted by placing opposite stimulation pairs such that the

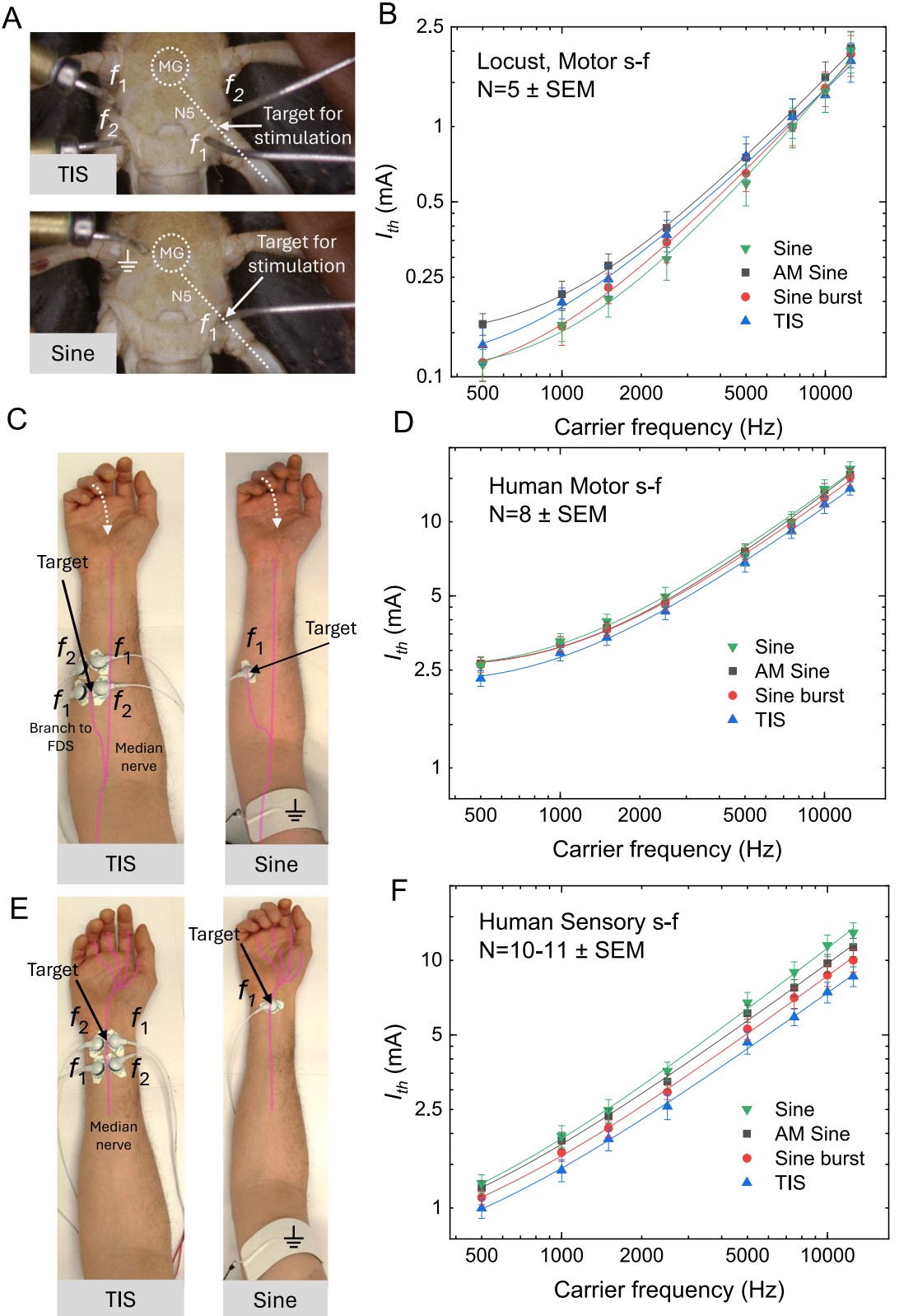

nerve lay along the midline between the electrodes, analogous to the electrode placement used for the locust N5. The location of the TIS electrodes was then used to define the position of the electrodes for two-electrode kHz waveforms; the electrode was placed directly over the median nerve, and stimulation was applied versus a relatively large ground electrode. We performed stimulation using carrier frequencies between 500–12500 Hz, first with AMF of 1 Hz. To ensure an approximately equal time for each trial, the incrementation step size was set differently for frequencies below 5 kHz compared to frequencies greater than or equal to 5 kHz. The same ramping procedure was used for each waveform and in both median nerve models. It consisted of increasing stimulation current from 0.25 mA in increments of 0.025 mA (<5 kHz) and 0.25 mA (≥5 kHz) every 0.5 s until threshold motor activation was reached, then the current was

**Fig. 3 | Comparison of strength-frequency (s-f) for 4-electrode TIS versus modulated and unmodulated kHz sinus waveforms in insect and human models.** AMF is 1 Hz for all experiments. **A** Experimental setup for N5 stimulation, with the two configurations: TIS 4-electrode and Sine for all other 2-electrode waveforms. **B** Current threshold/frequency s-f dependence for N5 nerve stimulation for the four tested stimulation types. To note: The reported current for TIS corresponds to one carrier only, i.e., the total injected current is ×2 higher. Sine stimulation results in tetanic contraction, while the other waveforms produce phasic motion at 1 Hz. Data from $N = 5$ locusts. **C** Stimulation of the median nerve, comparing TIS with sine waveforms. Stimulation of the median nerve (FDS branch) in the forearm results in reproducible motor response in the form of ring finger flexion a biomarker for stimulation. We arbitrarily define threshold response as the

ring finger flexing until it contacts the palm. **D** s-f dependence of motor threshold for FDS stimulation for different waveforms with 500–12500 Hz carrier frequency, $N = 8$ participants. All waveforms give phasic motion at 1 Hz, while unmodulated sine produces tetanic contraction. **E** Stimulation setup of the median nerve at the carpal tunnel to evoke sensory response in the hand. **F** Current threshold/frequency s-f dependence for sensory threshold for the four tested stimulation types. To note: The reported current for TIS corresponds to one carrier only, i.e., the total injected current is ×2 higher. Sine stimulation results in a constant sensory response, while the other waveforms produce phasic sensations. All lines between s-f data points are fits of the Reilly equation, $R^2 > 0.99$ in all cases. AM/Sine $N = 11$, TI/Burst N = 10 participants.

decreased until the evoked movement no longer persisted. At this point. The current was again ramped in the same fashion until the threshold was reached; here, the threshold current was recorded. The presentation of frequency was randomized for every trial. Statistical analysis was completed using a linear mixed effect model with fixed factors of frequency, waveform and their interaction, as well as a random factor for 'participant'. For the efferent motor nerve model, a box-cox transformation was applied to the data ($\lambda = 0.5$). This analysis highlighted a significant difference for the main effect of frequency (($F_{(7,158)} = 643.51$, $p < 0.0001$)) with the main effect of waveform (($F_{(2,158)} = 2.70$, $p = 0.070$)) and the interaction term ($p > 0.087$) not being significant. Therefore, stimulation thresholds across waveforms and the entire frequency range tested are not statistically significantly different from each other. The average threshold ratios for modulated/unmodulated sine waveforms are equivalent across the frequency range: AM/sine = $0.99 \pm 0.02$; Burst/sine = $0.97 \pm 0.02$. Further, the s-f dependence for all stimulation modes was found to be the same (Fig. 3D), and the Reilly equation (as shown in Fig. 2A) fit to each case. As before with the locust model, qualitatively, the movement evoked by each stimulation was identical. Participants reported that the sensation accompanying AM sine and TIS was essentially similar, with minimal or no skin sensation at the site of stimulation. The Sine burst was accompanied by obvious skin sensation and was subjectively less comfortable than the AM sine and TIS stimulation patterns. These differences in subjective perception may relate to the degree of synchrony generated by different modulated waveforms[54]. However, this topic lies beyond the scope of the present manuscript, in which we focus solely on the thresholds of sensory perception and motor movement. In our afferent median nerve model, an identical statistical approach was taken as with the motor stimulation data. The only difference being that a Box-Cox transformation was applied with $\lambda = 0.19$. The other features of the model were the same as stated above. This analysis highlighted a significant difference for the main effect of frequency ($F_{(7,221.01)} = 1146.55$, $p < 0.0001$)) and waveform ($F_{(2,221.12)} = 43.15$, $p < 0.0001$)), while the interaction term was not found to being significant ($p > 0.179$). A post-hoc pairwise comparison on the significant waveform effect using Tukey HSD pairwise comparison highlighted that this effect was driven by a difference in all three waveforms pairs, sine burst – AM sine (SE = 0.00537, 95% CI [−0.03876, −0.01340], $t_{(221.173)} = -4.85$, $p < 0.0001$) and sine – AM sine (SE = 0.00520, 95% CI [0.01174, 0.03628], $t_{(221.010)} = 4.62$, $p < 0.0001$) and sine−sine burst (SE = 0.00539, 95% CI [0.03737, 0.006281], $t_{(221.178)} = 9.29$, $p < 0.0001$). These differences equate to ratios of sensory thresholds which favor the modulated waveforms: AM/sine = $0.93 \pm 0.02$; Burst/sine = $0.84 \pm 0.02$. This trend is visible in Fig. 3F, and we would maintain that this is intrinsic to using reported subjective perception as a read-out. Signals with rapid on/off transitions provide greater sensory contrast, making them easier for participants to perceive. For the sine-burst waveform, the sharp onset provides a clear indicator of when the threshold is reached. This effect is less pronounced with the smoother on−off modulation of the AM sine, and in the continuous sine condition (with only slow ramping), it

becomes difficult to discern the exact moment of perceptual onset. TIS has the average lowest threshold, however we would indicate that this is due to a combination of its modulated nature and the 2× applied current, making it the clearest perceptual experience. Despite this caveat, again, a qualitatively identical s-f dependence is observed across all waveforms, further adding to the evidence that the mechanism underlying TIS and kHz stimulation is identical in afferent nerve fibers−the nerve fiber type with the typically lowest threshold for electrical stimulation. Taken together, these median nerve results evidence that TIS operates mechanistically via kHz stimulation. As with the locust model, our current thresholds for TIS are reported per single source. Thus, to achieve the same effect as a two-electrode setup, TIS requires roughly twice the total current. (See Supplementary Note 1 for a detailed discussion of current distribution and geometry.) With unmodulated sine, not only was the s-f relation the same as for the modulated waveforms, but the evoked motor movement was identical too, with the exception that contraction of the finger was followed by sustained tetanic contraction in all participants. In practical terms, for 4-electrode TIS, this means that the current threshold required for phasic stimulation is the same as tonic stimulation.

## Modulated and unmodulated sine frequencies in the range 10 − 100kHz share a common s-f relation

To further test the hypothesis that TIS is, in principle, a type of kHz stimulation and no carrier-independent envelope demodulation mechanism is occurring, it is interesting to extrapolate the s-f curve to even higher frequencies. Using a high-frequency voltage source, we attempted to apply modulated (AMF = 1 Hz) and unmodulated carriers in the range above 12.5 kHz and even up to 1 MHz. We found reproducible stimulation up to 100 kHz (Fig. 4). The Reilly s-f relation continues to hold up to this frequency, and we found no significant difference between AM and unmodulated sine. This was determined using the same statistical model as described above for the previous locust data ($\lambda = 0.02$) with a main effect of frequency being significant ($F_{(7,50.01)} = 85.70$, $p < 0.0001$) and the main effect of waveform ($F_{(1,50.02)} = 0.44$, $p = 0.5099$) and their interaction ($F_{(7,49.96)} = 0.81$, $p = 0.5795$) not being significant. We note that it was possible to observe stimulation at even higher frequencies, however, results were unreliable, and there were effects coming from Joule heating (including visible damage to insect tissue) due to the relatively high currents required to achieve threshold. Therefore, we report only the safe window we found up to 100 kHz. The fading of neural response at high frequencies and the transition into a thermal-dissipation dominated regime are reported in several kHz neurostimulation studies[36,41,43]. The fact that the s−f relation of modulated and unmodulated carriers still fits the Reilly relation points towards the conclusion that, as with conventional kHz frequencies, TIS acts via carrier signal rectification, and there is not another carrier-independent mechanism of low-frequency envelope demodulation emerging at increasing carrier frequencies.

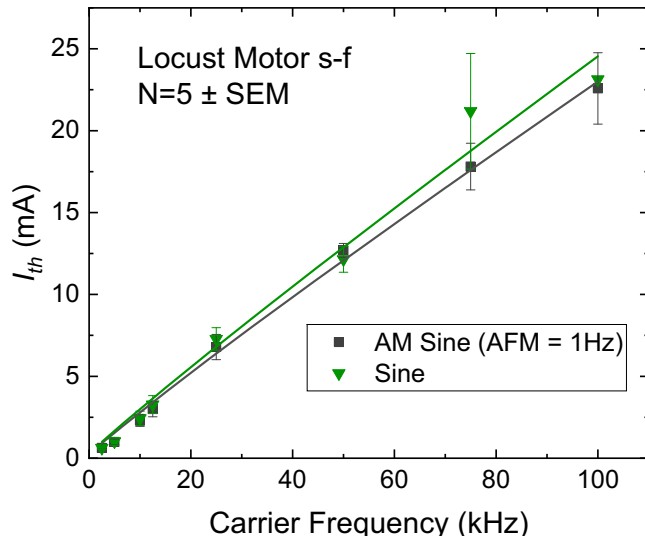

**Fig. 4 | Strength-frequency relation for AM sine versus unmodulated sine for frequencies up to 100 kHz.** The locust N5 nerve can be reliably stimulated with carrier frequency up to 100 kHz, in a tonic or phasic way using sine or AM-sine, respectively. Frequencies higher than 100 kHz result in excessive heating and unreliable results. Data from $N = 5$ locusts.

## Varying AMF has limited impact on the suprathreshold stimulation current threshold

From assessing the s-f relation of TIS and different modulated/unmodulated kHz waveforms, it would appear that threshold current varies strongly with carrier frequency, but at least at AMF = 1 Hz, there are no meaningful threshold differences caused by amplitude modulation itself. We next performed a series of experiments to interrogate the effect of AMF in locusts (looking at low AMF below 1 Hz), and in the human forearm models, in a larger range between AMF = 0.03 Hz and AMF = 50 Hz. In locusts, we decreased the AMF from 1 Hz down to 0 Hz while keeping the carrier at 2.5 kHz. In Fig. 5A, we plot the time of initiation of evoked kick and the associated threshold current as a function of the AMF envelope. The threshold does not change with AMF, and the kick is initiated once a common threshold value is reached. During the experiment, we monitored the deflection angle, θ, of the femoral-tibial joint, defining Δθ as the difference between the unstimulated resting angle and the maximum deflection angle evoked by stimulation (Fig. 5B). Stimulation at the minimum threshold current at AMF = 1 Hz gives a Δθ of 5°, and a phasic movement with a period of 1 s. As AMF is lowered, the value of Δθ increases. By AMF = 0.1 Hz, the Δθ has increased to 34°, with a phasic movement every 10 s. In essence, lower beat frequencies keep the N5 nerve in a suprathreshold state for longer durations. This sustains the *extensor tibiae* muscle contraction for a longer period and leads to a larger Δθ. The fraction of time over threshold, plotted on the right y-axis, corresponds to the deflection angle. This point is made more evident by decreasing the AMF to 0 Hz, that is, stimulation with unmodulated 2500 Hz. The leg then deflects to a maximum Δθ of 57° and remains at this position in a tetanic contraction. This experiment aligns with the original observation of Gildemeister[37] that the stimulation effect is proportional to the duration of application of kHz pulses, known as the temporal summation effect. The leg deflection is proportional to the time that the stimulation amplitude is over threshold, a concept which we will formalize in the upcoming section, as well as Supplementary Note 2. Moreover, Fig. 5C demonstrates the transition between phasic and tonic stimulation at threshold. First, the stimulus is applied at the threshold current with a 1 Hz modulated sine wave, yielding a 5° leg movement at 1 Hz. Next, with the current amplitude held constant, the modulation is

turned off; this tonic stimulation causes the leg to extend and remain in tetanic contraction. The implications of phasic vs. tonic stimulation are discussed in more detail in the following section.

The model of nonlinear neuronal membrane conduction properties, which result in rectification of sinusoidal signals, also predicts that this rectifier circuit responds to an optimum AMF, aka the preferred frequency of the neuron[17]. That is, a TIS frequency that gives the lowest activation current threshold[17]. For myelinated axons, the model predicts a U-shape of current threshold versus AMF, where an optimal minimum of current threshold should exist at a given AMF, and very low or very high AMF will lead to higher current thresholds. This optimal AMF is predicted to be in the range of 5–40 Hz. Experimental data, however, suggest that this effect may be weak in peripheral nerve stimulation. In his 1939 paper, Katz reported no threshold difference between rapid and slow amplitude ramps on kHz stimulation of the frog sciatic nerve[30]. In a study of sensor, motor, and pain thresholds using ICS, Palmer et al., found lack of dependence of thresholds on AMF, although this study used a window of frequencies of 5–100 Hz[55]. Budde et al.[25] using an implanted cuff electrode on the rat sciatic nerve, did demonstrate a U-curve dependence with AM-kHz stimulation (AMF range 1–1000 Hz) of the sciatic nerve, with dependence roughly ±10–20% lower thresholds for AMF = 10–50 Hz than for very slow or fast AM. We performed experiments on the human median nerve to test AMF dependence from very low frequencies approaching unmodulated sine waves (0.03 Hz) up to 50 Hz. We selectively stimulated sensory fibers at the location of the carpal tunnel to establish sensory threshold (setup from Fig. 3E), and motor threshold by stimulating the median nerve branch to the FDS at the upper forearm (setup in Fig. 3C). In both cases, we did observe a shallow U-shaped dependence on current threshold with an apparent optimal beat frequency of AMF = 5 Hz in the motor nerves and AMF = 1 Hz in the afferent nerves (Fig. 5D), where thresholds were lower by roughly 20% compared to low or high AMF. Participants were presented with randomized AMF signals to eliminate any effects of ascending or descending AMF presentation. Due to variation of absolute threshold values between participants, the data are normalized $i/i_{min}$ for each participant, then these values are averaged and evaluated. Not every participant had the $i_{min}$ value at the same AMF, thus the averaged normalized curves do not intersect exactly 1 at any point. Statistical analysis using a linear mixed effect model was employed to determine if the apparent observed decrease in threshold was statistically significant. A fixed factor of modulation frequency and a random factor of participant was set. For the sensory model it was found that AMF = 1 Hz was significantly different to frequencies, 0.03 Hz ($p < 0.0031$), 0.1 Hz ($p < 0.0121$), 20 Hz ($p < 0.0405$), 30 Hz ($p < 0.018935$) and 50 Hz ($p < 0.0058$). AMF = 5 Hz was found to be significantly different to frequencies, 50 Hz ($p < 0.1590$), 30 Hz ($p < 0.0471$), 0.1 Hz ($p < 0.0312$) and 0.03 Hz ($p < 0.0089$). This aligns with the observed pattern in Fig. 5D and its less 'sharp' dip compared to the motor data. An identical statistical model was used to determine significance levels in the motor nerve model. For AMF = 5 Hz the following frequencies were found to be significantly different, 50 Hz ($p < 0.0004$), 30 Hz ($p < 0.0074$), 1 Hz ($p < 0.0442$) and 0.1 ($p < 0.0157$). For AMF = 10 Hz the following frequencies were found to be significantly different, 30 Hz ($p < 0.0282$) and 50 Hz ($p < 0.0019$).

## TIS intrinsically gives bimodal stimulation with tonic and phasic regions, while two-electrode kHz stimulation is monomodal

In TIS, the kHz carriers are unmodulated, and amplitude modulation arises due to interference in the tissue. As a general rule, the maximum depth of AM will occur at midpoints between electrode pairs. This means that regions of tissue superficial to the point of maximum AM will be exposed to kHz carrier stimulation with a lower index of modulation. Moreover, since the electric field amplitude will fall with distance from the stimulation electrode, these areas could be exposed to relatively high magnitudes of an unmodulated kHz carrier. This will

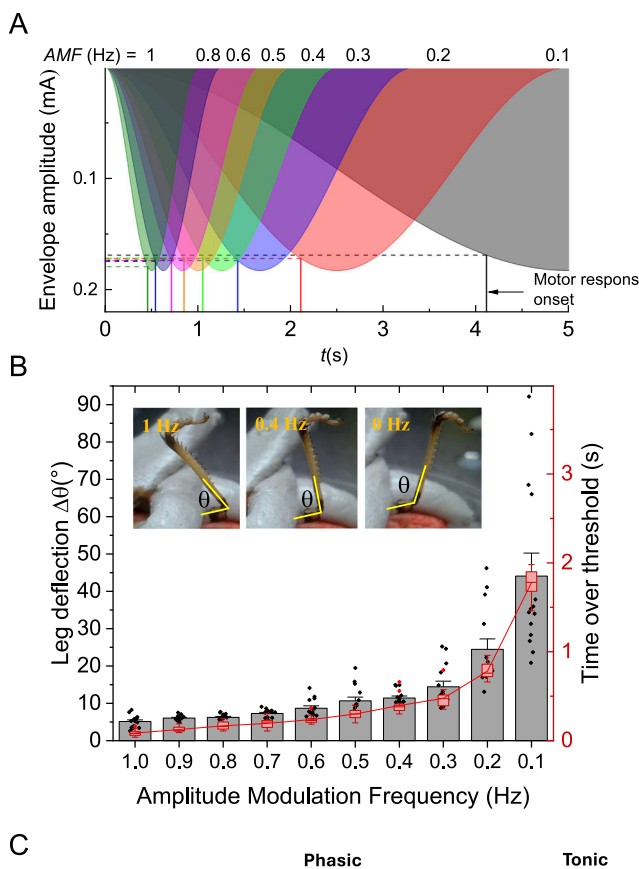

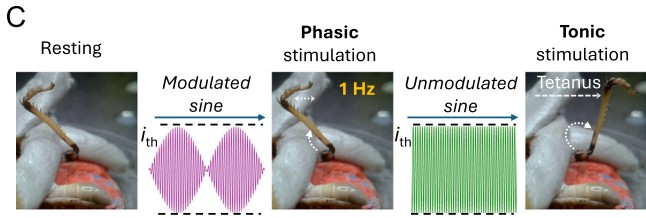

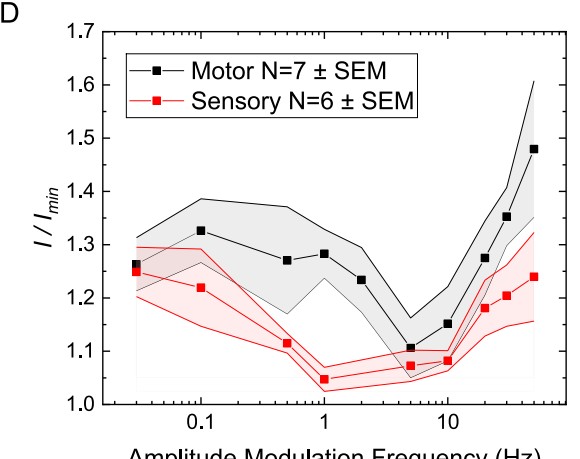

**Fig. 5 | Stimulation thresholds as a function of amplitude modulation frequency, AMF. A** Locust N5 nerve stimulation threshold current and timepoint of evoked movement as a function of AMF, plotted on top of the AM envelope of the kHz stimulation (carrier frequency = 2.5 kHz). In this range of AMF, the threshold does not vary with AMF, the kick is evoked once a constant current value is reached, regardless of the amplitude ramping conditions. **B** Leg deflection angle, $\Delta\theta$, caused by *extensor tibiae* actuation, as a function of AFM. The deflection angle correlates to the length of time that the kHz stimulation is over threshold, with unmodulated kHz giving the greatest deflection angle change. Bars represent the mean ± SEM. Individual data points are shown as black dots. Box plots represent the interquartile range (IQR) with the line inside each box indicating the median. Whiskers extend to the most extreme data points within 1.5×IQR; data points outside this range are plotted individually as outliers in red. $N = 14$ repetitions for each AMF in a single locust. **C** This experiment involves applying an at-threshold stimulus at AMF = 1 Hz, and then turning off the modulation, while maintaining the current setpoint. This causes tetanic contractions. Turning the modulation on and off will shift the stimulation result from phasic movement to tetanic contraction. This test shows that threshold current for the phasic stimulation and the tonic stimulation are effectively equal. **D** Effect of AMF on the motor and sensory thresholds in the median nerve stimulation for a fixed stimulation frequency of 3000 Hz. The characteristic U-shaped response predicted by theory is demonstrated, with statistically significant increases in threshold at low and high AMF, relative to the optimum AMF ≈ 5 Hz. Data from $N = 6$ participants for sensory stimulation, and $N = 7$ for motor stimulation.

the 4-electrodes by 1 cm, as shown in Fig. 6A, will cause a shift to tonic stimulation and tetanic contraction of the muscle. We use the finite element method in a forearm model to explore and visualize the implications of our experimental findings that the stimulation thresholds do not meaningfully differ for modulated and unmodulated signals (Fig. 6B). We perform frequency domain simulations and calculate the relevant activating functions in the nerve direction $A_1 = |d^2 V_1/dx^2|$ and $A_2 = |d^2 V_2/dx^2|$, where $V_1$ and $V_2$ are electric potential amplitudes generated by stimulation currents $I_1$ and $I_2$ respectively[56]. The simulation details can be found in the Computational modeling section. We plot the maximum amplitude $A_{MAX} = A_1 + A_2$ in the horizontal cross-section at the nerve depth, which is assumed to be 1 cm below the skin surface (Fig. 6C). We use our experimental finding of a single common threshold for stimulation $A_T$ to quantify and visualize the stimulation response mode (tonic, phasic, or subthreshold) in the various regions in the tissue. For this purpose, we use time over threshold (TOT) to mean the fraction of each modulation period that the nerve is suprathreshold, mathematically defined as (derivation in Supplementary Note 2):

$$\text{TOT} = \begin{cases} 0 & \text{if } A_1 + A_2 < A_T \\ \frac{1}{\pi}\arccos\left(\frac{A_T^2 - A_1^2 - A_2^2}{2A_1 A_2}\right) & \text{if } |A_1 - A_2| \le A_T \le A_1 + A_2 \\ 1 & \text{if } |A_1 - A_2| > A_T \end{cases} \quad (1)$$

We obtained the value for the stimulation threshold $A_T$ in the FEM simulation by using the experimental current threshold measurements from Fig. 3D. For 2.5 kHz carrier frequency calculated stimulation threshold $A_T$ is 1000 V/m². TOT = 0 represents an unstimulated region where stimulation is never over the threshold. TOT = 1 denotes a fully tonic region (stimulation is always above threshold), whereas values between 0 and 1 define phasic regions where stimulation periodically transitions from subthreshold to suprathreshold. TOT for different values of stimulation currents $I_1$ and $I_2$ is shown in Fig. 6D. It reveals regions around the electrodes which have a high unmodulated activating function (TOT = 1). Nervous tissue in these regions would be subject to tonic stimulation. In between the electrodes, there are regions of maximum modulation index of the activating function. Here, phasic stimulation will be observed at the beat frequency $\Delta f$.

thus cause regions of phasic simulation (AM region where interference is optimal), and regions of tonic stimulation (regions closer to the stimulation electrodes). We refer to this coexistence of tonic and phasic regions as *bimodal stimulation*. This intrinsic issue with TIS has been pointed out by other researchers[17,25,47]. The phasic/tonic effect can easily be demonstrated in the FDS motor stimulation experiment in the forearm (Fig. 6A and supplementary video 1). An optimal placement of the 4 TIS electrodes, producing maximum AM at midway points between electrodes, will cause phasic muscle activation. While keeping the stimulation current conditions the same, moving one of

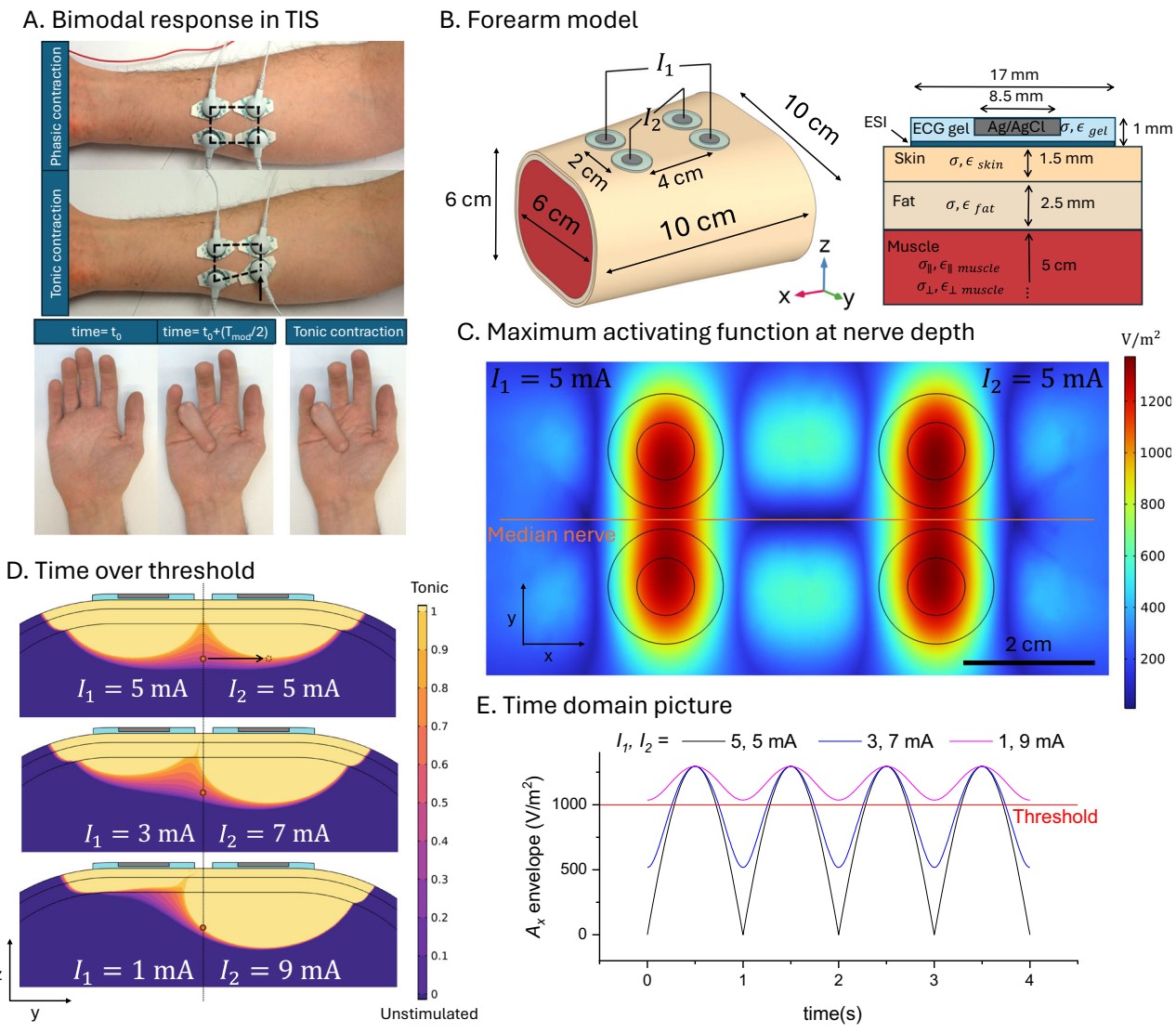

**Fig. 6 | Simulated tonic versus phasic stimulation regions in the model human forearm. A** Demonstration of the bimodal phasic/tonic response in TIS configuration. Moving one of the 4-electrodes directly over the nerve will result in tonic stimulation and tetanic contraction of the muscle (see also supplementary video 1). **B** Geometry and the 3-layer tissue model used for simulation. Electrode-skin interface (ESI) is modeled by a distributed impedance consisting of constant phase element in parallel to resistance. **C** Horizontal cross section showing the activating function maximum at the nerve depth, assumed to be 1 cm from the skin surface. **D** Vertical cross section showing the fraction of time over threshold (TOT) for different stimulation current ratios (current steering), and the spatial distribution of tonic (1) and phasic (0.5) regions. Moving the electrodes directly over the nerve shifts the response from phasic to tonic, as observed experimentally. **E** As the current ratios change, the nerve is exposed to different degrees of modulation, with the ratio 1, 9 mA causing the nerve to be stimulated over threshold in a tonic way.

These calculations align with experimental observations (Fig. 6A). Figure 6E shows the time-domain representations of the activating function envelope at the nerve, for the same current ratios as in Fig. 6D.

Since we have observed the essential equivalence of stimulation effects with TIS and AM-kHz, it can be useful to apply finite element modeling to visualize key similarities and differences between the two stimulation approaches (Fig. 7). We make a simple geometry to illustrate some general points: Assuming a uniformly-conductive medium with circular symmetry, one can see that performing 4-electrode TIS or 2-electrode AM sine will lead to equivalent stimulation in the center of the model: a phasic stimulus with the same amplitude and maximum modulation index. The critical difference is that TIS will require 4 electrodes and thus twice as much total current injection, and will create regions of relatively high-intensity tonic stimulation below the respective electrodes. This results in bimodal stimulation–some areas are phasic, some are tonic, and areas in between will have a gradient in amplitude modulation index. In 2-electrode sine stimulation, all regions are exposed to the same phasic stimulation with an identical modulation index. The attenuation of the E-field with distance will be the same in both TIS and 2-electrode kHz sine configurations. This we refer to as monomodal stimulation. It should be noted that in practical experiments, like we have done in Fig. 3, one of the two electrodes in AM sine can be much larger than the primary stimulation electrode, thus limiting unwanted stimulation at that electrode site by decreasing current density. In experimental situations where one wants to avoid tonic stimulation effects, a 2-electrode pre-modulated configuration may be preferred over the 4-electrode TIS. Moreover, the 2-electrode configuration uses less current than the 4-electrode TIS to achieve the same E-field.

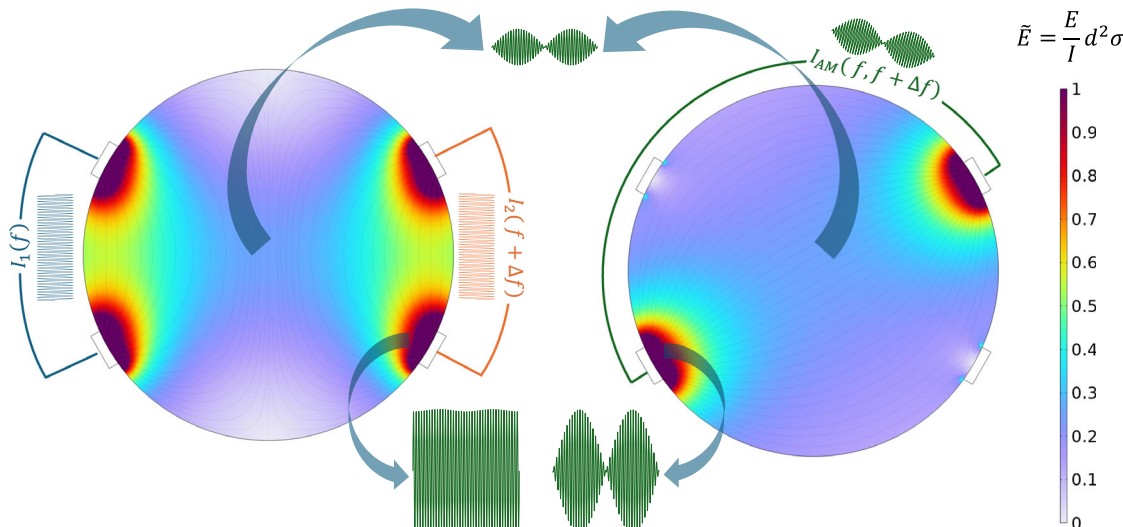

**Fig. 7 | Generalized calculation of the scaled electric field Ē, comparing TIS with a 2-electrode AM-kHz stimulation in a circular medium of diameter *d*, conductivity σ, stimulation current *I* and electric field amplitude *E*.** The value of the scaled electric field Ē is only a function of the geometry and is independent of the chosen values of *I*, σ, or spatixal scale *d*. This picture explains why TIS will always result in higher amplitudes of unmodulated carriers superficial to the region of maximum amplitude modulation, where phasic stimulation is delivered. Both stimulation methods lead to the same amplitude of phasic stimulation in the middle of the medium; however, TIS is bimodal, while AM-kHz is monomodal.

## Discussion

The TIS (*aka* ICS) method has been proposed as a noninvasive electrical stimulation technique which can yield focused stimulation at a target region at-depth. In recent papers describing the use of this method, it is assumed at the outset that the kHz frequencies used as carriers are not able to elicit stimulation on their own, at least not at the amplitudes being used in the experiments[14,19,57,58]. The experiments we are reporting, as well as recently published in vitro studies[59–61], indicate that this assumption is not correct. We find no evidence for selective extraction of the envelope AMF, but rather that the biophysics of stimulation originates from the sinusoidal carrier itself. Our work, corroborated by these in vitro studies, shows that the threshold for stimulation by kHz sine waves varies little, if at all, with amplitude modulation. The hypothesis that unmodulated and modulated kHz likely have similar thresholds can be anticipated from older literature, starting as early as Bernhard Katz in 1939 who reported no effect of ramp rate on threshold for kHz stimuli on a frog sciatic nerve[30]. Symmetric periodic signals in the kHz range will lead to net depolarization of neurons and can in fact stimulate. Modern scientists may overlook this due to the age of these studies (most of the work was published between the 1930s and 1970s), and the fact that the foundational papers by Gildemeister[37] and later Bromm[39,40,62] and Wyss[38] are not available in English and were rarely cited in later English-language literature. As shown in these works, the time of application of the signal matters, due to the principle known as temporal summation, the summation effect, or Gildemeister effect, where multiple successive subthreshold stimuli result in net suprathreshold stimulation[39,63]. We find, in three different peripheral nerve models (in an insect model and in humans), that direct neurostimulation is driven directly by the kHz stimulus itself, and that the biophysics behind stimulation in TIS versus a single sine wave appear to be the same, *i.e.* the characteristic s-f curve which is established in the literature. While the notion that kHz carriers cannot lead to net stimulation is not correct, it is possible when using optimal AMF (~5 Hz) to exploit the resonance effect to have a situation where the threshold current for the modulated signal is up to 20% lower than for the unmodulated carrier. However, the question is how practically relevant this drop in effective threshold is. TIS for brain stimulation attracted interest because of the promise of selectively targeting deep neural structures, without stimulating superficial structures. If an optimal AMF is chosen, would the resulting threshold drop counteract the effect of E-field decay with distance from the electrodes? It seems unlikely that one could stimulate a deep target via TIS without also exposing intervening tissue to high-amplitude tonic stimulation; at a minimum, this issue must be carefully considered when interpreting results. A recent in vitro study has shown that although the stimulation threshold for modulated vs. unmodulated sine may be the same for a single cell, due to inhibitory network effects, TIS may yield differential stimulation of excitatory cells due to input from inhibitory ones[60]. There is also ongoing debate about safety parameters for transcranial TIS[64] and about what constitutes an appropriate sham condition in TIS experiments. Given that an unmodulated kHz carrier can itself stimulate neural tissue, one can argue whether it truly constitutes a sham condition as opposed to an interfering-currents (AM) experimental condition. In other words, an unmodulated kHz stimulus is an active condition (capable of neural stimulation) and thus not a true sham in the sense of no stimulation. A proper sham would involve no current at all (or a placebo setup), whereas a single-carrier condition controls for off-target high-frequency effects rather than providing a null stimulus.

Our experimental design was motivated to study relatively simple biomarkers, and we focused on suprathreshold stimulation phenomena only. One limitation of our study is that it does not shed light on what may occur at the subthreshold level. From theoretical calculations (as shown in Fig. 2C), one expects that the ion channel rectification phenomenon will be occurring in the same way for subthreshold stimuli, and the cell will net depolarize with similar dynamics as one sees for suprathreshold stimulation. However, our work does not rule out that other parallel mechanisms related to the AMF might play a meaningful role at subthreshold levels. This paper was written to address two disparate communities. First, the functional electrical stimulation community, which has used ICS (a term more prevalent than TIS) for decades. ICS is standard in clinical practice and claims of better depth penetration along with assumptions that unmodulated sines do not stimulate are common in that literature. We believe our results unequivocally confront these issues (for suprathreshold stimulation). In contrast, the TIS research community (focused on brain

stimulation) has become increasingly concerned with subthreshold stimulation in humans, where additional phenomena (at cell and network level) may be at work. There are principled reasons to expect neural processing of amplitude-modulated inputs, for instance, the well-known example of auditory steady-state response[65]. In this sense, our results may be considered limited; however, at a minimum, they should serve to challenge the assumption that high-frequency carriers are electrophysiologically inert and should encourage always testing stimulation of unmodulated carrier as one of the experimental conditions.

What is remarkable compared to other modulated kHz stimulation is that TIS can deliver bimodal stimulation. That is, there will be tonic and phasic regions, with gradients of amplitude modulation depth in between. An advantage of TIS over a 2-electrode kHz stimulation configuration is the opportunity to use current ratios to steer the position of AM hotspots, and/or exploiting the 180-degree phase shift that will exist between hotspot regions perpendicular to electrode pairs. These features allow degrees of control of phasic/tonic stimulation, which has been used for peripheral nerve stimulation. These advantages have been documented by Budde et al. for achieving a core/shell stimulation effect on peripheral nerves with cuff electrodes −where phasic stimulation is delivered at the center of the nerve while more superficial fibers are exposed to tonic, and thus potentially conduction-blocking, stimulation[25]. Another recent approach with intermittent TIS on peripheral nerve cuffs published by Rossetti et al.[66] essentially produces an inverse pattern: TIS is used to produce a *lower* stimulation amplitude at the interference hotspot, while other regions are exposed to higher stimulation intensities. Thus overall the use of interfering currents from multiple electrodes can produce patterns that 2-electrode configurations cannot do. In completely noninvasive applications, peripheral or central, one must consider the intervening tissue, which under TIS conditions is necessarily exposed to higher-amplitude kHz signals with a lower modulation index. A two-electrode premodulated AM-sine current stimulation avoids the issue of tonic stimulation.

In conclusion, while TIS can be a powerful tool to direct stimulation to a region of interest, it is not simply a surrogate for delivering low-frequency stimulation. In the literature, there has been controversy about the mechanism by which TIS works. Many studies have oversimplified the picture of TIS, maintaining that the high-frequency carriers are relatively inert. Our experiments provide evidence that this picture is incorrect. We conclude that the stimulation observed with TIS can be explained by conventional kHz stimulation models, and we encourage revisiting the older literature on this topic. This mechanistic basis must be considered in planning safe and effective TIS applications. A key practical information that our study provides is the carrier frequency dependence relevant to TIS, which should serve to inform experimental design in the future. Moreover, in many cases, we would argue that a premodulated kHz stimulation 2-eletrode arrangement may provide some key benefits of TIS but in a monomodal way with less total current needed, eliminating tonic stimulation effects.

## Methods

### Ethics
Experiments on locusts, as invertebrates, do not fall under legislation in the Czech Republic as animal experiments, therefore no special ethical permission is required (Act No. 246/1992 Sb.). Experiments on human healthy volunteers were performed in accordance with approval from the CEITEC Ethical committee for research on human subjects. Stimulation is delivered using the Digitimer DS5 device which is clinically-approved in the EU.

**Electrical stimulation hardware.** The kHz stimulation experiments were conducted using the Digitimer DS4 (for locust experiments), and the Digitimer DS5 (for experiments on human participants). For high-frequency experiments in the range 10–100 kHz, we used an A-301HS amplifier from A.A. Lab systems Ltd. Stimulation electrodes for locust experiments were Ag wires (0.6 mm diameter) with chloridated tips (made via immersion in commercial chlorine bleach solution). For human transcutaneous stimulation, the primary stimulation electrodes always consisted of commercial ECG Ag/AgCl gel-assisted adhesive electrodes. For 2-electrode modulated kHz experiments, we used larger return electrodes consisting of gel-assisted carbon TENS pads. Stimulation waveforms are generated by a 2-channel function generator (Keysight Technologies EDU33212A). Synchronization of clock-circuit for each of the channels should be performed via the internal software prior to each experimental session. For premodulation of a sine wave stimulus (AM-sine), two protocols were used: either internal AM modulation provided on one channel of the function generator (AM by multiplication), or by having each channel output a carrier and then mixing the carriers in a BNC T-spitter, which was then fed into the DS5 or DS4 (AM by addition, essentially mimicking the AM created in TIS). While we noted no differences in results from the two methods of AM generation, most experiments were carried out by the carrier mixing for convenience and to more faithfully compare with TIS conditions. Both current simulators (DS4 and DS5) suffer from current amplitude roll-off with increasing frequency > 1 kHz, as can be found in the documentation for each device. This effect was described in detail by FallahRad et al.[33]. To correct for this roll-off, the actual output stimulation current was always monitored in all experiments by registering voltage drop across a calibrated shunt resistor installed in series, using a digital oscilloscope (Picoscope 2206D, input impedance 1 MΩ, software: PicoScope 7 v7.1.39.3737). All currents reported in this paper are these measured current values. The stimulation experiments were controlled by interfacing the function generator with a Python code to run the experimental sequence allowing the trial parameters to be randomized within an experimental block and ramping procedure to be consistent across trials.

**Stimulation experiments on *Locusta migratoria*.** Animals were obtained at a local pet shop and kept in the laboratory in a terrarium with heat lamp with grass and oats *ad libitum*. The only inclusion criteria for performing the experiment were adult individuals with fully intact anatomy and lack of any obvious superficial injuries. Prior to the stimulation experiment, we anesthetized the locust by cooling to 5 °C for around 15 min. Next, the animal was immobilized with the ventral side facing upward, in a bed made of modeling clay, and allowed to warm up to room temperature prior to starting the experiment. Stimulation was delivered via 2 or 4 Ag/AgCl wires placed onto the cuticle using *xyz* micromanipulators (S-725CRM, Signatone USA). Contact between the wire and the cuticle was enhanced by adding a drop of EKG gel onto the tip of the wire prior to making contact with the cuticle. In the optimization of the 4-electrode TIS experiment, we used an oscilloscope and microwire probe, relative to a ground electrode in the tail of the animal, to verify the hot spot of high amplitude modulation depth and high relative amplitude. The video of the leg movement was recorded using a digital camera (Sony DSC-RX II) with a frame rate of 100 fps. An initial threshold at AMF = 1 Hz was obtained to detect the minimum detectable leg twitch, approx. 5 degrees. Stimulation was started and AMF frequency changed once 5 kicks were observed at a given frequency. Deflection angles as a function of AMF were obtained using a free and open-source video analysis tool *Tracker* v6.3.0 (https://opensourcephysics.github.io/tracker-website/). Temporal syncing of the video to the stimulation waveform was done by triggering an LED which produced a light impulse (visible in one frame of video) at the beginning of the waveform and again at the AM maximum of the waveform. Data were obtained from naïve animals, and individuals were not reused for subsequent experiments with data collection. We would note, however, that the animals survive this experimental sequence without a problem and can be returned to the

**Table 1 | Materials parameters used for the finite element model**

| Material | Conductivity $\sigma$ [S/m] | Relative permittivity $\epsilon_r$ |
|---|---|---|
| Muscle along | 0.6 | $1.5 \cdot 10^5$ |
| Muscle across | 0.35 | $1 \cdot 10^5$ |
| Adipose tissue (fat) | 0.024 | 3000 |
| Skin | 0.0015 | 40000 |
| ECG gel | 0.2 | 70 |

terrarium after marking with a colored permanent marker. The locust experiment to determine the s-f curve was coded in such a way that the presentation of a given frequency was randomized within a waveform block. A 30 s to 1-min gap was taken in between each trial. The exact location of the two-electrode stimulation was always determined from the position of the TIS electrodes, therefore the block of TIS stimulation always came first. After this the choice of waveforms was randomized. Ramping to threshold was carried out identically for each waveform. In all locust experiments the ramping procedure was as follows: The Keysight was set to 0.005 $V_{pp}$ and stimulation started. Every 0.5 s the amplitude was increased by 0.005 $V_{pp}$ until the threshold was observed. At this point the amplitude was decreased by 0.005 $V_{pp}$ every 0.5 s until the evoked activity disappeared, then the amplitude was again increased by 0.005 $V_{pp}$ until the evoked action reappeared. To ensure an approximately equal time for each trial, frequencies >5 kHz had a max output current set to 10 mA, trials with frequencies <5 kHz had a max output current set to 1 mA. Each trial began again from the same initial amplitude value. For experiments with frequencies >12.5 kHz an automated ramping procedure could not be used due to the increased variability in the ability to stimulate the N5 nerve. As such manual ramping was used so that the threshold values could precisely be determined. Similarly, because there is a higher likelihood of damaging the ability of the leg to kick at higher frequencies, randomization of frequency presentation was not possible. Instead, frequencies were applied in increasing order.

**Stimulation experiments on human subjects.** Subjects were healthy volunteers who expressed written informed consent. To ensure pressure exerted on the forearm, and thus movement on the muscles during the experiment, was minimized the participants hand was placed on a cushion with their elbow on the table. This meant the forearm was not in contact with the table and a more stable position obtained. Electrodes were placed onto the skin and adhered in place using tape. The primary stimulation electrodes were always gel-assisted AgCl EKG electrodes (Ceracarta), in two-electrode stimulation protocols the ground electrode was a larger TENS pad electrode $4 \times 4$ cm$^2$. Subjects were not told of the specific sequence of stimulation or variable changes but were not otherwise blinded. The experiment was coded in such a way that the presentation of a given frequency was randomized within a waveform block. A 30-s to 1-min gap was taken between each trial. The exact location of the two-electrode stimulation was always determined from the position of the TIS electrodes, therefore the block of TIS stimulation always came first. After this, the choice of waveforms was randomized. Ramping to threshold was carried out identically for each waveform. In all human experiments the ramping procedure was as follows: The Keysight was set to 0.05 $V_{pp}$ and stimulation started. Every 0.5 s the amplitude was increased by 0.05 $V_{pp}$ until the threshold was observed (or reported by the participant). At this point the amplitude was decreased by 0.05 $V_{pp}$ every 0.5 s until the evoked activity disappeared, then the amplitude was again increased by 0.05 $V_{pp}$ until the evoked action reappeared. To ensure an approximately equal time for each trial frequencies <5 kHz had an incrementation rate of 0.005 $V_{pp}$. Each trial began again from

the same initial amplitude value. This general scheme was applied to all human experiments.

**Participants**
Participants were recruited via an advertisement in the local community and on the institute's campus. A total of $N = 8$ participants (average age $34.25 \pm 11.2$, 62% Female) were recruited for the motor stimulation of the median nerve. For our afferent nerve stimulation experiments $N = 11$ participants (average age = $36.3 \pm 8.21$, 36% Female) were recruited. For the determination of the 'U-curve' in the afferent model $N = 6$ participants were recruited (average age = $33 \pm 10$, 83% Female); for the efferent model $N = 7$ participants (average age = $27 \pm 5$, 28% female). Participants were not compensated for their participation. Each participant gave informed written consent.

**Computational modeling.** NEURON model (used for Fig. 1d) was taken from Mirzakhalili et al.[17], the model files are published and freely available online at https://github.com/mirzakhalili/Mirzakhalili-et-al--CellSystems-2020. The computation for Fig. 1d was made by simulating 1 kHz extracellular axonal stimulation for two different current amplitudes 2 mA (subthreshold) and 3 mA (suprathreshold). Sinusoidal stimulation was obtained by setting $\Delta f = 0$ Hz and TIS stimulation was obtained by setting $\Delta f = 5$ Hz. The results were evaluated at node 45.

Finite-element modeling (Figs. 4, 5): Calculations were done in COMSOL Multiphysics 6.3 using the AC/DC Module and Electric currents physics interface. The study was solved in frequency domain for 2500 Hz carrier frequency. The forearm is implemented as a three-layer model inspired by the work by Medina and Grill where the layers represent skin, fat and muscle[7]. Between the skin and the gelled ECG electrodes is a distributed electrode-skin interface (ESI) impedance consisting of a constant phase element CPE with parameters $K = 35$ M$\Omega$s$^{-\alpha}$cm$^2$ and $\alpha = 0.9$ in parallel to a large charge transfer resistance $R_p = 4.7$ M$\Omega$cm$^2$ with the parameter values as reported by McAdams et al.[67]. The boundary condition on the ESI is given by $\mathbf{J} \cdot \hat{\mathbf{n}} = (V_{skin} - V_{gel}) \cdot (R_p^{-1} + Q(j\omega)^\alpha)$, where $\mathbf{J} \cdot \hat{\mathbf{n}}$ is normal current density at the interface elements, $V_{skin}$ and $V_{gel}$ are the electric potentials on each side of the ESI, while the second bracket is the admittance of the CPE. The boundary condition on the electrodes is set by the total current $I_{tot}$ flowing through the electrode surface $\partial\Omega$, $\int_{\partial\Omega} \mathbf{J} \cdot \hat{\mathbf{n}} \, dS = I_{tot}$. We perform two independent simulations to calculate the current contributions from each electrode pair, in one simulation we set $I_{tot} = \pm I_1$ on the first pair of electrodes and $I_{tot} = 0$ for the second pair, and in the second simulation $I_{tot} = 0$ for the first pair and $I_{tot} = \pm I_2$ for the second pair, finally, the total current is obtained by adding the contributions from each pair. It is important to do the simulation in this way because the current stimulators used in the experiment are isolated. All other external boundaries have electrical insulation boundary condition $\mathbf{J} \cdot \hat{\mathbf{n}} = 0$. In the bulk the current continuity equation $-\nabla \cdot (\sigma \nabla V + \epsilon_0 \epsilon_r j\omega \nabla V) = 0$ is solved for electric potential $V$, where the first contribution is from the Ohmic current parameterized by bulk conductivity $\sigma$ and the second contribution is displacement current parameterized by relative permittivity $\epsilon_r$. For the muscle tissues $\sigma$ and $\epsilon_r$ are anisotropic with different values along the muscle fibers (x-direction) and across the muscle fibers (y,z direction). The entire geometry is meshed with a tetrahedral mesh with $1.2 \cdot 10^6$ domain elements with sizes ranging from 0.5 mm near the surface layers to 5 mm deep in the bulk. Activating functions are calculated as second-order spatial derivatives of the electric potential in the nerve direction (x-direction). Values of the material parameter used in the simulation for the frequency 2.5 kHz are listed in Table 1 are taken and adapted from Gabriel et al.[68].

## Reporting summary

Further information on research design is available in the Nature Portfolio Reporting Summary linked to this article.

## Data availability

All data supporting the findings of this study are available within the article, and both raw and processed data source data is available at https://doi.org/10.6084/m9.figshare.28535426. Any additional requests for information can be directed to, and will be fulfilled by, the corresponding author. Source data are provided with this paper.

## Code availability

The code used to analyze the stimulation data and run the *PsychoPy* (v2024.2.4) experiments is available on GitHub under the MIT license: https://github.com/dsrose314/TIS_Custom_Code/tree/main Aleksandar Opančar, Petra Ondráčková, David Samuel Rose, Jan Trajlinek, Vedran Đerek, Eric Daniel Głowacki, The same biophysical mechanism is involved in both temporal interference and direct kHz stimulation of peripheral nerves, TIS_Custom_Code, 10.5281/zenodo.16098578, 2025.

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

## Acknowledgments

This work was supported by the European Research Council (ERC) under the European Union's Horizon 2020 research and innovation program (Grant Agreement No. 949191, E.D.G.), by funding from the National Center for Neurological Research, supported by the Czech Ministry of Education, Youth, and Sports MEYS CR (LX22NPO5107), and funding from the Czech Science Foundation GAČR (grant agreement No. 25-18184X, E.D.G.). Device fabrication was made possible by *CzechNanoLab* Research Infrastructure supported by MEYS CR (LM2023051). The team is grateful for financial and hardware support from *Stimvia s.r.o.* This work has been supported by the Croatian Science Foundation under the project UIP-2019-04-1753 (V.Đ.). V.Đ. and A.O. acknowledge the support of project CeNIKS co-financed by the Croatian Government and the European Union through the European Regional Development Fund—Competitiveness and Cohesion Operational Program (Grant No. KK.01.1.1.02.0013), and the QuantiXLie Center of Excellence, a project co-financed by the Croatian Government and European Union through the European Regional Development Fund—the Competitiveness and Cohesion Operational Program (Grant KK.01.1.1.01.0004).

## Author contributions

A.O. conducted all computational modeling, with support from V. Đ. P.O. and E.D.G., supported by J.T., performed all preliminary experiments on TIS comparisons with kHz in locusts and humans, and P.O. made the first observation that thresholds of kHz stimulation appear similar for modulated and unmodulated waveforms. A.O. and D.S.R. performed all final locust experiments and data analysis. Final human data was collected and analyzed by D.S.R. V.Đ. and E.D.G. supervised the work and obtained funding. A.O., V.Đ., and E.D.G. conceived the project idea and performed conceptualization. The manuscript was written by E.D.G. with input from all authors.

## Competing interests

The authors declare no competing interests.
