## [Transparent Peer Review file · Nature Communications]

The same biophysical mechanism is involved in both temporal interference and direct kHz stimulation of peripheral nerves

Corresponding Author: Professor Eric Glowacki

Version 0:

Reviewer comments:

Reviewer #1

(Remarks to the Author)

Temporal interference stimulation (TIS) has received major attention in recent years because it has been advertised to provide targeted deep brain stimulation noninvasively without major side effects. Yet, there is still no consensus about the mechanisms of action for TIS or whether it can be safely and effectively be applied in humans.

The current manuscript investigates if TIS is different compared to pure kHz stimulation for peripheral nerve stimulation. The current study is different compared to most other TIS studies because it studies the applicability of TIS in peripheral nerves and not the brain and to my knowledge, it is the only study that has both animal model and human subjects. Altogether, the authors have designed experiments to address some of misunderstandings that are commonly believed about TIS. In summary, using the results from their animal model and human subjects, the authors have reported that there is essentially no difference between TIS and direct kHz stimulation. While their results are compelling, I believe the authors need to provide more evidence for some of their claims and I am not sure if some of the results provided fully support their claims. I hope my comments can help the authors to improve their manuscript:

Introduction:

1- The authors have thoroughly reviewed the prior TIS research in their introduction. It is especially encouraging that they have acknowledged that the concept of TIS is older compared to what is commonly believed. However, the authors have only focused on direct activation of cells using TIS in their introduction. Yet, recent studies are hinting that TIS might be more of a subthreshold stimulation method. Even though such alternative arguments about the effectiveness and mechanism of action for TIS are related to the central nervous system and not much related to peripheral nerves, I believe the introduction can be improved if the authors discuss them.

2- Figure 1-d is a little hard to follow and can be improved by better organization and labeling. For example, the authors have mentioned in the caption that all the dashed lines are low-pass filtered but the resting potential is also a dashed line, and I don't think it is low-pass filtered.

3- It is not clear what is the electrode configuration for the results shown in Figure-1d but I believe the node that the authors are showing the results for is not where the action potentials are generated because I do not see much amplitude modulation in the voltage traces (blue or green lines). I recommend showing the results on a node where amplitude modulated stimulation is strong so that it can be compared with the authors already shown in Figure1-d to demonstrate how current summation is different in presence/absence of amplitude modulated stimulation.

Results:

1- The main take away from the current manuscript is that TIS is not different from direct kHz stimulation. The main point for this argument that authors are using is based on their results that the threshold of activation is not that different from direct kHz stimulation of amplitude modulated stimulation (AM-SW; the authors need to define this term in their manuscript). The results that they are showing in their manuscript (Figure 2-b and Figure 2-d, for example) supports this argument. However,

the authors need to run proper statistical tests to argue that the thresholds are not significantly different from each other. I understand that the N is low for the animal model and human subjects and the authors might not be able to conduct proper statistical tests. If that is the case, then the authors need to either increase their N, which is relatively low or state clearly that they cannot show that TIS is different from direct kHz or AM-SW stimulation in terms of activation threshold.

2- The authors have stated that the participants have felt “less comfortable” with SW burst stimulation. It would be interesting to provide more specific feelings (pain, itch or heat, for example) in case the participant mentioned it. Also, was their level of “uncomfortable” feeling related to the frequency of stimulation.

3- At the end of page 10, the authors have used 0.1-1 MHz stimulation to argue that AM envelope demodulation does not occur. First, I could not find any figures associated with these claims. Second, I am not sure how the authors are using changes in the carrier frequency to argue that AM demodulation is not present in their models.

4- Using the results in Figure3-a, the authors have argued that their animal model does not have a “preferred” stimulation offset frequency in which the activation threshold is lower. First, I am not sure how the authors are relating the level of deflection to activation thresholds. Second, the range that they are changing the offset frequency is small (0-1 Hz). Therefore, I am not even sure if they could see a preferred offset frequency in that range. Third, there seems to be a change in the trend of deflection angle at 0.4 Hz, which needs some explanation.

5- The authors have argued that amplitude demodulation through rectification is due to nonlinearities in the ion channels. Yet, they argue that their animal model does not show amplitude demodulation because it has unmyelinated axons. I am not sure how not having unmyelinated axons is related to not being able to rectify stimulations if it is related to nonlinearities of ion channels.

6- The authors have mentioned that for 1 Hz frequency offset, they observed 1 Hz phasic response and for 0.1 Hz frequency offset they observed 0.1 Hz phasic response. However, it would be insightful if they plotted the phasic frequency of response for all the offsets they used. Moreover, the authors should show the onset of response relative to the phase of stimulation for all offsets. Similarly, they should show the time when the leg returned to resting position relative to the phase of the envelope.

7- It is not clear what is i_{max} that the authors are using to normalize the results especially when there is no point on the plot that is at unity. Also, the authors have reported 20% lower activation threshold at the preferred frequency. However, is this reduction statistically significant? Elsewhere, the authors have mentioned 3 human subjects, but it seems like they have 7 subjects for Figure3-C. They need to clearly state how many human subjects they had and basic demographics such as age and gender for them in their method section. If they had to exclude some of them for some of the results, then they need to mention that and provide exclusion criteria.

8- Is there any explanation why there is more variability in the normalized stimulation thresholds at the lower beat frequencies?

9- The authors need to report more details regarding the experiment on human subjects when they changed the frequency offset. For example, it would be helpful if the author provided similar results to Figure 3-A for their human subjects. Additionally, the same metrics that I suggested to add for the animal model can be added for the human subjects. Specifically, the frequency of phasic response and the onset/offset of the response compared to the phase of stimulation beat.

10-The authors have indicated that they “arbitrary” used flexing of the middle finger to define thresholds. Then, if we assume that movement of the middle finger was the target, how did the movement of other fingers compare to the movement of the target. This is important information that needs to be provided for all the experiments they conducted (stimulation types, carrier and offset frequencies). Even though the authors did not see an obvious difference in thresholds, there can be other subtle differences in stimulation types that the authors did not investigate. For lack of a better word, the movement of other fingers can be considered as a side effect, which could be different across different experiment protocols they used.

11-The authors have provided some modeling results in Figure 4. First, they need to provide details of their model such as boundary conditions and some characteristics for the computational mesh they used. Second, I am not sure in what direction the authors are showing the activating function. I assume they are looking at the second derivative in the x direction but that needs to be clarified. Also, I am not sure about the average activating function and how this non-scalar metric was averaged. Regardless, I am not sure about the purpose of the results the authors are showing in Figure 4. I understand that they are arguing that by changing the current ratio it is possible to alter between phasic and tonic response for human subjects. However, they did run experiments on human subjects in which they altered the currents. In general, I am not sure if I could find any results showing that the author showed tonic response in human subjects.

12-The authors need to provide details of their modeling for Figure 5 such as boundary conditions.

Discussion:

1- Similar to my first comment for the introduction, the authors should provide some discussion that their results can rule out how TIS might be effective if its main mechanism of action is a subthreshold phenomenon.

2- The authors have discussed how the TIS leads to a “constant” DC shift. I am not sure this is an accurate statement. I believe the DC shift will be constant where there is no amplitude modulation (in some regions of TIS and everywhere for direct kHz stimulation), but it will have some dynamics where there is amplitude modulation. The authors should comment on this and the investigation I suggested for Figure 1-d would help to clarify this argument.

3- The authors have argued that an amplitude modulated amplitude modulated kHz stimulation can provide key benefits of TIS more straightforwardly. The authors should be more specific about the benefits. For example, I am not sure the premodulated signal can be completely steered. Is premodulated stimulation provide phasic response throughout the tissue or it is going to lead to tonic response near the electrodes depending on its amplitude? Is the common benefit limited to feeling less uncomfortable?

4- Considering premodulated stimulation, if it is going to provide a phasic response in the majority of the tissue, is there going to be a phase difference between the regions? For example, will the regions near the electrode respond at an earlier phase? And is that going to be a benefit or disadvantage? There are some arguments (Barra et al. 2024 biorxiv, for example) that completely phasic response leads to less “natural” neuronal response and TIS can feel more natural because the phase of the response is more variable. Then, can premodulated stimulation achieve a naturalistic response?

Experiments:

1- The authors have mentioned that they increase the stimulation to find the threshold. However, some other key information is missing. Did they also increase the carrier frequency in increasing order? If so, did they search for the threshold from 0 at each carrier frequency? If not, did they start from the current from the lower carrier frequency? Same question can be asked about experiments in which the authors altered beat frequency.

2- What was the order for finding the threshold for different stimulation protocols? For example, did the author perform TIS first, SW first? And did they perform all the experiments for one stimulation type and then switch to another waveform or they altered between the stimulation types for each carrier frequency.

3- It is not clear how long the authors stimulated at each carrier/beat frequency. Was it always the same duration or was it a function of carrier/beat frequency?

Reviewer #2

(Remarks to the Author)

Thank you for inviting me to review “There is no biophysical distinction between TIS and Direct kHz stimulation” by Opančar et al. (NCOMMS-24-55626).

Although originally discovered in the 1950s, the past few years have seen an explosion of interest in temporal interference stimulation (TIS) but it remains unclear how this technique works, or indeed if it does anything at all. Conventionally, TIS is thought to act through the superposition of two high-frequency (kHz) electric fields: it is assumed that each individual field oscillates too rapidly to entrain neural activity, but their overlap produces a slower “beat” (or amplitude modulation; AM) which, when demodulated, somehow affects the neurons.

Here, Opančar and colleagues argue for an alternate interpretation of TIS where the neurons are directly driven by the kHz carriers and apparent AM occurs just because beats reflect periods where the two carriers are jointly maximal; no envelope extraction or demodulation is required. To support this hypothesis, they perform two simple experiments to describe the strength-frequency curve, using limb (in locust) and finger (in human) movement as a readout. In both cases, they find that TIS, amplitude-modulated sine waves, and bursts behave similarly across these preps.

I am very sympathetic to the idea that the effects of kHz stimulation need to be examined more carefully, though I do think Mirzakhaili et al. (2020; Cell Systems) has thoroughly covered this ground already in terms of modeling and lit review. The experiments presented in this paper do add to the debate, but very little data is shown for some of them (especially, the 10 kHz - 1 MHz experiment) and the methods are also scanty. The procedure for measuring thresholds in humans, for example, only gets a sentence: how many repetitions collected for each subject? How were they structured?

However, my biggest concern relates to interpretation: how do these data inform understanding of human TIS? This is not related to the model system, but rather the nature of the stimulation. Human TIS is generally subthreshold (well below it, in fact) and applied centrally. It is not clear to me that we can sensibly extrapolate from threshold measurements in the periphery to cranial TIS at ~0.5 V/m AM. I would appreciate a lot more discussion on this point. The authors do refer to the Gildemeister effect, but it’s not clear to me if that’s enough (and if it is, shouldn’t there be interesting dynamics in the data, as the effect builds up over time?).

Similarly, the authors also seem to attack some strawman positions. For example, they outline three assumptions behind TIS on Page 3, but none of these descriptions are totally faithful to the literature. To wit:

- The assumption that neurons ignore kHz stimuli is usually not based on firing rates, but the fact that neuronal membranes are effectively low pass filters. The authors are correct that other mechanisms, like hysteresis in ion channels, could nevertheless render neurons sensitive to frequencies well above the membrane time constant.

- Electric fish also serve as an existence proof for demodulation of electrical signals; envelope extraction appears to occur in other sensory signals as well. Thus, there are principled reasons to expect AM.

- I have never seen a claim that TIS is uniquely able to target deep structures due to reduced tissue impedance. There are both theoretical (Huang and Parra, 2018) and experimental work in humans (Louviot et al. 2022) and non-human primates (Krause et al. 2019). Indeed, it would seem that reduced impedance of the skin, skull, and CSF might reduce, rather than enhance, penetration into the brain instead. As I understand it, the appeal of TI is that it can stimulate deep structures without affecting everything else on top of them.

Additionally, I think the claim is not that there's something special about amplitude modulation per se, but that the AM occurs in a frequency range (~10 Hz) where it can have a larger biological effect. This could be true even if the AM is, in physical terms, much weaker. I believe this is the argument behind e.g., Esmaeilpour et al. (in vitro) and Viera et al. (in primates).

In summary, this paper makes interesting but very strong claims that I'm not sure are as generalizable as the authors claim.

Reviewer #3

(Remarks to the Author)

This manuscript presents a quite comprehensive study of the effects of temporal interference stimulation (TIS) on peripheral nerves, measured in invertebrate and in human subjects. The study was well conducted, and the results represent a good contribution to the understanding of nerve stimulation using kilohertz-frequency signals. More specifically, the authors compared 4-electrode temporal interference against 2-electrode amplitude-modulated stimulation to demonstrate that the effects of the former are due to presence of kilohertz carrier rather than a purported interference on the target tissue, namely, the nerve fiber.

The authors did a quite good job in describing supporting evidence for their rationale, but in my opinion the introduction is unnecessarily long, it was written as a review. I value all the background material presented and thoroughly discussed here, but it seems it is too much for an Introduction. The interpretation and assumptions under TIS have been reviewed and discussed elsewhere (again, see for example Ward et al. 2009 cited by the authors), and similar arguments to challenge (some of) the assumptions have been presented in the past (again, Mirzakhaili et al, 2020).

In addition, the work may lack a bit of originality for two reasons:

1) The experimental measurements in human subjects presented here add up to the existing body of work, and the measurements in invertebrates seem quite novel indeed. However, comparing the effects on both motor responses and discomfort using interferential, Russian, and other modulated currents was performed in the past, as reviewed in a work cited by the authors (Ward et al, 2009).

2) Theoretical work has discussed the biophysical mechanisms of temporal interference with fairly similar arguments (Mirzakhaili et al, 2020, also cited by this work). Written as it is, it would appear that many ideas are original. Again, the review-like approach of the Introduction may play a role here. I would recommend significantly shortening the Introduction and Discussion, acknowledging previous work in the argumentation, and moving faster to presenting the experimental results.

Now, a major issue for me is the theoretical component of the study. Indeed, I'm having trouble to understand at all the contribution of the modeling work. For starters, the geometry not only looks too simplistic for a FEM approach but also seems to lack the details needed to establish the differences in activation regions due to different stimulation modalities. For instance, Fig 4 shows that a rather large electrode imbalance of 4x amplitude would result in a small displacement of the region of tonic activity. More importantly, the second derivative of the potential does not account for the nonlinear dynamics of nerve fiber needed to determine action potential generation, so the SMI defined here may be quite off. In addition, the model was not validated with experimental data, so it lacks predictive value. Instead, it was used to reinforce and illustrate the idea of modulated vs unmodulated regions, an idea that was, nonetheless, already demonstrated using a much more comprehensive model that included nonlinear HH-based myelinated fibers (Mirzakhaili et al, 2020).

As well, a long paragraph in the Introduction is devoted to challenge assumption (3), but I think the argument is somewhat dubious. The power spectrum of a monophasic rectangular pulse of 0.2 ms is very different to that of a 5 kHz sine wave, regardless the former having a 5 kHz component. So it is highly imprecise to claim that these signals will encounter similar impedance in the skin.

Finally, the Discussion repeats many of the ideas developed in the Introduction, with little-to-none mention to possible limitations of the findings. Further, it caught my attention that very little is discussed about the differences and/or similarities observed between the measurements in myelinated (human experiment) and unmyelinated (locust experiment) fibers. Again, a more complete modeling approach could have supported hypothesis related to the differences in the ion channel dynamics of both type of fibers.

In sum, a more concise manuscript emphasizing the contribution of the experimental measurements, accompanied by a more complete and detailed model suitable for studying nerve fiber dynamics, may be needed for further consideration.

Reviewer #4

(Remarks to the Author)

This manuscript describes a study exploring the potential mechanisms of neural activation during temporal interference stimulation (TIS). The study explores these mechanisms in the context of peripheral nerve stimulation (PNS) and utilizes multiple approaches, including a locust model, experiments in healthy human subjects, and computational modeling. The authors consider multiple conditions in their experiments, including simple kHz frequency stimulation, amplitude modulated kHz frequency stimulation, and TIS. The authors look at the recruitment profiles as a function of frequency (i.e., strength-frequency (s-f) curves) for these different waveform paradigms and found that these curves overlapped. Based on their results, the authors conclude that the mechanisms driving peripheral nerve excitation are the same between kHz frequency stimulation and TIS. These results suggest that, at least in the context of PNS, the possible practical advantages of TIS can more easily be achieved with amplitude-modulated kHz waveforms.

Since 2017, TIS has been pursued with renewed interest as a potential non-invasive approach to modulate activity in the central nervous system as well as in the PNS. However, studies have emerged questioning the ability for the TIS fields to effectively penetrate and reach deep neural structures. Furthermore, research has suggested that the amplitudes necessary to directly activate deep neural structures would be higher than safe amplitudes and would also produce side effects due to modulation of off-target structures. Therefore, further studies are necessary to characterize the neuromodulatory effects of TIS and to evaluate its ability to selectively target neural structures. Due to the continued interest in TIS technologies and their use in clinical trials, the current manuscript is timely and its results represent a significant contribution to the field of neurostimulation and TIS. However, there are several issues that should be addressed before the manuscript is considered acceptable for publication.

In general, the manuscript is well written. However, throughout the manuscript, there are several grammatical errors and places where the wording and structure could be improved.

The authors should make it explicitly clear in the Introduction and/or Discussion sections that they are investigating the mechanisms of suprathreshold TIS. This distinction is important because many people in the field now assume that TIS works via subthreshold mechanisms of action.

At the end of the Introduction, the authors conclude that “some other, parallel, mechanism of demodulation is not present.” However, if this was true, then why do they observe a U-shaped response as a function of modulation frequency in which there is a modulation frequency that provides the minimum threshold (e.g., 5 Hz in Figure 3c)?

The authors state that they utilized electrode orientations that would achieve the maximal modulation index aligned with the target nerve. How did they determine these orientations? Were the overall field orientations similar for both the TIS configuration and the configuration for AM-SW, constant SW, and SW burst?

The authors could consider providing additional discussion regarding the location and quality of the paresthesias that were observed in the human subjects during the different types of stimulation. These results could have interesting implications for the neural response to the different types of surface stimulation.

The authors need to specify how many experiments were performed for each set of results and each figure.

Figure 4

-Show dimensions in part d and provide units in the plot of AM

-Are the authors considering the activating function or amplitude modulation of the activating function?

-The authors used their experimental current threshold measurements to estimate the stimulation threshold value of the activating function in their computational model and obtained a threshold of 2000 V/m². There are potential issues with translating absolute experimental values to a computational model. How did this estimate of 2000V/m² compare to other studies? The authors could run simulations with a multicompartment axon model and look at the actual activating function values required for activation within their computational model and see if the values based on their experimental thresholds are close to the model predictions.

Sentences 4-6, First paragraph, Page 16: The authors discuss the issue of specificity and conclude that it does not happen with TIS based on their results. However, it is not clear if the experimental models utilized in this study provide a means to test specificity in which they would also need to consider off target effects. The authors should provide discussion regarding this potential advantage of TIS and the limitation of their experimental models and the continued need to test for off-target effects in TIS.

In the Discussion section, the authors should describe some of the major limitations of their study.

Minor comments

Bottom of Page 1: The authors should specify what they mean by “addressed” in the statement “This means that only superficial targets can be addressed.”

Top of Page 2: Instead of just specifying > 1 V/m, the authors should provide estimates of the range of field strengths required for direct neural activation and the field strengths that are estimated to be possible during TIS utilizing safe current amplitudes.

Page 2: Provide a relevant reference(s) for the sentence beginning with, “A suggested approach to overcome this basic issue is using high-frequency electric fields,…”

Second sentence, Second Paragraph, Page 2: The authors should explain what they mean by “follow.”

Sixth sentence, second paragraph, Page 3: The authors should provide a better description of the s-f plots. Explain the accommodation region at low frequencies and the thermal dissipation effects that begin to overtake direct electrical stimulation at high frequencies.

Bottom of second paragraph, Page 3: The authors state that kHz frequency stimulation can lead to activation due to the summation of subthreshold depolarizations because at kHz frequencies successive half cycles arrive within the absolute refractory period of neurons. However, because the cell has not generated an action potential, is this response related to the absolute refractory period?

Figure 1

-The authors describe the characteristic s-f curve for sinusoidal stimulation with the threshold minimum around 50Hz. How dependent is this minimum on neuron type?

-The authors should consider providing labels for the dashed green and blue lines in Figure 1d.

Figure 2

-Why is the plot for the constant SW, ramped shown in part e not included in the plot for part d?

Figure 3

-Why did the authors utilize a carrier frequency of 2.5 and 3.0 kHz in the locust and human experiments, respectively?

-Figure 3c – it seems unnecessary to include the text stating the carrier frequency, AM, n, SEM, etc. within the plot. But these values and what the gray shaded region represents should be described in the figure legend.

-Figure 3a-b – the authors should describe the number of experiments included in this data

Top of Page 11

-The authors describe recordings of the low-frequency AM envelopes inside of the mesothorax. The authors could consider adding a figure showing these recordings.

Bottom of Page 13: The authors may want to consider providing a basic description of the model design here within the Results.

Second sentence, First paragraph, Page 16: provide the relevant reference(s)

Version 1:

Reviewer comments:

Reviewer #1

(Remarks to the Author)

The authors have extensively revised their original manuscript and have properly addressed majority of my concerns and comments. Most importantly, they have conducted new experiments to investigate some of the questions raised by me and/or other reviewers and have perform statistical analysis to investigate if their findings are statistically significant or not. Understandably, a few of my questions were not directly answered as they seem to be questions that the authors have decided to investigate for future studies, or they were already part of on-going investigations. I do look forward to learning more about them in their future publications.

Considering the current manuscript, I have only a few minor questions:

1- The authors have stated that “unmodulated sine has, on average across frequencies, lower threshold than modulated waveforms” for the results shown in Figure 3B (locust motor response). Next, for the human motor response (Figure 3F), the authors have stated that “TIS has the average lowest threshold, however we would indicate that this is due to a combination of its modulated nature and the 2x applied current, making it the clearest perceptual”. Therefore, I am not sure why the argument the authors are making for lower threshold for TIS in humans (both sensory and motor) does not apply for the locust model as they don't see lower thresholds for TIS in their animal model. The authors should provide some explanation why they see a different order for what stimulation has lower threshold for the animal model vs. human subjects.

2- Considering the results for the animal model and human subjects in Figure 3B and D, the s-f curves, the relation between the current threshold and the carrier frequency seems to be nonlinear for motor response in the lower carrier frequencies but the response seems to be look linear for the sensory response (Figure 3F). It would be helpful if the authors could provide some explanation for this behavior.

3- Figure 5 suggests that for some amplitude modulation frequencies, the motor threshold and sensory thresholds are closer to each other compared to the others (is the data for 2 Hz for sensory missing?). The authors should explain why such

difference exists and if it having these thresholds close to each other beneficial or not.

Reviewer #2

(Remarks to the Author)

Thank you to the authors and editors for considering my comments on Opančar and colleagues' manuscript "There is no biophysical distinction between TIS and Direct kHz stimulation of peripheral nerves" (NCOMMS-24-55626). The revised manuscript contains much more detailed descriptions of the experimental procedures and formal statistical analyses of the results. These are essential for scientific publications. On the other hand, these new analyses seem to undercut the paper's main claims.

The authors' main argument is that if strength-frequency curves are equivalent across the stimulation modalities, those modalities must act via similar mechanisms. The curves in Figures 3 and 4 certainly do look alike, but the new analysis suggests a significant effect of waveform in two conditions** and a marginal one in the other ($p=0.07$, but with a $N=10$ is hard to interpret). The authors do argue that this difference may be driven by their readout/task and refer to the qualitative similarities between the curves instead. That may be, but I'm uncomfortable recommending a paper where all of the formal analysis is ignored. I'm also not convinced that failing to find a significant difference would be the right way to demonstrate this; an equivalence test would be more appropriate.

The Time-Over-Threshold analysis is interesting but it would be nice if it were tested or analyzed a little more rigorously, perhaps by plotting electrode position or current ratio as a function of finger position.

Additionally, the manuscript still contains a very long and assertive review of the literature on kHz stimulation. As I and the other reviewers noted, there is useful material in here, but much of it is already available elsewhere (e.g., the Mirzakhaili et al. model is discussed at great length, including much of the material in Figure 2 and 7) and it's unusually long (8 pages). There is useful material in here, but the empirical portion probably won't resonate with the audience that most needs to read the review: I am skeptical that most human NIBS users are familiar with strength-duration curves, for example. In the cover letter, the authors describe this as an empirical test of Mirzakhaili et al.; this would be a much more succinct framing.

I do agree with the authors that there has been a lot of hype/misconceptions about TIS and we should better understand if/how it works before claiming it is a panacea. However, this work needs to be done and communicated carefully.

Misc:

** $p < 0.07$ is alternately described as significant (p. 13) and not significant (p. 12). Either way, I would be reluctant to interpret this as strong evidence against a difference, given the sample size.

The title is confusing. I understand that the authors are arguing that the same biophysical mechanism underlies TIS and "direct" kHz stimulation, but the outcomes are (per their own data), quite different. In addition to the tonic modulation, TIS produces phasic modulation at the foci. Thus, a title like "The same biophysical mechanism is involved in..." might better reflect the claims.

Figure 2: It'd be helpful to put the Reilly equation in the text itself, so it's searchable.

Page 12: Did the authors run an entirely new set of experiments or analyses? The finger used as a biomarker seems to have switched from middle to ring?

Page 13: Subjective reports. It's not clear to me what this adds to the mechanistic story that forms the main thrust of the paper.

Page 22: "Based on the potential of the unmodulated sine carrier to stimulate neural tissue, it is arguable if this can be considered a "sham" condition versus the experimental condition of interfering currents and AM."

This strikes me as exactly backwards. Most uses of TIS (in the CNS) test a hypothesis regarding the role of neural activity at a stimulation focus and the AMF frequency. In this sense, a "sham" condition using the one of the carriers accounts for any off-target effects. A no-stimulation condition would be useful for interpreting null results (TI == high frequency sham), but a main effect of TIS seems interpretable to me.

Page 23: "Moreover, in many cases, we would argue that a premodulated kHz stimulation arrangement may provide some key benefits of TIS."

Could the authors elaborate on the "key benefits"? Given the argument that TIS is equivalent to direct stimulation, why not just apply a sine wave at the AM frequency?

Reviewer #3

(Remarks to the Author)

The manuscript was greatly improved, and the text in general flows much smoother now with a bit shorter Introduction and

revised discussion. Nearly all Figures were revised, and I think they better convey the main message of the manuscript now. I still have one concern about the modeling work. I'm not convinced that the TOT is a more intuitive metric. For starters, since a frequency domain approach was used, A1 and A2 do not depend on time, and therefore it is unclear how TOT can be defined as "the fraction of each modulation period that the nerve is suprathreshold". There is no time involved at all. After looking Figure 6E, I thought that TOT was somehow calculated using A1(t) and A1(t), but nothing in the equations and definitions suggests this. Perhaps simply changing the name of TOT can do the trick in terms of the metric, but certainly a clearer explanation of what is illustrated in Figure 6 for frequency versus time domain is needed.

Reviewer #4

(Remarks to the Author)

Based on the reviewer comments, the authors have made significant efforts to improve their manuscript, including rewriting and reformatting text and figures, performing additional locust and human subjects experiments, and providing additional details regarding their methods and results. These efforts are much appreciated and have significantly improved the quality of the manuscript. This manuscript represents an important contribution to the field and I commend the authors for their efforts. While I am largely satisfied with the current state of the manuscript, I have a few remaining minor comments:

The new sections of text contain several grammatical errors and inconsistent verb tenses that need to be corrected. Furthermore, the structure and formality of several sections could be improved.

Figure 1: The amplitude modulated waveform in Figure 1A is not drawn to scale with regards to the actual carrier and beat frequencies. This is acceptable because this figure is illustrative, but the fact that the waveforms are not drawn to scale should at least be noted in the figure caption.

At the top of page 19, the authors should consider removing the following statement: "Overall, the "U-curve" can be considered borderline significant." This statement is subjective and is not explicitly addressed by the statistical comparisons mentioned in the previous sentences, especially when their results do show several statistically significant differences between the different beat frequencies.

Version 2:

Reviewer comments:

Reviewer #1

(Remarks to the Author)

The authors have addressed my minor comments and I appreciate that the authors have tried to thoroughly answer my concern regarding the comparison between different waveforms and configurations and why comparing some of them directly is not straightforward. However, I am afraid there are several errors in their analytic approach that they are planning to include in their Appendix A in response to my question, which must be corrected before publication:

1- In order to compare the activation threshold, the authors are comparing activating functions. That is a valid approach because action potentials are most likely initiated where the activating function is maximum. However, the authors are comparing the activating function between two configurations at the same spatial location ($x=0$). That spatial location is where the maximum activating function occurs for a single source (the authors have assumed that implicitly). However, $x=0$ is not necessarily where the activating function is maximum for the paired sources. In order to compare the activation threshold between configurations based on activating function, the authors need to compare the maximum of activating function regardless of where they occur. Indeed, the amplitude of activating function for TIS configuration depends on the distance between the pairs of electrodes but so does the location of the maximum amplitude of activating function and that was not accounted for.

2- Regardless of the first issue, the equation the authors are using for the activating function of TIS in the point they are making their comparison is incorrect. The equation for the single source is:

$$V_{xx} = \frac{1}{2\pi\sigma} \frac{(2X^2 - Y^2)}{(X^2 + Y^2)^{5/2}}$$

, which simplifies to $\frac{1}{2\pi\sigma} \frac{1}{d^3}$ for $x=0$ and that is correct. However, X is no longer 0 for the paired sources where they are doing their comparison (note that the equation above assumes a coordinate system with its origin at the source). Therefore, the equations they are setting equal to each other to find a relationship between I and d is incorrect. In fact, the equation for sources that are located at $+l$ and $-l$ would be completely different if the authors want to use a global coordinate system with the origin where the single source is located at.

(Remarks on code availability)

Reviewer #3

(Remarks to the Author)

All my concerns were addressed.

(Remarks on code availability)

Reviewer #4

(Remarks to the Author)

The authors have satisfactorily addressed my comments.

However, unless I am mistaken, I believe there are mathematical errors in the authors' response to the first comment from Reviewer #1. Specifically, with regards to both their equation for current density (J) and the corresponding expression for potential (V).

For a point-source electrode, the potential field in an infinite homogeneous medium should be expressed as: $V=I/(4\pi\sigma r)$

Additionally, I believe there is an issue with their expression for the activating function, A_x . Since the potential V for a point-source electrode is derived in terms of r, and the calculation of A_x requires taking the second-order partial derivative of V with respect to x (along the axon), the authors must specifically account for how r varies as a function of x. Therefore, their expressions for A_x require correction, particularly given that this analysis is now included as part of the Appendix in the manuscript.

(Remarks on code availability)

Version 3:

Reviewer comments:

Reviewer #4

(Remarks to the Author)

I was a little confused by their system and approach and so, yes, my comments were incorrect. The additional information that the authors have provided in response to Reviewer 1 will hopefully help avoid confusion by readers.

(Remarks on code availability)

Article reference: NCOMMS-24-55626

There were several key questions/requests/critiques which were pointed out by multiple reviewers. Therefore, to increase readability, we have split this response letter into a first section, which addresses these common points, which we have taken the liberty of summarizing; and a second section, which addresses each individual reviewer's comments separately.

Section I. Responses to common points raised by reviewers

1. From a practical and clinical perspective, TIS is being increasingly considered for subthreshold stimulation effects, why does our paper focus on suprathreshold phenomena? How are the findings and claims relevant to subthreshold stimulation?

Answer: We agree that our study should take account of the relevance to subthreshold stimulation, because we would argue that our main claim concerning stimulation by unmodulated kHz currents is pertinent to subthreshold stimulation just the same as it is in the suprathreshold regime. The same effect of membrane rectification which is responsible for kHz sinusoids leading to APs also leads to subthreshold depolarization when the amplitude of the sinusoidal kHz current is low. The phasic/tonic dynamics that come from amplitude modulating or not modulating the kHz will be the same whether one is sub- or suprathreshold. We do not litigate if there are other mechanisms of subthreshold AMF which may modulate ongoing neural activity, we just state that the evidence from the experiments we have performed points to the kHz stimulation model.

In terms of motivation and scope: While the human transcranial studies involve amplitudes < 2 mA and are thus necessarily subthreshold (There is consensus that amplitudes < 2 mA are very unlikely to produce E-fields in excess of 1 V/m as would be necessary to reach the suprathreshold regime in human transcranial stimulation ([1] Y. Huang et al., "Measurements and models of electric fields in the in vivo human brain during transcranial electric stimulation," *Elife*, vol. 6, pp. 1–26, 2017, doi: 10.7554/eLife.18834.), the majority of TIS/ICS studies being published in recent years are addressing suprathreshold stimulation effects – this applies to both peripheral (human, animal) and central nervous system targets (animals). The quantity of these studies still outnumbers those which study stimulation effects at a subthreshold level.

In our study the main goal is to elucidate if there is a difference in thresholds between modulated and unmodulated kHz carriers, and see if there is some amplitude-modulation frequency effects and non-carrier associated demodulation phenomena occurring in excitable tissue. Working in the suprathreshold regime gives less equivocal biomarkers. Since these unmodulated/modulated carriers depolarize the cell membranes to the suprathreshold level, we expect that finding to carry over into the subthreshold regime as well with the same dynamics.

We have now discussed the subthreshold picture in more detail in the introduction sections, and in the final discussion part, and we have revised the modeling (new panel shown here) to emphasize the membrane potential dynamics when exposed to a **suprathreshold** stimulus versus a **subthreshold** stimulus. An unmodulated subthreshold sine (2 mA) amounts to a situation analogous to DC depolarization of the membrane (the “envelope” of an unmodulated sine is DC). The transition between sub- and suprathreshold can be seen in the figure above: 2 mA sine leads to membrane depolarization, and the membrane will remain at this depolarized potential at a subthreshold level. Once the amplitude is increased to 3 mA, the membrane potential depolarizes enough to AP threshold. The difference in membrane potential dynamics comparing sub/suprathreshold can be nicely seen in the low-pass filtered traces in the plot above (dashed lines).

2. *N*-numbers for critical experiments, such as threshold comparisons between kHz waveforms, are low, and statistical treatment should be more rigorously applied.

We fully agree that all these experiments are not convincing enough without more robust experimental repetition and statistical analysis, and thus have redone all the experiments with larger subject pools. All the original s-f threshold experiments involved $N=3$ (locusts and humans). In the revised experiments, we now have $N=5$ for locusts, and $N=8-11$ for human experiments. Moreover, to further test the hypothesis that modulated and unmodulated kHz stimulate via the same mechanism we added a new set of experiments on an independent model: median nerve stimulation at the carpal tunnel. This model specifically targets sensory fibers which have a lower threshold than motor fibers, which allows verification of our claims on a model which is not neuromuscular (new **Figure 3C**). We also interrogated the AMF dependence on the carpal tunnel for $N=7$ additional participants (new **Figure 5D**). In the initial submission, unmodulated sine condition was tested at only three carrier frequencies, now it is tested for every frequency and in every s-f experiment. Moreover, the s-f curve for locusts was retested up to 100 kHz, as described in the next answer.

3. *The authors comment that carrier frequencies > 12 kHz stimulate inefficiently, and no envelope demodulation is observed using these higher carrier frequencies approaching 1 MHz - but they only describe the effect and do not show any data.*

We are grateful for this criticism, as it challenged us to design an experiment to properly compare unmodulated and modulated kHz stimuli over this large range. Due to limitations of experimental equipment and ethical permissions, this type of dataset can only be obtained on the locust model. In the revised manuscript, we now report the s-f curve for amplitude-modulated versus unmodulated sinusoidal

stimuli in a range up to 100 kHz. We hope this experimental data will serve the community thinking about possible stimulation effects in this less unexplored frequency range. This data backs up our previous claim of a lack of evidence for a parallel mechanism demodulating the amplitude-modulation frequency. There continues to be no significant difference between modulated and unmodulated stimuli.

Reference Note: Shortly after submission of our manuscript, the following article was published: Caldas-Martinez S, Goswami C, Forssell M, Cao J, Barth A L and Grover P. 2024 Cell-specific effects of temporal interference stimulation on cortical function *Commun. Biol.* 7 1076

This work provides *in vitro* evidence for the fact that unmodulated kHz sine waves can depolarize cells to the same extent as modulated kHz. We acknowledge that this strengthens our claims, and we now cite this reference. We would note that this work is complementary to our own. They work in different systems and models, and do not evaluate carrier frequency dependence (save for two frequencies), which is a mainstay of our paper where we build our conclusions on strength-frequency characteristics.

Section II. Individual Reviewer responses

Itemized responses to the referee comment are found below, with original comments in red, and our response in blue. All text changes are marked in the manuscript with yellow highlights.

Reviewer #1 (Remarks to the Author):

Temporal interference stimulation (TIS) has received major attention in recent years because it has been advertised to provide targeted deep brain stimulation noninvasively without major side effects. Yet, there is still no consensus about the mechanisms of action for TIS or whether it can be safely and effectively be applied in humans.

The current manuscript investigates if TIS is different compared to pure kHz stimulation for peripheral nerve stimulation. The current study is different compared to most other TIS studies because it studies the applicability of TIS in peripheral nerves and not the brain and to my knowledge, it is the only study that has both animal model and human subjects. Altogether, the authors have designed experiments to address some of misunderstandings that are commonly believed about TIS. In summary, using the results from their animal model and human subjects, the authors have reported that there is essentially no difference between TIS and direct kHz stimulation. While their results are compelling, I believe the authors need to provide more evidence for some of their claims and I am not sure if some of the results provided fully support their claims. I hope my comments can help the authors to improve their manuscript:

We appreciate this positive endorsement of our work and the approach of combining animal and human experiments, and the highly constructive criticism to improve the manuscript.

Introduction:

1- The authors have thoroughly reviewed the prior TIS research in their introduction. It is especially encouraging that they have acknowledged that the concept of TIS is older compared to what is commonly believed. However, the authors have only focused on direct activation of cells using TIS in their introduction. Yet, recent studies are hinting that TIS might be more of a subthreshold stimulation method. Even though such alternative arguments about the effectiveness and mechanism of action for TIS are related to the central nervous system and not much related to peripheral nerves, I believe the introduction can be improved if the authors discuss them.

We fully agree with this comment: we must carefully sharpen the distinction between subthreshold and suprathreshold stimulation effects. We have added considerations of subthreshold stimulation to the introduction and discussion sections. Otherwise, we have covered this question in the common response 1 above.

2- Figure 1-d is a little hard to follow and can be improved by better organization and labeling. For example, the authors have mentioned in the caption that all the dashed lines are low-pass filtered but the resting potential is also a dashed line, and I don't think it is low-pass filtered.

Thank you for this point, the original design of the figure was very hard to follow and ineffective in getting the main points across, we have redone the modelling from Figure 1 and split Figure 1d into two separate panels to make things clearer. Using a dashed line for the resting membrane was indeed confusing. To make things easier, we plot $V_{\text{membrane}} - V_{\text{resting}}$. This also makes it easier to see the subthreshold stimulation calculations. Now these calculations are all in a revised Figure 2, together with a newly added calculated s-f curve.

3- It is not clear what is the electrode configuration for the results shown in Figure-1d but I believe the node that the authors are showing the results for is not where the action potentials are generated because I do not see much amplitude modulation in the voltage traces (blue or green lines). I recommend showing the results on a node where amplitude modulated stimulation is strong so that it can be compared with the authors already shown in Figure1-d to demonstrate how current summation is different in presence/absence of amplitude modulated stimulation.

Thank you for these points. We agree that Figure 1 needed a complete revision, the original was too “busy” and important details are either hard to follow or missing. We have extensively revised the whole figure making it easier to follow and get across the main points about the rectification mechanism behind kHz stimulation. We have now included the exact configuration used for the simulation in the figure inset, showing the node at which we evaluate the membrane currents and potentials. The node in question is directly between the opposite pair's electrode which is the location where the maximum modulation of the activation function in the TIS configuration occurs, and the location where the action potentials are initiated. Apart from subthreshold and suprathreshold sine we have now also included 5 Hz TIS in which the modulation of the membrane potential is clearly visible going between maxima at $t_{\text{max}}=0, 200$ and 400 ms and minima at $t_{\text{min}}=100, 300$ ms.

Results:

1- The main take away from the current manuscript is that TIS is not different from direct kHz stimulation.

The main point for this argument that authors are using is based on their results that the threshold of activation is not that different from direct kHz stimulation of amplitude modulated stimulation (AM-SW; the authors need to define this term in their manuscript). The results that they are showing in their manuscript (Figure 2-b and Figure 2-d, for example) supports this argument. However, the authors need to run proper statistical tests to argue that the thresholds are not significantly different from each other. I understand that the N is low for the animal model and human subjects and the authors might not be able to conduct proper statistical tests. If that is the case, then the authors need to either increase their N, which is relatively low or state clearly that they cannot show that TIS is different from direct kHz or AM-SW stimulation in terms of activation threshold.

To eliminate any ambiguity on the waveforms we are comparing, we now plot them in a new and simplified intro Figure 1. We agree a proper statistical analysis must be reported, and higher N numbers needed. We now report statistical tests more carefully. In the s-f experiments, the differences between the tested waveforms at a given carrier frequency are not significant, except in some cases like in the locust where unmodulated sine is borderline significantly lower than modulated waveforms. In the new sensory experiment at the carpal tunnel, there are some statistically-significant differences between waveforms, which is a result of the nature of experiment which relies on subjects reporting sensory percepts. Modulated “on-off” waveforms are more readily detected than unmodulated ones, producing a shift in the absolute values of the threshold values along the s-f curves. These aspects of statistical significance, however, do not in our opinion affect the conclusions about the s-f curves proving mechanistic equivalence across the kHz stimulation types.

2- The authors have stated that the participants have felt “less comfortable” with SW burst stimulation. It would be interesting to provide more specific feelings (pain, itch or heat, for example) in case the participant mentioned it. Also, was their level of “uncomfortable” feeling related to the frequency of stimulation.

We have expanded our discussion on subject-reported feeling for different waveforms. In case of the burst modulation, we should be specific to report that at the threshold value (sensory, motor) there is no acute discomfort, yet participants reported that subjectively, burst feels less pleasant than the amplitude-modulated waveform. Quantifying comfort and tolerability of different kHz waveforms is a topic of further study in our team, where we are now trying to quantify in detail the differences between burst and AM waveforms, however this is a subject for an upcoming manuscript.

3- At the end of page 10, the authors have used 0.1-1 MHz stimulation to argue that AM envelope demodulation does not occur. First, I could not find any figures associated with these claims. Second, I am not sure how the authors are using changes in the carrier frequency to argue that AM demodulation is not present in their models.

We have now expanded the s-f curve for the locust model out to 100 kHz in a new Figure 4. The argument is aimed at ruling out a carrier independent demodulation mechanism which would work regardless of the carrier frequency. We found that the threshold for stimulation correlates only with the carrier frequency, there is no other apparent mechanism that yields some low-frequency phasic stimulation regardless of carrier frequency. The comparison of modulated and unmodulated kHz (the s-f relations are the same) in this larger range shows that there does not appear to be any carrier independent mechanism present that leads to phasic stimulation. It all follows the familiar s-f relation = there is nothing else apparently going on (except heating). We clarified in the manuscript that this larger carrier frequency range measurement just adds evidence of the lack of some carrier independent demodulation mechanism.

4- Using the results in Figure3-a, the authors have argued that their animal model does not have a “preferred” stimulation offset frequency in which the activation threshold is lower. First, I am not sure how the authors are relating the level of deflection to activation thresholds. Second, the range that they are changing the offset frequency is small (0-1 Hz). Therefore, I am not even sure if they could see a preferred offset frequency in that range. Third, there seems to be a change in the trend of deflection angle at 0.4 Hz, which needs some explanation. 5- The authors have argued that amplitude demodulation through rectification is due to nonlinearities in the ion channels. Yet, they argue that their animal model does not show amplitude demodulation because it has unmyelinated axons. I am not sure how not having unmyelinated axons is related to not being able to rectify stimulations if it is related to nonlinearities of ion channels.

We retract the suggestions we have made concerning unmyelinated axons, and there being some possible difference in the activation thresholds in the animal model versus the human experiments, we do not have evidence for this or any solid theoretical basis to claim this.

We have redone these experiments with higher N and using optimized methods (high-speed video analysis synced to the stimulation waveform) to provide more robust data, and we have reworked the associated figure (new Figure 4A-C). The kink in the original deflection angle plot at 0.4 Hz was an artefact from just having too small a data set. Now, the deflection angle plot appears monotonic, and standard error is low.

6- The authors have mentioned that for 1 Hz frequency offset, they observed 1 Hz phasic response and for 0.1 Hz frequency offset they observed 0.1 Hz phasic response. However, it would be insightful if they plotted the phasic frequency of response for all the offsets they used. Moreover, the authors should show the onset of response relative to the phase of stimulation for all offsets. Similarly, they should show the time when the leg returned to resting position relative to the phase of the envelope.

This is an excellent point and we now add a new graph (Figure 4A) which shows threshold current and movement onset (from highspeed video) synced to the stimulation envelope. Also, to the leg deflection plot, we add another y-axis which plots “time over threshold” for each waveform, which neatly explains the correlation between AMF and motor movement.

7- It is not clear what is i_{\max} that the authors are using to normalize the results especially when there is no point on the plot that is at unity. Also, the authors have reported 20% lower activation threshold at the preferred frequency. However, is this reduction statistically significant? Elsewhere, the authors have mentioned 3 human subjects, but it seems like they have 7 subjects for Figure3-C. They need to clearly state how many human subjects they had and basic demographics such as age and gender for them in their method section. If they had to exclude some of them for some of the results, then they need to mention that and provide exclusion criteria.

The data were normalized by dividing each threshold current value/ i_{\max} current value for the dataset from each participant (i/i_{\max}). However, since not every participant had i_{\max} at the same AMF value, once the normalized data are pooled, the whole set is shifted away from unity. We agree this looks awkward at first glance, yet we think this way of normalization makes more sense and gives the reader a feeling for the relative magnitudes in a way that [0,1] normalization would not. In the revision, we now normalize to i/i_{\min} to better reflect the “optimal minimum” AMF around 5 Hz, and that higher and lower AMF result in effect threshold increasing (revised Figure 4D), though the same “unity problem” remains here. We have explained this in the text.

8- Is there any explanation why there is more variability in the normalized stimulation thresholds at the lower beat frequencies?

Yes, the explanation for this lies in the experimental method. Participants and experimenters find it easier to determine a sensory or motor threshold for a stimulus which either has a rapid onset, or has some rhythmicity. Especially with sensory thresholds these kinds of rhythms make their determination more obvious. When AMF goes to low extremes (i.e. slow, long ramping of intensity), or also higher frequencies > 20 Hz, there will be more variability in when the participant will indicate the exact threshold point. We take this question, along with others, as a prompt to provide detailed experimental descriptions in the revision we are submitting. Part of that is explaining this effect of variability in the AMF experiment.

9- The authors need to report more details regarding the experiment on human subjects when they changed the frequency offset. For example, it would be helpful if the author provided similar results to Figure 3-A for their human subjects. Additionally, the same metrics that I suggested to add for the animal model can be added for the human subjects. Specifically, the frequency of phasic response and the onset/offset of the response compared to the phase of stimulation beat.

We have added these missing experimental details to the results and also the methods sections.

10-The authors have indicated that they “arbitrary” used flexing of the middle finger to define thresholds. Then, if we assume that movement of the middle finger was the target, how did the movement of other fingers compare to the movement of the target. This is important information that needs to be provided for all the experiments they conducted (stimulation types, carrier and offset frequencies). Even though the authors did not see an obvious difference in thresholds, there can be other subtle differences in stimulation types that the authors did not investigate. For lack of a better word, the movement of other fingers can be considered as a side effect, which could be different across different experiment protocols they used.

We have now expanded the description of the experimental methods: We are stimulating the motor branch of the median nerve that enervates the *flexor digitorum superficialis* (FDS) muscle. The median nerve/FDS stimulation model is well-established and relatively easy to landmark and to achieve reproducible stimulation results across participants.

11-The authors have provided some modeling results in Figure 4. First, they need to provide details of their model such as boundary conditions and some characteristics for the computational mesh they used. Second, I am not sure in what direction the authors are showing the activating function. I assume they are looking at the second derivative in the x direction but that needs to be clarified. Also, I am not sure about the average activating function and how this non-scalar metric was averaged. Regardless, I am not sure about the purpose of the results the authors are showing in Figure 4. I understand that they are arguing that by changing the current ratio it is possible to alter between phasic and tonic response for human subjects. However, they did run experiments on human subjects in which they altered the currents. In general, I am not sure if I could find any results showing that the author showed tonic response in human subjects.

We have taken this prompt to properly detail modeling conditions and fill in missing info in the methods section.

The activating function is always calculated in the x direction which is the direction of the nerve which is now clarified in the manuscript text. For a single current source in the frequency domain, the activating function in x direction $A = d^2V/dx^2$ is, as the electric potential V , a phasor quantity consisting of amplitude and phase. In the TIS case, the activating function at any point will result from the summation of activating functions produced by the two current pairs I_1 and I_2 . The total activating function amplitude can be obtained by adding the phasor values corresponding to current pairs A_1 and A_2 using law of cosines to obtain $A(t) = \sqrt{A_1^2 + A_2^2 + 2A_1A_2\cos(2\pi\Delta f t)}$. This total amplitude changes between $A_{MIN} = |A_1 - A_2|$ and $A_{MAX} = A_1 + A_2$ with frequency Δf which is the frequency difference between the carriers $\Delta f = |f_2 - f_1|$. The average activating function amplitude we used is the average of A_{MIN} and A_{MAX} .

We however agree with the reviewer that the average activating function might not be the most relevant metric to plot and in the revised figure we chose to plot the maximum activating function amplitude A_{MAX} as ultimately this is what determines if the nerve will be stimulated or not.

We have revised this figure to start with experimental demonstration of the phasic/tonic problem, and then we use the model to support this finding. Overall, the computational model has been refined and improved, and all details are now provided.

12-The authors need to provide details of their modeling for Figure 5 such as boundary conditions.

Likewise, we have added these methods details in the revision.

Discussion:

1- Similar to my first comment for the introduction, the authors should provide some discussion that their results can rule out how TIS might be effective if its main mechanism of action is a subthreshold phenomenon.

We have added, as discussed in the common answer 1 above, our perspective on how this may inform the subthreshold stimulation picture as well.

2- The authors have discussed how the TIS leads to a “constant” DC shift. I am not sure this is an accurate statement. I believe the DC shift will be constant where there is no amplitude modulation (in some regions of TIS and everywhere for direct kHz stimulation), but it will have some dynamics where there is amplitude modulation. The authors should comment on this and the investigation I suggested for Figure 1-d would help to clarify this argument.

We completely agree with the reviewer on this point, TIS will only result in the constant DC shift close to the electrodes where the signal is largely unmodulated. We believe the revision to the mentioned figure (now figure 2C) helps to clarify this. The “DC stimulation” effect is relevant to unmodulated carrier at the subthreshold level. Essentially, the effect on the cell membrane potential is analogous to the application of a subthreshold DC stimulus.

3- The authors have argued that an amplitude modulated kHz stimulation can provide key benefits of TIS more straightforwardly. The authors should be more specific about the benefits. For example, I am not sure the premodulated signal can be completely steered. Is premodulated stimulation provide phasic response throughout the tissue or it is going to lead to tonic response near the electrodes depending on its amplitude? Is the common benefit limited to feeling less uncomfortable? 4- Considering premodulated stimulation, if it is going to provide a phasic response in the majority of the tissue, is there going to be a phase difference between the regions? For example, will the regions near the electrode respond at an earlier phase? And is that going to be a benefit or disadvantage? There are some arguments (Barra et al. 2024 bioRxiv, for example) that completely phasic response leads to less “natural” neuronal response and TIS can feel more natural because the phase of the response is more variable. Then, can premodulated stimulation achieve a naturalistic response?

In the discussion section we have now better compared and contrasted 4-electrode TIS versus premodulated kHz (2-electrode TIS as it is sometimes called). The multielectrode arrangement definitely provides the advantage of steering, however it will always be bimodal: tonic and phasic. The 2-electrode should essentially always be phasic everywhere, and with kHz wavelengths there should be no phase differences. As to the phasic nature, however, this type of AM-sine still ramps the intensity of the kHz stimulus, thus when one considers the “synchronicity” aspects raised by the Barra 2024 manuscript, this type of stimulation provides “low synchrony”. The kHz carrier itself enforces lower synchrony, while the amplitude ramping also creates a temporal distribution in when firing begins in axons which are being stimulated. Thus, from the “naturalistic” response point of view, we think the two-electrode premodulated kHz is a compelling approach, and this is now a subject of intensive investigation in our team.

Experiments:

1- The authors have mentioned that they increase the stimulation to find the threshold. However, some other key information is missing. Did they also increase the carrier frequency in increasing order? If so, did they search for the threshold from 0 at each carrier frequency? If not, did they start from the current from the lower carrier frequency? Same question can be asked about experiments in which the authors altered beat frequency.

We have now added these missing details to the results and methods sections, describing the ramping procedures and threshold finding protocols. The presentation of carrier frequency or amplitude modulation frequency was always randomized by the python code running the stimulation. The current amplitudes for the beginning of the ramping were grouped for < 5 kHz (where lower currents are necessary) and > 5 kHz, to not “waste time” with a ramp from zero in the latter case and to make each trial of similar length. These details can be found in the revised methods section.

2- What was the order for finding the threshold for different stimulation protocols? For example, did the author perform TIS first, SW first? And did they perform all the experiments for one stimulation type and then switch to another waveform or they altered between the stimulation types for each carrier frequency.

In the case of waveforms, as well as carrier frequency or AM frequency, presentation to the subject was always random. In the case of carrier or amplitude modulation frequency, randomization was ensured by the python program running the stimulation. The only exception to this randomization rule was in the locust and human experiments, where 4-electrode TIS was always conducted prior to other 2-electrode experiments. This was due to the importance of properly placing electrodes relative to the central point of the TIS electrodes. Thus, TIS was done first, then the 2-electrode position was always set to be at the midpoint between the two TIS electrodes to ensure as accurate a comparison as possible. We have now specified these missing details in the results and methods sections.

3-It is not clear how long the authors stimulated at each carrier/beat frequency. Was it always the same duration or was it a function of carrier/beat frequency?

These missing details were also added. The protocol was designed to try and match total experimental trial times.

Reviewer #2 (Remarks to the Author):

Thank you for inviting me to review “There is no biophysical distinction between TIS and Direct kHz stimulation” by Opančar et al. (NCOMMS-24-55626).

Although originally discovered in the 1950s, the past few years have seen an explosion of interest in temporal interference stimulation (TIS) but it remains unclear how this technique works, or indeed if it does anything at all. Conventionally, TIS is thought to act through the superposition of two high-frequency (kHz) electric fields: it is assumed that each individual field oscillates too rapidly to entrain neural activity, but their overlap produces a slower “beat” (or amplitude modulation; AM) which, when demodulated, somehow affects the neurons.

Here, Opančar and colleagues argue for an alternate interpretation of TIS where the neurons are directly driven by the kHz carriers and apparent AM occurs just because beats reflect periods where the two carriers are jointly maximal; no envelope extraction or demodulation is required. To support this hypothesis, they perform two simple experiments to describe the strength-frequency curve, using limb (in locust) and finger (in human) movement as a readout. In both cases, they find that TIS, amplitude-modulated sine waves, and bursts behave similarly across these preps.

I am very sympathetic to the idea that the effects of kHz stimulation need to be examined more carefully, though I do think Mirzakhilili et al. (2020; Cell Systems) has thoroughly covered this ground already in terms of modeling and lit review.

We agree that Mirzakhilili et al., published in 2020, provides a model for how TIS works and is essentially correct and accurate, and we refer to it extensively in our manuscript, citing it as a motivation. However, that work is 100% theory and modelling, without experimental data. The authors themselves state in their conclusions that they encourage experimental follow-up, especially in peripheral nerve models, to confirm their conclusions. This is a direct prompt behind performing the study that we did, and we believe our (now supplemented and revised) experimental data strongly support the theoretical model outlined by Mirzakhilili and colleagues.

In terms of confronting the debate about stimulation via unmodulated and modulated kHz E-fields: Encouraged by this reviewer point, we have revised the manuscript to emphasize our methodological novelty in measuring strength-frequency (s-f) characteristics, analogous to the well-established strength-duration (s-d) curves, but in the frequency domain. By measuring the frequency dependence, one can precisely discriminate if there is some difference in threshold based on modulating the carrier or not. While s-f curves of sinusoidal stimuli have been conceptualized in much older literature (notably starting with Bernard Katz, Nerve Excitation By High-Frequency Alternating Current *J. Physiol.* **96** 202–24, 1939), to the best of our knowledge, the notion of constructing s-f curves in the field of TIS has never been done. We would argue that our finding that modulated and unmodulated kHz have an identical s-f relation in the insect model and the two human models gives unequivocal evidence that the stimulation of TIS is explained by the kHz stimulation effect (= Gildemeister effect, temporal summation effect). Moreover, tracking s-f relations of other modulated waveforms can be a useful method to tease out if there are indeed discrete AMF demodulation effects in other situations than the ones we have measured.

The experiments presented in this paper do add to the debate, but very little data is shown for some of them (especially, the 10 kHz - 1 MHz experiment) and the methods are also scanty. The procedure for measuring thresholds in humans, for example, only gets a sentence: how many repetitions collected for each subject? How were they structured?

We take the point that more data and higher N-numbers would strengthen our manuscripts claims. We have now repeated all experiments (human and locust) with larger N-numbers, added another human experiment (median nerve stimulation at the carpal tunnel), and add a new figure (Figure 4) to show the higher-frequency region up to 100 kHz (only in locusts, for reasons of ethical permission). Unmodulated sine is now done in all experiments and at all carrier frequencies.

We have carried out the following additional experiments:

- Threshold determination s-f curve N=8 human participants, motor fiber stimulation, *flexor digitorum superficialis*.
- Threshold determination s-f curve for N=10 human participants using a new model: sensory fiber stimulation of the median nerve at the carpal tunnel. We chose this to specifically target sensory fibers with typically the lowest electrical stimulation threshold, as the carpal tunnel contains primarily sensory fibers which have a low threshold for electrical stimulation (lower than motor).
- AM-frequency dependence for N=7 participants for both motor, and sensory (new experiment), stimulation of the median nerve.
- Threshold determination s-f curve up to 100 kHz, N=5 locusts (N5 motor neuron stimulation, AM-kHz versus unmodulated kHz)

Method details have been extensively revised and updated to explain how all experiments are performed, including randomization of carrier frequency and AMF presentation, etc.

However, my biggest concern relates to interpretation: how do these data inform understanding of human TIS? This is not related to the model system, but rather the nature of the stimulation. Human TIS is generally subthreshold (well below it, in fact) and applied centrally. It is not clear to me that we can sensibly extrapolate from threshold measurements in the periphery to cranial TIS at ~0.5 V/m AM. I would appreciate a lot more discussion on this point. The authors do refer to the Gildemeister effect, but it's not clear to me if that's enough (and if it is, shouldn't there be interesting dynamics in the data, as the effect builds up over time?).

We agree that the extrapolation of our findings to subthreshold stimulation is absolutely critical, and we have tried to drive these points home in the revision. We kindly refer the reviewer to the sub/suprathreshold discussion in common answer #1.

Similarly, the authors also seem to attack some strawman positions. For example, they outline three assumptions behind TIS on Page 3, but none of these descriptions are totally faithful to the literature. To begin with:

We are essentially in agreement with the reviewer's sentiments here. Indeed, the term "strawman" came up in our discussions during preparation of the manuscript, and we debated on whether to include these positions, and how to describe them – because we agree that these positions are weak. However, many of these claims are in fact often repeated in not only the older literature, but in many publications from the past few years. Overall, we have, in line with other reviewer comments, shortened the introduction and completely rewritten it, and refined our discussion of these points. We have removed this structure of "three

assumptions”, and instead briefly mention some of the erroneous assumptions in the literature and rather focus on the question of stimulation of modulated versus unmodulated kHz, which is relevant both to supra/subthreshold stimulation.

- The assumption that neurons ignore kHz stimuli is usually not based on firing rates, but the fact that neuronal membranes are effectively low pass filters. The authors are correct that other mechanisms, like hysteresis in ion channels, could nevertheless render neurons sensitive to frequencies well above the membrane time constant.

On one hand, we would cite that the “firing rate” assumption is often repeated in papers on TIS, and therefore is worth pointing out in our manuscript. On the other hand, we are grateful for the point that we need to confront and better describe the “low pass filter” justification of TIS that has been repeated since the seminal 2017 paper on TIS in the brain as well. We would maintain that the low-pass filtering argument is incorrect. In the power spectrum of TIS, with high-frequency carriers, there is no power appearing at low frequency. A filtering effect may attenuate the carrier amplitude, but a filter cannot extract low-frequency power which is not otherwise there (several theory papers have already pointed this out, e.g. Wang B, Aberra A S, Grill W M and Peterchev A V. 2022 Responses of model cortical neurons to temporal interference stimulation and related transcranial alternating current stimulation modalities *J. Neural Eng.* **19** 066047).

The 2017 Grossman paper indeed claims that neurons could ignore kHz stimuli due to low-pass filtering, and this claim references the paper of Hutcheon and Yarom (Resonance, oscillation and the intrinsic frequency preferences of neurons, 2000). Hutcheon and Yarom describe filtering effects originating from the passive membrane properties of neurons. In that paper, they refer to filtering from the point of view of the magnitude of voltage change on the cell membrane by a given input current:

They consider a constant input current magnitude and the output voltage on the membrane. Thus, due to its higher impedance to low-frequency current, the cell membrane would experience a larger voltage change than it would for a higher-frequency current. This is what this manuscript means by “low-pass filtering effect”. This effect almost certainly explains, in part, the shape of the current-frequency (s-f) curve for higher frequencies: if current is held constant, higher frequencies induce lower E-fields across the cell membrane. This is one of the reasons why the higher the carrier frequency, the more current is needed to elicit the same depolarization effect. In this sense, this passive filtering effect helps explain why kHz sinusoids can be less efficient at stimulation, but this does not in any way confront why amplitude-

modulating the kHz carrier in particular results in stimulation, because invoking the filtering model does create low-frequency E-fields. Moreover, typical biphasic current pulses have phase length of 0.1-1 ms, just like the kHz sine waves used in TIS, therefore the E-field “filtering” at the cell membrane level is not expected to be different for conventional pulsed current stimulation versus kHz sines.

Mirzakhilili et al. (2020; Cell Systems) also argue that any filtering effect would lead to attenuation of carrier amplitudes only, but not result in the appearance of a low-frequency E-field component and cannot explain why phasic stimulation is observable in TIS. Their explanation points entirely to a kHz rectification only as the origin of stimulation effects, and this is what we experimentally arrive at in our paper too. The cornerstone result is the threshold s-f curves and the fact that thresholds for motor fibers, already shown in the first version of the manuscript, and sensory fibers (now newly added in the revision) are essentially not different for modulated and unmodulated kHz sines.

We have now carefully, but briefly, discussed this “low-pass” filtering argument in our introduction, and explain why we think this notion is incorrect.

- Electric fish also serve as an existence proof for demodulation of electrical signals; envelope extraction appears to occur in other sensory signals as well. Thus, there are principled reasons to expect AM.

We agree that there are examples of neural processing which show extraction of AM envelopes, we would suggest (we cite in the revision) the phenomenon of auditory steady-state response (ASSR) as an excellent example. We take this point to discuss this in the text, that there are indeed principled reasons to expect AM extraction phenomena. The point of our findings is that when using kHz electrical stimuli, one has to count with the rectification of the carrier signal which will result in net depolarization of cells. We would agree with others in the TIS community that in brain stimulation there well may be certain situations where phasic stimulation at the AMF is effective. However it seems unlikely one can ignore the possible effects of the tonic carrier.

- I have never seen a claim that TIS is uniquely able to target deep structures due to reduced tissue impedance. There are both theoretical (Huang and Parra, 2018) and experimental work in humans (Louviot et al. 2022) and non-human primates (Krause et al. 2019). Indeed, it would seem that reduced impedance of the skin, skull, and CSF might reduce, rather than enhance, penetrance into the brain instead. As I understand it, the appeal of TI is that it can stimulate deep structures without affecting everything else on top of them.

We think there may be a difference in community perspective between us and the reviewer: in the brain stimulation community working around TIS, indeed this claim is rare, however in the community working on peripheral nerve stimulation, this “depth penetration” claim is prevalent and repeated as a given in the introduction of papers. Therefore, we decided, and maintain, that this deserves to be described and confronted here. Here we provide a few quotes from recent papers to back up our point:

“...In comparison to traditional current stimulation (<100 Hz), the kHz frequencies encounter lower skin impedance and can therefore penetrate deeper into the tissues” [10.3390/biomedicines11071813]

“The fundamental aspects of interferential current involve reducing cutaneous nerve stimulation and maximizing the current that permeates the tissues [3] with a higher carrier frequency, making it more suitable for treating deeper tissue layers” [10.3390/medicina58010141]

“The underlying premise of IFC was the use of two medium frequency circuits (e.g. 4000 and 4100 Hz) creating a low frequency beat effect in the tissue thought to be capable of producing similar physiological effects to low frequency currents with less discomfort [2,3]. The use of two currents at around 4 kHz was

claimed to overcome skin impedance and permit current penetration into the deeper tissues [3,4].” [10.1016/j.physio.2011.01.008]

“IFC is a transcutaneous alternating medium-frequency (1 to 10 kHz) electric current that is amplitude modulated in low frequency. It has been claimed that IFC reduces skin impedance and deeply penetrates the tissues.” [10.1111/papr.12888]

“Notably, because of its high carrier frequency, interferential stimulation had been postulated to penetrate to deeper tissues than TENS.” [10.1093/ptj/pzz005]

Additionally, I think the claim is not that there’s something special about amplitude modulation per se, but that the AM occurs in a frequency range (~10 Hz) where it can have a larger biological effect. This could be true even if the AM is, in physical terms, much weaker. I believe this is the argument behind e.g., Esmailpour et al. (in vitro) and Viera et al. (in primates).

We fully agree that there are obvious, and often beneficial, biological effects that are synchronous to the low-frequency AMF, however our point is that the electrical stimulation is not coming from the AMF itself, as there is no power there, but from the carrier.

In summary, this paper makes interesting but very strong claims that I’m not sure are as generalizable as the authors claim.

We hope that we may be able to convince the reviewer with our revision, where we have revised the introduction and discussion, and provide more evidence to back up our claim, the central one being that unmodulated versus modulated kHz stimulate via the same mechanism, and that measuring s-f dependence provides strong evidence to that effect. We agree that TIS can be highly useful, however the effects of tonic off-target stimulation cannot be ignored. We concede, and now discuss in the paper, that there can be other effects at the subthreshold level to which our findings are not generalizable, however we try to point out the evidence that the phasic/tonic problem will not be different when one considers subthreshold. Overall we have now pointed out limitations of our study in the final discussion section, inspired by feedback from this reviewer. We do not think the debate on TIS is resolved, there is plenty more to study.

Reviewer #3 (Remarks to the Author):

This manuscript presents a quite comprehensive study of the effects of temporal interference stimulation (TIS) on peripheral nerves, measured in invertebrate and in human subjects. The study was well conducted, and the results represent a good contribution to the understanding of nerve stimulation using kilohertz-frequency signals. More specifically, the authors compared 4-electrode temporal interference against 2-electrode amplitude-modulated stimulation to demonstrate that the effects of the former are due to presence of kilohertz carrier rather than a purported interference on the target tissue, namely, the nerve fiber.

The authors did a quite good job in describing supporting evidence for their rationale, but in my opinion the introduction is unnecessarily long, it was written as a review. I value all the background material presented and thoroughly discussed here, but it seems it is too much for an Introduction. The interpretation and assumptions under TIS have been reviewed and discussed elsewhere (again, see for example Ward et al, 2009 cited by the authors), and similar arguments to challenge (some of) the assumptions have been presented in the past (again, Mirzakhaelili et al, 2020).

We have revised the introduction for clarity, especially with regards to characterizing the current assumptions and misunderstandings around TIS, and have shortened it by more than half of the original length.

In the revised version of the introduction, we reworked the text to highlight the aspects in our approach and thinking which have not been considered before, for example by the Ward and Mirzakhelili papers cited above:

-We do go into some detail on the fundamental works on biophysics of sinusoidal kHz waveforms for stimulation (starting from Katz 1939, up through 1960 papers by Bromm, 1970-1980s papers by Dalziel and Reilly), because these have been completely ignored by the older and newer literature on interferential currents and TIS.

- We have never seen the TIS literature make any reference to strength-frequency dependence (s-f curves), and only rarely to the Gildemeister effect, which are core concepts in understanding stimulation by sinusoidal stimuli. We have shortened our introduction by more than half, and wrote it in a much less “review” style, trying to stick only to what is directly pertinent to our study. In the revision we have endeavored to shave down the text while still putting attention on the concepts behind kHz stimulation, s-f curves, and earlier literature in an original and useful way.

We have cited Mirzakhelili et al, 2020 extensively and we believe we have given due credit to this important work and expressly stated that our motivation was to provide experimental evidence for the claims of that theoretical paper, as the authors themselves provoke in their discussion section that their model should be verified in a peripheral nerve model which tests for direct neural activation.

In addition, the work may lack a bit of originality for two reasons:

1) The experimental measurements in human subjects presented here add up to the existing body of work, and the measurements in invertebrates seem quite novel indeed. However, comparing the effects on both motor responses and discomfort using interferential, Russian, and other modulated currents was performed in the past, as reviewed in a work cited by the authors (Ward et al, 2009).

The body of work of Alex Ward is fundamental in functional electrical stimulation using kHz frequencies, and he was an early critic of ICS. He published several studies reporting interferential currents and finds that there is minimal qualitative difference in sensory/motor/pain responses when comparing different types of modulation. However, Ward never reported stimulation threshold current comparisons, or threshold currents as a function of carrier frequency, which is the cornerstone of our current manuscript.

2) Theoretical work has discussed the biophysical mechanisms of temporal interference with fairly similar arguments (Mirzakhelili et al, 2020, also cited by this work). Written as it is, it would appear that many ideas are original. Again, the review-like approach of the Introduction may play a role here. I would recommend significantly shortening the Introduction and Discussion, acknowledging previous work in the argumentation, and moving faster to presenting the experimental results.

We have taken this advice to shorten the intro and make it more compact and readable, yet we still feel a review “from the basics” is beneficial and such a perspective on TIS does carry with it some originality. We should properly qualify the originality and novel contribution of our work with respect to these two key earlier papers (A. Ward, 2009; and Mirzakhelili et al, 2020)

- Mirzakhelili et al, 2020 presents a theoretical model for explaining TIS via the ion channel rectification effect which is responsible for rectifying of kHz frequency stimuli, and follows to raise

concerns that TIS will inevitably cause tonic stimulation in some areas and phasic in others. It postulates that there may be a difference in effective current threshold between tonic/phasic regions, where “preferred phasic frequencies” around 5 Hz may have lower effective threshold. The authors encourage follow up work in their conclusion, that these theoretical models should be tested in biological systems, especially something less equivocal like a peripheral nerve. Our work experimentally confirms these key postulates from Mirzakhilili et al, 2020, using two (and now three, in the revision) distinct nerve models. We agree that the model explaining kHz/TIS is 100% the work of Mirzakhilili et al, 2020, and we feel we never implied that we were presenting this concept as our own (we have taken care in the revision to uphold this). Our original contribution is an experimental confirmation of that model. Moreover, in our opinion, the Mirzakhilili paper is sadly misunderstood by many scientists working in the TIS field, motivating us to carry out relatively simple demonstration experiments to drive home the tonic/phasic carrier stimulation problem.

- A. Ward, 2009 is a critical review article, covering much of the author’s own fundamental work on kHz frequency stimulation for muscle activation and comparing the potential advantages and disadvantages of kHz stimulation waveforms versus more common pulsed current waveforms. In this article, Alex Ward takes a very critical stance to interferential current approaches, stating that the purported claims about depth of penetration and “low frequency stimulation at depth”, etc, do not make sense. We agree with Prof. Ward’s critiques here, however in this highly influential body of work on kHz stimuli in physiotherapy and rehabilitation, there is no current threshold comparison, a lack which has been pointed out by more recent literature which reexamines some of his original comparisons between TENS and kHz stimulation (Luu, B. et al. (2024). Pain tolerance and the thresholds of human sensory and motor axons to single and repetitive bursts of kilohertz-frequency stimulation. *Journal of Physiology*, <https://doi.org/10.1113/JP286976>).

Now, a major issue for me is the theoretical component of the study. Indeed, I'm having trouble to understand at all the contribution of the modeling work. For starters, the geometry not only looks too simplistic for a FEM approach but also seems to lack the details needed to establish the differences in activation regions due to different stimulation modalities. For instance, Fig 4 shows that a rather large electrode imbalance of 4x amplitude would result in a small displacement of the region of tonic activity. More importantly, the second derivative of the potential does not account for the nonlinear dynamics of nerve fiber needed to determine action potential generation, so the SMI defined here may be quite off. In addition, the model was not validated with experimental data, so it lacks predictive value. Instead, it was used to reinforce and illustrate the idea of modulated vs unmodulated regions, an idea that was, nonetheless, already demonstrated using a much more comprehensive model that included nonlinear HH-based myelinated fibers (Mirzakhilili et al, 2020).

We agree that the paper would benefit from a reworking of the modelling work, and we have completely revised and expanded it in the R1. We take the point of too simplistic modeling geometry and implemented a more realistic 3-layer model that considers skin, fat and muscle tissue and the electrode-skin interface impedance. From the inputs we got it was also clear that the SMI metric used here was not intuitive enough and readers had trouble understanding its purpose. We have therefore switched to a similar metric that gives the same predictions, but is hopefully more intuitive, time over threshold (TOT). It represents the fraction of each modulation period that the target is exposed to over an threshold stimulus. The predictions for tonic, phasic and unstimulated remain completely the same as with SMI model, but we hope it’s easier to intuitively grasp.

The modeling work done in this study is predicated on our experimental finding that the stimulation thresholds do not meaningfully differ for modulated and unmodulated stimulation. This was the key insight that enabled us to define the SMI the way we did. The main goal of the modeling work was to explore the implications of the threshold equivalencies and to use that fact to visualize the spatial distribution of the tonic, phasic and unstimulated regions for the electrode configurations used in the study. The model predictions qualitatively agree with our observations with the phasic region being located in the midpoints between the electrode pairs and tonic regions directly below the electrodes.

We feel that the second derivative of the potential in the nerve direction (activation function) that we use in SMI is the relevant metric, as although it does not consider the nonlinear dynamics itself, it is in fact the driving force for the nonlinear models as can be seen in this quote from the *Applied Bioelectricity* textbook by Riley [DOI 10.1007/978-1-4612-1664-3] or in the original works by Rattay.

“One conclusion that can be drawn is that a second spatial derivative of voltage (or equivalently a first derivative of the electric field) must exist along the long axis of an excitable fiber in order to support excitation. The second spatial derivative of the external voltage has been included in an "activation function" in order to emphasize this essential aspect of stimulation (Rattay, 1986, 1989; Plonsey and Barr, 1995).” - *Applied Bioelectricity*, Riley [DOI 10.1007/978-1-4612-1664-3], p. 119

We would argue that, although the magnitude of the activation function itself would not be sufficient to determine with certainty if the nerve will be stimulated or not without introducing a model of ion channel dynamics, if the stimulation does occur, we expect it to happen precisely at the locations predicted by the maxima of the activation function, as it serves as the driving mechanism of excitation.

We agree that the work by Mirzakhali et al is illuminating, however in their work they used a simplified potential distribution derived from analytic expression for a point current source, and they explored moving the nerve only in 1 dimension. This is understandable, as for every nerve position one must run a separate NEURON simulation which could become prohibitive for exploring stimulation in 2D space or more realistic electrode configurations. Our experimental finding that the stimulation threshold doesn't meaningfully differ for modulated and unmodulated regions allowed us to define SMI to evaluate the tonic, phasic and unstimulated regions in a much simpler way that can easily be adopted by someone familiar with finite element modeling for any geometry or electrode configuration. We believe the simplicity of the SMI defined here based on experimental insight to be the advantage of our model as it helps minimize the confusion caused by having to rely on overparametrized HH nerve models as well as enabling rapid visualization of any electrode configuration on a 3D geometry.

We emphasized in the revised manuscript text that the entire premise of our model is based on our experimental finding that the stimulation thresholds for modulated and unmodulated stimulation do not meaningfully differ and that the purpose of the modelling work is explore and visualize the implications of that finding. We hope this clarification makes the purpose of the modeling part more useful to readers.

As well, a long paragraph in the Introduction is devoted to challenge assumption (3), but I think the argument is somewhat dubious. The power spectrum of a monophasic rectangular pulse of 0.2 ms is very different to that of a 5 kHz sine wave, regardless the former having a 5 kHz component. So it is highly imprecise to claim that these signals will encounter similar impedance in the skin.

We thank the reviewer for catching this sloppy mistake. The power spectrum of a monophasic rectangular pulse indeed contains significant low-frequency content. We changed the example pulse from monophasic

rectangular to biphasic rectangular (with 0.1 ms per phase) which should result in much better comparison between the pulse and the 5 kHz sine. We attach a plot of spectral power density and cumulative power for a biphasic rectangular pulse with 0.1 ms phase duration repeated every 1s.

The frequency components in the shaded grey area spanning from 2.4 kHz to 5 kHz contain more than 50% of the total pulse power, while the median frequency of the power distribution is 4.2 kHz. Although the comparison made is still not exact, we do believe it to be a useful one since we have noticed some researchers in the medical fields confusing the phase duration with the repetition frequency. With our analogy we show that for short pulses that repeated on a much larger timescale the meaningful value as far as the impedance is concerned is the phase duration, and not the repetition period.

Finally, the Discussion repeats many of the ideas developed in the Introduction, with little-to-none mention to possible limitations of the findings. Further, it caught my attention that very little is discussed about the differences and/or similarities observed between the measurements in myelinated (human experiment) and unmyelinated (locust experiment) fibers. Again, a more complete modeling approach could have supported hypothesis related to the differences in the ion channel dynamics of both type of fibers. In sum, a more concise manuscript emphasizing the contribution of the experimental measurements, accompanied by a more complete and detailed model suitable for studying nerve fiber dynamics, may be needed for further consideration.

We take the reviewers point that the conclusion/discussion was ineffective. We have rewritten it and added a section for limitations. We do not see any significant differences between the insect and mammal nerve stimulation situations for modulated/unmodulated sine, and retract our previous unfounded claim about there possibly being some important distinction explaining the lack of AMF difference – this point was wrong.

Reviewer #4 (Remarks to the Author):

This manuscript describes a study exploring the potential mechanisms of neural activation during temporal interference stimulation (TIS). The study explores these mechanisms in the context of peripheral nerve stimulation (PNS) and utilizes multiple approaches, including a locust model, experiments in healthy human subjects, and computational modeling. The authors consider multiple conditions in their experiments, including simple kHz frequency stimulation, amplitude modulated kHz frequency stimulation, and TIS. The authors look at the recruitment profiles as a function of frequency (i.e., strength-frequency (s-f) curves) for these different waveform paradigms and found that these curves overlapped. Based on their results, the authors conclude that the mechanisms driving peripheral nerve excitation are the same between kHz frequency stimulation and TIS. These results suggest that, at least in the context of PNS, the possible practical advantages of TIS can more easily be achieved with amplitude-modulated kHz waveforms.

Since 2017, TIS has been pursued with renewed interest as a potential non-invasive approach to modulate activity in the central nervous system as well as in the PNS. However, studies have emerged questioning the ability for the TIS fields to effectively penetrate and reach deep neural structures. Furthermore, research has suggested that the amplitudes necessary to directly activate deep neural structures would be higher than safe amplitudes and would also produce side effects due to modulation of off-target structures. Therefore, further studies are necessary to characterize the neuromodulatory effects of TIS and to evaluate its ability to selectively target neural structures. Due to the continued interest in TIS technologies and their use in clinical trials, the current manuscript is timely and its results represent a significant contribution to the field of neurostimulation and TIS. However, there are several issues that should be addressed before the manuscript is considered acceptable for publication.

We appreciate this assessment, especially pointing out the timely nature of the contribution. This motivates us to revise the work to be as understandable as possible to a broader community of scientists, and to provide actionable advice in designing effective experiments.

In general, the manuscript is well written. However, throughout the manuscript, there are several grammatical errors and places where the wording and structure could be improved.

We have now gone over the revision together with two English native speaking scientists to improve grammar and optimize phrasing, nearly all the text has been now rewritten.

The authors should make it explicitly clear in the Introduction and/or Discussion sections that they are investigating the mechanisms of suprathreshold TIS. This distinction is important because many people in the field now assume that TIS works via subthreshold mechanisms of action.

[See Common answer #1] Agreed, the distinction between sub/suprathreshold is vital. We have litigated this distinction in more detail in the introduction and conclusions, and pointed out how the findings from suprathreshold experiments must extrapolate to the subthreshold level as well, and back this up with modelling in the revised figure 2C. The rectification mechanism that leads to suprathreshold activation will be present when subthreshold field strengths are present, and the distinction between phasic and tonic stimulation will be analogously present whether one is sub- or suprathreshold.

At the end of the Introduction, the authors conclude that “some other, parallel, mechanism of demodulation is not present.” However, if this was true, then why do they observe a U-shaped response as a function of modulation frequency in which there is a modulation frequency that provides the minimum threshold (e.g., 5 Hz in Figure 3c)?

This is a good point that deserves more attention. The U-shaped response, where there are “preferred” AM frequencies where the effective stimulation threshold is lower, is fully consistent with the ion channel rectification-mediated stimulation effect of kHz, and is predicted by the model. The reason the model predicts this is due to the resonance properties given by the RC time constant of the cell membrane. A result around 4-10 Hz AM frequency is consistent with where resonance is expected to occur (Hutcheon and Yarom, 2000).

Mirzakhali E, Barra B, Capogrosso M and Lempka S F 2020 Biophysics of Temporal Interference Stimulation Cell Syst. 11 557-572.e5 predict that the optimum beat frequency will cause this U-shaped dip in the activation threshold current up to around 20%. Our experimental results agree with this prediction, both in terms of absolute magnitude and the optimal AM frequency. Our experimental U-curve finding indeed provides further evidence, together with the s-f dependence, that TIS is explainable via the model for kHz stimulation, and does not provide evidence for another parallel mechanism.

The authors state that they utilized electrode orientations that would achieve the maximal modulation index aligned with the target nerve. How did they determine these orientations? Were the overall field orientations similar for both the TIS configuration and the configuration for AM-SW, constant SW, and SW burst?

In the case of all two-electrode stimulation waveforms, the electrode placement (human and locust) were not moved during a given experimental session, therefore even if the placement of the stimulation electrodes are less optimal, the results of s-f curves, for example, would be self-consistent with one another when the goal is comparing the waveforms. In terms of 4-electrode TIS, orientation and electrode position indeed can be a trickier variable. For this reason we tried to keep things simple by using a square criss-cross arrangement of electrodes (this is the most common arrangement in interferential stimulation used in physiotherapy and functional electrical stimulation) which will give 4 max AM hotspots at midlines between electrodes (white discs are current sources):

The authors could consider providing additional discussion regarding the location and quality of the paresthesias that were observed in the human subjects during the different types of stimulation. These results could have interesting implications for the neural response to the different types of surface stimulation.

We have now added more description of this, and we specify that research on quality of the paresthesia experienced by subjects is a major topic for our group now, and we consider this to be highly interesting. The comparison of burst versus sinusoidally-modulated kHz waveforms can lead to very different precepts based on synchrony, as has been suggested in an interesting new study we now cite:

Barra, B. *et al.* High-frequency amplitude-modulated sinusoidal stimulation induces desynchronized yet controllable neural firing. *bioRxiv* 2024.02.14.580219 (2024).

The authors need to be specify how many experiments were performed for each set of results and each figure.

We have now revised this in all cases, and increased the N-numbers in the reported experiments.

Figure 4

-Show dimensions in part d and provide units in the plot of AM

-Are the authors considering the activating function or amplitude modulation of the activating function?

-The authors used their experimental current threshold measurements to estimate the stimulation threshold value of the activating function in their computational model and obtained a threshold of 2000 V/m². There are potential issues with translating absolute experimental values to a computational model. How did this estimate of 2000V/m² compare to other studies? The authors could run simulations with an multicompartment axon model and look at the actual activating function values required for activation within their computational model and see if the values based on their experimental thresholds are close to the model predictions.

We redesigned the figure in question and the dimensions and units are now shown. Our experimental findings comparing the modulated and unmodulated thresholds show that the most important factor for

stimulation is the amplitude, no matter if stimulation is modulated or unmodulated. What modulation does is that it simply changes the stimulation amplitude periodically from over the threshold to under the threshold. So for stimulation, the amplitude of the activating function is what is important, but to understand if the stimulation will be tonic (always over the threshold), phasic (sometimes over the threshold and sometimes below the threshold) or unstimulated (always below the threshold), both the amplitude of the activating function and the modulation need to be considered, which is what we show in the old Figure 4d updated as Figure 5D.

The absolute value for the activation function threshold will certainly depend on factors such as nerve diameter and geometry with smaller nerves exhibiting higher thresholds than the larger diameter and would thus be difficult to precisely calculate even when using multicompartment axon models such as the one in updated Figure 2B and C. However, the main points we are making here do not really depend on the absolute value of the activation function threshold used. If we change the nerve depth in our forearm model in a reasonable physiological range from 0.5 cm to 2 cm, the activation function threshold required for stimulation (assuming our experimentally measured current threshold) changes one order of magnitude from 3000 V/m² for the shallowest nerve to 300 V/m² for the deepest nerve. The model predictions on the other hand stay largely the same, predicting large tonically stimulated regions under the electrodes, phasic region between them and unstimulated regions elsewhere. This makes us believe that the model predictions are robust and nicely outline the main issues with the TIS that is currently not widely recognized in the community.

Sentences 4-6, First paragraph, Page 16: The authors discuss the issue of specificity and conclude that it does not happen with TIS based on their results. However, it is not clear if the experimental models utilized in this study provide a means to test specificity in which they would also need to consider off target effects. The authors should provide discussion regarding this potential advantage of TIS and the limitation of their experimental models and the continued need to test for off-target effects in TIS.

Thank you for this point, we agree this was weak in the original manuscript. We now connect the modelling in figure 6 to a simple experimental demonstration of tonic/phasic stimulation in practice, in addition to a supplementary video, showing how moving one electrode by a few mm will result in a transition from phasic motor activation to a tetanic contraction.

In the Discussion section, the authors should describe some of the major limitations of their study.

Thanks for this prompt. We have now explained key limitations, the foremost of which is that our study does not definitively prove anything with regards to subthreshold stimulation in the brain.

Minor comments

Bottom of Page 1: The authors should specify what they mean by “addressed” in the statement “This means that only superficial targets can be addressed.”

We have refined this imprecise language to “effectively stimulated”

Top of Page 2: Instead of just specifying > 1 V/m, the authors should provide estimates of the range of field strengths required for direct neural activation and the field strengths that are estimated to be possible during TIS utilizing safe current amplitudes.

This is a bit of a deeper rabbit hole to go into in the introduction. We reference this fine review for considerations of sub and suprathreshold transcranial E-fields: 10.7554/eLife.18834 and do not want to litigate potential safety margins, this debate is very much underway in the literature now.

Page 2: Provide a relevant reference(s) for the sentence beginning with, “A suggested approach to overcome this basic issue is using high-frequency electric fields,...”

While there are many earlier ICS papers that “handwaving” claim using higher frequency carriers to overcome impedance, we think the most appropriate work to cite is one that critically assessing this idea Medina, L. E., & Grill, W. M. (2014). Volume conductor model of transcutaneous electrical stimulation with kilohertz signals. *Journal of Neural Engineering*, 11(6), 066012. <https://doi.org/10.1088/1741-2560/11/6/066012>

Second sentence, Second Paragraph, Page 2: The authors should explain what they mean by “follow.”

We have largely re-written this section of the intro, now we explain the “firing rate” argument.

Sixth sentence, second paragraph, Page 3: The authors should provide a better description of the s-f plots. Explain the accommodation region at low frequencies and the thermal dissipation effects that begin to overtake direct electrical stimulation at high frequencies.

We now explain this, and take a more rigorous approach to introducing s-f plots in general, in the new Figure 2.

Bottom of second paragraph, Page 3: The authors state that kHz frequency stimulation can lead to activation due to the summation of subthreshold depolarizations because at kHz frequencies successive half cycles arrive within the absolute refractory period of neurons. However, because the cell has not generated an action potential, is this response related to the absolute refractory period?

Thanks for catching this mistake, this is wrong as-written. We have now fixed this.

Figure 1

-The authors describe the characteristic s-f curve for sinusoidal stimulation with the threshold minimum around 50Hz. How dependent is this minimum on neuron type?

From earlier literature, the minimum is described as between 50-200 Hz, with relatively flat response over this region. We now plot the s-f plot according to the Reilly model for a myelinated neuron (new Figure 2a).

-The authors should consider providing labels for the dashed green and blue lines in Figure 1d.

We have reworked this figure for clarity.

Figure 2

-Why is the plot for the constant SW, ramped shown in part e not included in the plot for part d?

This is now fixed in the revision – now unmodulated sine is part of all s-f curve figures. In the original submission, we had compared the unmodulated sine in only some experimental conditions, only three or four frequencies, etc. In this revised version, we measure unmodulated sine at all frequencies and as part of essentially every experiment, as we realized how comparing thresholds of modulated and unmodulated kHz is really the most critical message to the TIS community, thus our manuscript and results now focus much more on that.

Figure 3

-Why did the authors utilize a carrier frequency of 2.5 and 3.0 kHz in the locust and human experiments, respectively?

There is no significance to this difference. Historically, these types of locust experiments were done with a 2500 Hz carrier, and we continued this throughout the revision for consistency. Considering the s-f curve, there is not much difference between 2500 and 3000 Hz.

-Figure 3c – it seems unnecessary to include the text stating the carrier frequency, AM, n, SEM, etc. within the plot. But these values and what the gray shaded region represents should be described in the figure legend.

We are following the stylistic guideline that the figures should be able to, when possible, stand alone when exported from the webpage version of the article, which is useful when someone for example copies it into a presentation. For this reason we use empty space within the graphic to add info.

-Figure 3a-b – the authors should describe the number of experiments included in this data

This figure has been revised with new experiments, higher N-numbers, now specified in the caption.

Top of Page 11

-The authors describe recordings of the low-frequency AM envelopes inside of the mesothorax. The authors could consider adding a figure showing these recordings.

The section in question has been heavily redacted in the revised version, with a new figure 4, and that particular part of the text has been removed. We have, however, added more details on the methods of measuring the stimulation artefacts, which are very useful in planning a TIS experiment.

Bottom of Page 13: The authors may want to consider providing a basic description of the model design here within the Results.

This has now been revised and modelling details added.

Second sentence, First paragraph, Page 16: provide the relevant reference(s)

Fully agreed. The statement in question is: “In many recent papers describing the use of this method, it is assumed at the outset that the kHz frequencies used as carriers are not able to elicit stimulation on their own, at least not at the amplitudes being used in the experiments.” We had referenced the relevant papers in the introductory discussion, however we take the point that this belongs here as well. We have now pointed out several notable examples where the papers clearly articulate that they assume that the kHz carrier does not stimulate.

Article reference: NCOMMS-24-55626

Itemized responses to the referee comment are found below, with original comments in red, and our response in blue. All text changes are marked in the manuscript with yellow highlights.

Reviewer #1 (Remarks to the Author):

The authors have extensively revised their original manuscript and have properly addressed majority of my concerns and comments. Most importantly, they have conducted new experiments to investigate some of the questions raised by me and/or other reviewers and have perform statistical analysis to investigate if their findings are statistically significant or not. Understandably, a few of my questions were not directly answered as they seem to be questions that the authors have decided to investigate for future studies, or they were already part of on-going investigations. I do look forward to learning more about them in their future publications.

We are grateful for the positive assessment of our revised work and also appreciate the understanding about our ongoing studies.

Considering the current manuscript, I have only a few minor questions:

1- The authors have stated that “unmodulated sine has, on average across frequencies, lower threshold than modulated waveforms” for the results shown in Figure 3B (locust motor response). Next, for the human motor response (Figure 3F), the authors have stated that “TIS has the average lowest threshold, however we would indicate that this is due to a combination of its modulated nature and the 2x applied current, making it the clearest perceptual”. Therefore, I am not sure why the argument the authors are making for lower threshold for TIS in humans (both sensory and motor) does not apply for the locust model as they don't see lower thresholds for TIS in their animal model. The authors should provide some explanation why they see a different order for what stimulation has lower threshold for the animal model vs. human subjects.

We think that the point the reviewer is making here is a very important one, so we chose to cover the topic fully in the appendix of the manuscript.

Directly comparing 4-electrode TIS threshold to the thresholds obtained with 2-electrode configurations is generally not possible since the different configurations will not produce the same stimulus amplitude (activating function) at the nerve location for the same stimulation current. We have measured the 4-electrode TIS to make sure we observe the same general trend in the s-f relationship, but the direct threshold comparisons should only be made between the unmodulated and modulated sine since we measure them in the same configuration, and they can thus be directly compared.

The effect of the configuration on the current threshold is a complicated issue which needs to be modeled with proper finite element techniques and realistic tissue and contact impedances. We will try to demonstrate the concept, however, using a simplified model that can be solved analytically to try to build intuition and to get a sense of the important parameters.

For this purpose, we will assume a semi-infinite tissue of uniform electrical conductivity σ with point current sources for electrodes. We will compare the two configurations and see in which cases each configuration produces the larger stimulus at the target location. The two configurations in this simplified

geometry are depicted in the figure below with the nerve depicted in orange propagating perpendicular to the depicted plane.

Since the tissue is isotropic for the bottom hemisphere, the symmetry results in isotropic current density of magnitude:

$$J(r) = \frac{I}{2\pi r^2}$$

Where r is the distance from the current source. Ohms law $J = \sigma E$ allows us to find the expression for the electric field and electric potential.

$$E = \frac{J}{\sigma} = \frac{I}{2\pi\sigma r^2} = -\frac{dV(r)}{dr}$$

$$V = \frac{I}{2\pi\sigma r}$$

Finally, we can find the expression for activating function in the nerve direction (labeled x direction for consistency with the manuscript) evaluated on the depicted plane:

$$A_x = \frac{d^2V}{dx^2} \Big|_{x=0} = -\frac{I}{2\pi\sigma r^3}$$

If we want to have the same activating functions at the nerve location in both configurations for the same stimulation current I , we must have:

$$\frac{I}{2\pi\sigma d^3} = 2 \cdot \frac{I}{2\pi\sigma(l^2 + d^2)^{3/2}}$$

Which gives the solution:

$$l = \sqrt{2^{2/3} - 1} d \approx 0.77 d$$

Only in the special geometry where this condition is satisfied will both configurations give the same activating function at the nerve location. If the electrodes are farther apart (or equivalently, if the nerve is shallower), the additional distance from the nerve in the 4-electrode configuration will not be fully compensated by 2x the injected current (one I through each electrode pair) and the 2-electrode configuration will have apparently lower stimulation threshold. If the electrodes are closer (or the nerve deeper) the 4-electrode configuration will have the advantage and it will have the lower threshold.

In practice, the situation is complicated by different tissue type (skin, fat, muscle) conductivities and electrodes of finite dimensions so the l to d ratio that gives the same stimulus at the nerve location will most likely differ from the value calculated in the simplified geometry. Moreover, since the locusts and humans have different tissue types and distributions, this ratio will also differ between the two models. **In general, if the electrodes are closer (relative to the nerve depth), the 4-electrode configuration will have an advantage due to twice the injected current, and if the electrodes are farther apart, the 2-electrode configurations will have an advantage due to smaller distance to the nerve.**

The important point here is that we should not generally expect the thresholds to be equal for the two configurations and if it turns out to be the case, it will be a result of picking a specific geometry, and not a general rule. In our case, the chosen geometries happen to favor the TIS configuration in the case of the human model and 2 electrode configuration in the case of the locust model. By moving the electrodes closer in the locust and further apart in the human experiments we would expect to achieve the opposite result. The threshold comparison between unmodulated sine and AM sine does not have the same issue since they use the same electrode configuration.

2- Considering the results for the animal model and human subjects in Figure 3B and D, the s-f curves, the relation between the current threshold and the carrier frequency seems to be nonlinear for motor response in the lower carrier frequencies but the response seems to be look linear for the sensory response (Figure 3F). It would be helpful if the authors could provide some explanation for this behavior.

Neither of these dependences is truly linear. The high frequency behavior has the asymptotic form (Reilly equation):

$$I_t \approx I_0 \left(\frac{f}{f_e} \right)^a$$

It appears linear when viewed on the log-log plot, but the coefficient a obtained by fitting to the Reilly equation is (0.84 ± 0.03) for sensory data and (0.83 ± 0.06) for motor data which differs significantly from unity (i.e. linear dependence) in both cases. The low frequency non-linearity apparent in the motor data even on the log-log plot is due to the value of the f_e constant governing high frequency behavior of the S-F relation. The full high frequency behavior predicted by the Reilly equation is given by expression:

$$I_t = I_0 \left(1 - e^{-\frac{f_e}{f}} \right)^{-a}$$

Which simplifies to the asymptotic form above only when f is sufficiently larger than f_e to approximate $e^{-\frac{f_e}{f}} \approx 1 - \frac{f_e}{f}$ by its first two terms in the Taylor expansion. For frequencies for which this requirement is not satisfied we will observe nonlinear behavior visible also in Figure 2a for the intermediate frequencies.

For sensory measurements the constant f_e has a value $f_e = (400 \pm 100)$ Hz which is smaller than our smallest measured frequency, while for motor measurements the constant is $f_e = (1500 \pm 300)$ Hz which makes the high frequency approximation invalid at lowest measured frequencies. If we were to measure even lower frequencies we would expect to observe the same nonlinear dependence also in the sensory nerves (it is slightly visible at the lowest frequencies even with our current measurement range.)

Experimental values for f_e are known to differ significantly for different nerve fiber diameters and types (*Applied Bioelectricity*, Riley [DOI 10.1007/978-1-4612-1664-3]: Chapters 6 and 7).

3- Figure 5 suggests that for some amplitude modulation frequencies, the motor threshold and sensory thresholds are closer to each other compared to the others (is the data for 2 Hz for sensory missing?). The authors should explain why such difference exists and if it having these thresholds close to each other beneficial or not.

Although it is true that the two thresholds are closer for some modulation frequencies than others, we are not sure that it is practically as consequential as it seems at first glance since one has to consider that the values in Figure 5 are normalized to the minimal threshold, and that minimal threshold is not the same for motor and sensory fibers. We have not measured 2 Hz data point for the sensory model, unfortunately. As for the difference between the two models, it could result from multiple factors that make the two models different such as different nerve fiber types, diameters and myelination percentages. We are not sure which of the factors is the main contributor.

Reviewer #2 (Remarks to the Author):

Thank you to the authors and editors for considering my comments on Opančar and colleagues' manuscript "There is no biophysical distinction between TIS and Direct kHz stimulation of peripheral nerves" (NCOMMS-24-55626). The revised manuscript contains much more detailed descriptions of the experimental procedures and formal statistical analyses of the results. These are essential for scientific publications. On the other hand, these new analyses seem to undercut the paper's main claims.

Our main claim is that experimental evidence supports the conclusion that TIS is driven by the kHz carrier itself, and as we argue below, we do not think that the new analyses undercut the main claims for the paper.

The authors' main argument is that if strength-frequency curves are equivalent across the stimulation modalities, those modalities must act via similar mechanisms. The curves in Figures 3 and 4 certainly do look alike, but the new analysis suggests a significant effect of waveform in two conditions** and a marginal one in the other ($p=0.07$, but with a $N=10$ is hard to interpret). The authors do argue that this difference may be driven by their readout/task and refer to the qualitative similarities between the curves instead. That may be, but I'm uncomfortable recommending a paper where all of the formal analysis is ignored. I'm also not convinced that failing to find a significant difference would be the right way to demonstrate this; an equivalence test would be more appropriate.

Our main claim is that the frequency dependence of TIS is consistent with the kHz stimulation mechanism, because sinusoidal kHz stimulation has a characteristic s-f relation according to the Reilly equation, and to the theoretical model of Mizrakhilili. The TIS data and amplitude-modulated sine data all fit this expected s-f relation.

We do not claim that the s-f curves are equivalent across all waveforms in terms of absolute values of current, nor do we expect they should be equivalent across waveforms. This being said, the reviewer's comment is on-point and well-taken that we must be very careful in our language and the words we choose to be precise. We fix the one instance where we use the word "equivalent" inaccurately: "The average threshold ratios for modulated/unmodulated sine waveforms ~~are equivalent~~ across the frequency range are: AM/sine = 0.99 ± 0.02 ; Burst/sine = 0.97 ± 0.02 ." In the paper we stated that "TIS *has the same strength-frequency dependence* as 2-electrode kHz stimulation waveforms", which we would maintain conveys the message correctly, but in a few instances we wrote the "...has the same *curve*.." this is wrong (as the reviewer points out, it would suggest we are implying that the curves are equivalent in terms of absolute values) and we have corrected it – the curves themselves are not exactly the same, there are in some cases statistically-significant differences between them, but the s-f behavior is the same across all waveforms, they all follow the Reilly equation. If TIS is driven by the kHz stimulation mechanism, one would expect the same type of s-f dependence, but one should not necessarily expect the slope of that dependence to be identical or for absolute values of the current thresholds should have some variability between each other. We have already explained why, due to different geometry and injected currents, 4-electrode arrangements will usually not lead to the same absolute current threshold values as 2-electrode arrangements (Newly detailed in Appendix A).

Our evidence for maintaining that TIS is driven by kHz stimulation is based on not only the characteristic s-f curves, but also the AMF dependence which is consistent with the Mizrakhilili model. We have now gone through the paper to tighten the language concerning our core conclusion that the consistency of finding a strength-frequency dependence across waveforms implies a shared activation mechanism. All these findings taken together support, rather than undermine, the core conclusion that suprathreshold stimulation observed under application of TIS is consistent with the kHz stimulation mechanism.

Regarding the comment about the equivalence testing we agree with the reviewer, but we would argue that it is already present in the manuscript in the form of threshold ratios with their respective uncertainties. For example, in the human experiments, we have the following for the motor nerves:

"The average threshold ratios for modulated/unmodulated sine waveforms across the frequency range: AM/sine = 0.99 ± 0.02 ; Burst/sine = 0.97 ± 0.02 ."

And the sensory nerves:

"These differences equate to ratios of sensory thresholds which favor the modulated waveforms: AM/sine = 0.93 ± 0.02 ; Burst/sine = 0.84 ± 0.02 ."

These ratios are written in the standard form (Mean \pm Standard error of the mean), so the 95% confidence interval can be calculated as [Mean – 1.96 SEM, Mean + 1.96 SEM]. For the motor nerves that results in 95% CI for AM/sine = [0.95, 1.03] which means that for motor case modulated and unmodulated thresholds do not differ by more than 5% (with $p=0.05$). For the sensory nerves on the other hand the 95% CI turns out as AM/sine = [0.89, 0.97] which means the thresholds do not differ by more than 11% (with $p=0.05$) and that the threshold ratio differs significantly from unity as we found with significance testing.

We did not explicitly list the confidence intervals for the different threshold ratios in the manuscript as we feel it does not contribute to getting our main point across, so although there may be some variability in thresholds between waveforms, the overall s–f behavior is consistent with a single mechanism. As to how practically consequential the magnitude of the differences between modulated/unmodulated thresholds and differences between s–f curves are, this is for the readership to judge for themselves in the context of their own experimental interests, and this we tried to convey in the discussion/conclusions part of the manuscript and the final Figure 7.

The Time-Over-Threshold analysis is interesting but it would be nice if it were tested or analyzed a little more rigorously, perhaps by plotting electrode position or current ratio as a function of finger position.

We agree that conceptually the displacement caused by muscle movement, such as displacement of the finger, correlates to time-over-threshold, and we think this is illustrated very well in the example of the locust leg, which provides an easy to quantify and highly reproducible displacement (relative to measuring the human finger). We realize that in the text we have presented the locust and human modelling rather separately, but they are indeed conceptually connected. Therefore, to address this comment, we have in the text pointed out the ToT measurement in the locust leg as a perfect illustration of this concept.

Additionally, the manuscript still contains a very long and assertive review of the literature on kHz stimulation. As I and the other reviewers noted, there is useful material in here, but much of it is already available elsewhere (e.g., the Mirzakhilili et al. model is discussed at great length, including much of the material in Figure 2 and 7) and it's unusually long (8 pages). There is useful material in here, but the empirical portion probably won't resonate with the audience that most needs to read the review: I am skeptical that most human NIBS users are familiar with strength-duration curves, for example. In the cover letter, the authors describe this as an empirical test of Mirzakhilili et al.; this would be a much more succinct framing.

We appreciate the reviewer's continued engagement and thoughtful critique. In our first revision, we recognized the initial introduction was overly lengthy and reiterative of prior work, particularly that of Mirzakhilili et al. and Ward et al. To address this, we undertook a substantial revision, reducing the introduction's length by more than half and focusing on streamlining the narrative. We limited background discussion to what we believe is strictly necessary to contextualize our empirical contributions.

In response to the current comment, we have now gone a step further and made additional efforts to further condense the introduction. We have focused it more tightly on the novel contributions of our study, specifically the empirical demonstration of carrier frequency dependence and the strength-frequency framework in the context of temporal interference stimulation. While some literature review remains, we believe it is essential to justify and anchor our framing. Carrier frequency dependence remains underexplored in the TIS literature and is rarely addressed explicitly in either modeling or experimental studies of temporal interference. We aim to foreground this concept as a fundamental parameter deserving closer attention by the community, for instance when choosing carrier frequency in NIBS studies, experimenters can consider that higher frequencies may lead to a weaker stimulation effect.

The concept of strength-frequency dependence and nerve accommodation to different frequencies has historical precedent, dating back to Reilly's work in the 1980s–1990s and earlier studies by A.V. Hill, Gildemeister, and Bromm & Lullies in the 1940s and 1960s, yet it is absent in the contemporary temporal

interference discourse. These studies have not been cited in prior work in the TIS/ICS field, and we believe our effort to revive and apply this framework is both timely and original. While we acknowledge that parts of the introduction may read as "review-like," we maintain that they are not redundant restatements of widely known facts but rather necessary theoretical groundwork to interpret and frame our findings. Moreover, our target readership extends beyond the noninvasive brain stimulation (NIBS) community. As an author team engaged in peripheral nerve stimulation research, we would like to emphasize that kHz-modulated and interferential stimulation is already widely implemented in clinical practice for peripheral applications, particularly in rehabilitation settings across the UK, Germany, Czech Republic, Austria, and other regions. Thus, the manuscript is also directed toward researchers and clinicians in this community, for whom our framework and findings are directly relevant.

To better reflect this dual audience, we have revised both the introduction and conclusion to more clearly communicate the relevance of our work to both brain and peripheral stimulation domains. We hope the reviewer finds these revisions responsive and effective in enhancing the focus and accessibility of the manuscript.

I do agree with the authors that there has been a lot of hype/misconceptions about TIS and we should better understand if/how it works before claiming it is a panacea. However, this work needs to be done and communicated carefully i.

We fully agree with the reviewer that careful and rigorous communication is essential, especially in a field where misconceptions and premature claims have gained traction. Our intention throughout has been to contribute constructively to the discourse on temporal interference stimulation by critically examining its underlying mechanisms, particularly with regard to carrier frequency dependence. We have revised the manuscript to further clarify our framing, moderate the tone where needed, and ensure that our conclusions are appropriately cautious and grounded in the presented evidence.

Misc:

** $p < 0.07$ is alternately described as significant (p. 13) and not significant (p. 12). Either way, I would be reluctant to interpret this as strong evidence against a difference, given the sample size.

We thank the reviewer for their watchful eye. This was in fact a typo caused by copying text from the motor experiment to the sensory experiment, in fact the waveform effect is significant in the case of the sensory experiment with $p < 0.0001$, the $p = 0.07$ refers only to the motor experiment. We now correct the text and have carefully double-checked all reporting of statistics.

The title is confusing. I understand that the authors are arguing that the same biophysical mechanism underlies TIS and "direct" kHz stimulation, but the outcomes are (per their own data), quite different. In addition to the tonic modulation, TIS produces phasic modulation at the foci. Thus, a title like "The same biophysical mechanism is involved in..." might better reflect the claims.

We are grateful for this comment, this is a very good point and a useful suggestion on how to reconfigure the title to be more fitting and indeed clear up any ambiguity as discussed above about our findings and claims. We have changed it to "**The same biophysical mechanism is involved in both temporal interference and direct kHz stimulation of peripheral nerves**"

Figure 2: It'd be helpful to put the Reilly equation in the text itself, so it's searchable.

Done.

Page 12: Did the authors run an entirely new set of experiments or analyses? The finger used as a biomarker seems to have switched from middle to ring?

Yes, all these experiments were redone using a refined set of methods, with software-generated randomization of waveform presentation and automatic amplitude ramping. None of the data from the original submission was used or pooled with the new data. As to the matter of the middle versus ring finger: there are differences in tendon structure between individuals. In the majority of participants, FDS contraction moves primarily the ring finger, and in some minority, it is primarily the middle finger. We realized it was rather unusual to have an example photo showing the rarer form of the response therefore we updated the photos to show the much more common ring finger movement.

Page 13: Subjective reports. It's not clear to me what this adds to the mechanistic story that forms the main thrust of of the paper.

The issue of comfort and tolerability is important in the field of functional electrical stimulation, and there is an ongoing debate about the use of different kHz waveforms and the subjective comfort, for instance very recent papers: 10.1088/1741-2552/adc33b and 10.1113/JP286976 and this is a topic of our ongoing research (as reviewer 1 pointed out above as well) and therefore we think that mentioning our findings here is appropriate, even though it does not pertain to the main message of the manuscript concerning mechanism considerations.

Page 22: “ Based on the potential of the unmodulated sine carrier to stimulate neural tissue, it is arguable if this can be considered a “sham” condition versus the experimental condition of interfering currents and AM.”

This strikes me as exactly backwards. Most uses of TIS (in the CNS) test a hypothesis regarding the role of neural activity at a stimulation focus and the AMF frequency. In this sense, a “sham” condition using the one of the carriers accounts for any off-target effects. A no-stimulation condition would be useful for interpreting null results (TI == high frequency sham), but a main effect of TIS seems interpretable to me.

We believe we are on the same page with the reviewer in terms of the experimental design described, and agree our statement needs clarification. As we understand it, sham stimulation would refer to a condition where the subject is not actually receiving any electrical stimulation at all, but is only experiencing a procedural mimic of an actual stimulation condition. In this respect, application of unmodulated carrier alone is not a sham, because it is clear that the carrier alone can lead to stimulation of neural tissue. We agree, that a proper experimental design should compare, 0) true negative control, subject without any semblance of electrical stimulation, 1) no applied electrical stimulation at all, but the subject is led to believe they are undergoing stimulation, 2) unmodulated carrier only, and 3) TIS which is applying AMF at some focus. However, we think that semantically, the “sham” condition here is (1), and not (2). Calling condition (2) a sham presupposes that one is expecting no possibility of impact on neural behavior, and this is not correct.

We have reworded the statement to make it clearer and added extra clarification in the revised manuscript.

Page 23: “Moreover, in many cases, we would argue that a premodulated kHz stimulation arrangement may provide some key benefits of TIS.”

Could the authors elaborate on the “key benefits”? Given the argument that TIS is equivalent to direct stimulation, why not just apply a sine wave at the AM frequency?

We agree with this conclusion, that one should in most cases choose to apply a 2-electrode AM sine instead of bothering with a 4-electrode TIS arrangement, and we tried to get this point across in the conclusion and also in the final Figure 7. This was actually more emphasized in the first version of the paper and was somehow downplayed in the revision. Now we take this comment as a prompt to reemphasize this and to carefully detail what the benefits of a 4-electrode arrangement can be. We would argue that in most practical cases, especially for noninvasive peripheral nerve stimulation (which is the primary area of work for the author team), an AM sine is better and more practical than 4-electrode TIS. However, there are two cases where 4-electrode TIS provides a possible advantage: 1) A case where one wants to target superficial areas of a tissue with tonic stimulation, and achieve phasic stimulation somewhere at-depth. This kind of core/shell stimulation paradigm can be interesting for cuff electrodes implanted around a peripheral nerve, and has been suggested in recent patent and scientific literature. 2) A case where current steering of a phasic stimulation hotspot is useful. By changing the current ratios between pairs, one can move the position of the hotspot. This has been explored by many authors, and in our team is used to adjust the aim of the phasic hotspot on peripheral nerve targets which tend to move around a lot with subject movement, *e.g.* the peroneal nerve behind the knee. We can steer the hotspot instead of adjusting electrode position.

Reviewer #3 (Remarks to the Author):

The manuscript was greatly improved, and the text in general flows much smoother now with a bit shorter Introduction and revised discussion. Nearly all Figures were revised, and I think they better convey the main message of the manuscript now.

I still have one concern about the modeling work. I'm not convinced that the TOT is a more intuitive metric. For starters, since a frequency domain approach was used, A_1 and A_2 do not depend on time, and therefore it is unclear how TOT can be defined as "the fraction of each modulation period that the nerve is suprathreshold". There is no time involved at all. After looking Figure 6E, I thought that TOT was somehow calculated using $A_1(t)$ and $A_1(t)$, but nothing in the equations and definitions suggests this. Perhaps simply changing the name of TOT can do the trick in terms of the metric, but certainly a clearer explanation of what is illustrated in Figure 6 for frequency versus time domain is needed.

We agree our discussion on TOT needs refinement, in particular we note that we have defined TOT as a “fraction of each modulation period” and not an actual time value, and have gone through the text to add “fraction” where missing. We note, however, that when using a modulation period of 1 s which is what we were using for human experiments the fraction value and the time value have the same numerical value, and the x-axis of in units of seconds used in Figure 6E is correct. To resolve any misunderstanding we have added a new Appendix B which describes the derivation of TOT.

As for the question of time vs. frequency domain, for sinusoidal stimulation it is possible to transform the values between frequency domain and time domain which is what we have done to derive the expression for TOT. The electric potential and the activating function in x direction $A = d^2V/dx^2$ are both phasor quantities that are fully characterized by their amplitude and phase (and frequency which is known). The amplitudes A_1, A_2 and phases φ_1, φ_2 are indeed time independent, but it is easy to write down the expressions for time dependent activation functions:

$$A_1(t) = A_1 \cdot \cos(\omega_1 t + \varphi_1)$$

$$A_2(t) = A_2 \cdot \cos(\omega_2 t + \varphi_2)$$

That gives the total time dependent activation function as:

$$A(t) = A_1(t) + A_2(t) = A_1 \cdot \cos(\omega_1 t + \varphi_1) + A_2 \cdot \cos(\omega_2 t + \varphi_2)$$

For the purposes of calculating TOT, we are interested in the envelope of the resulting activating function $A(t)$ which we will term $E(t)$. We can find the expression for $E(t)$ from simple geometric considerations by representing the quantities $A_1(t)$ and $A_2(t)$ as rotating vectors in the complex plane.

We can use the law of cosines to obtain the time dependent envelope $E(t)$ while using the identity $\cos(180 - \theta) = -\cos \theta$.

$$E(t) = \sqrt{A_1^2 + A_2^2 + 2A_1A_2 \cos \theta}$$

The angle θ will equal the phase difference between $A_2(t)$ and $A_1(t)$:

$$\theta = \omega_2 t + \varphi_2 - (\omega_1 t + \varphi_1) = (\omega_2 - \omega_1)t + \Delta\varphi$$

Where $\Delta\varphi \equiv \varphi_2 - \varphi_1$

The time dependent expression for $E(t)$ then becomes:

$$E(t) = \sqrt{A_1^2 + A_2^2 + 2A_1A_2 \cos((\omega_2 - \omega_1)t + \Delta\varphi)}$$

Time over threshold can then be found by calculating the time difference between two consecutive solutions of the equation $E(t) = A_T$:

$$\sqrt{A_1^2 + A_2^2 + 2A_1A_2 \cos((\omega_2 - \omega_1)t + \Delta\varphi)} = A_T$$

Which after squaring both sides and rearranging gives:

$$\cos((\omega_2 - \omega_1)t + \Delta\varphi) = \frac{A_T^2 - A_1^2 - A_2^2}{2A_1A_2}$$

Solving this equation gives:

$$(\omega_2 - \omega_1)t + \Delta\varphi = \pm \arccos\left(\frac{A_T^2 - A_1^2 - A_2^2}{2A_1A_2}\right) + 2k\pi \quad k \in \mathbb{Z}$$

And the time difference between the two consecutive solutions for t is then:

$$\Delta t = \frac{2}{\omega_2 - \omega_1} \arccos\left(\frac{A_T^2 - A_1^2 - A_2^2}{2A_1A_2}\right) \quad \text{if} \quad |A_1 - A_2| \leq A_T \leq A_1 + A_2$$

The definition for the TOT that we give in the manuscript is “the fraction of each modulation period that the nerve is suprathreshold” so to obtain the expression for TOT we need to divide Δt by the modulation period which is $T_{MOD} = \frac{2\pi}{\omega_2 - \omega_1}$ which gives the solution for TOT that is written in the manuscript:

$$TOT = \begin{cases} 0 & \text{if} \quad A_1 + A_2 < A_T \\ \frac{1}{\pi} \arccos\left(\frac{A_T^2 - A_1^2 - A_2^2}{2A_1A_2}\right) & \text{if} \quad |A_1 - A_2| \leq A_T \leq A_1 + A_2 \\ 1 & \text{if} \quad |A_1 - A_2| > A_T \end{cases} \quad (1)$$

We have added this derivation to the manuscript appendix (new Appendix B) to make the mathematical background of equation (1) more transparent.

Reviewer #4 (Remarks to the Author):

Based on the reviewer comments, the authors have made significant efforts to improve their manuscript, including rewriting and reformatting text and figures, performing additional locust and human subjects experiments, and providing additional details regarding their methods and results. These efforts are much appreciated and have significantly improved the quality of the manuscript. This manuscript represents an important contribution to the field and I commend the authors for their efforts. While I am largely satisfied with the current state of the manuscript, I have a few remaining minor comments:

We appreciate this positive endorsement of our work.

The new sections of text contain several grammatical errors and inconsistent verb tenses that need to be corrected. Furthermore, the structure and formality of several sections could be improved. We have now carefully proofread the revision with the help of an external reviewer/proof-reader, who has helped edit for clarity and style. We would note that much of the text has been revised for grammar and style, and such corrections are not highlighted in yellow – we only highlight revisions that are in response to technical/scientific comments.

Figure 1: The amplitude modulated waveform in Figure 1A is not drawn to scale with regards to the actual carrier and beat frequencies. This is acceptable because this figure is illustrative, but the fact that the waveforms are not drawn to scale should at least be noted in the figure caption. Agreed – we were confused how to handle this, since if one draws it to scale in time the whole envelope would be filled. We take this useful advice to simply clarify in the caption.

At the top of page 19, the authors should consider removing the following statement: “Overall, the “U-curve” can be considered borderline significant.” This statement is subjective and is not explicitly addressed by the statistical comparisons mentioned in the previous sentences, especially when their results do show several statistically significant differences between the different beat frequencies.

Agreed, this is subjective as-written. We remove this and let the pairwise significance assays speak for themselves.

Article reference: NCOMMS-24-55626

R3: Itemized responses to the referee comment are found below, with original comments in red, and our response in blue. All text changes are marked in the manuscript with yellow highlights.

Reviewer #1 (Remarks to the Author):

Rev. 1

The authors have addressed my minor comments and I appreciate that the authors have tried to thoroughly answer my concern regarding the comparison between different waveforms and configurations and why comparing some of them directly is not straightforward. However, I am afraid there are several errors in their analytic approach that they are planning to include in their Appendix A in response to my question, which must be corrected before publication:

1-In order to compare the activation threshold, the authors are comparing activating functions. That is a valid approach because action potentials are most likely initiated where the activating function is maximum. However, the authors are comparing the activating function between two configurations at the same spatial location ($x=0$). That spatial location is where the maximum activating function occurs for a single source (the authors have assumed that implicitly). However, $x=0$ is not necessarily where the activating function is maximum for the paired sources. In order to compare the activation threshold between configurations based on activating function, the authors need to compare the maximum of activating function regardless of where they occur. Indeed, the amplitude of activating function for TIS configuration depends on the distance between the pairs of electrodes but so does the location of the maximum amplitude of activating function and that was not accounted for.

We appreciate the reviewer's points and attention to detail. We think that the disagreement here is a result of misunderstanding how we set up the coordinates of our model. This can be fully solved if we better explain the assumptions we are making for this analytical approach as well as better visualize and label what the geometry we are using represents and how the coordinate system is oriented. We realize that we should explain better the geometry we used in the model, as it corresponds to what is shown in Figure 6. In both the locust and the human experiments for TIS we are using a rectangular 4-electrode configuration where the same frequency stimulus is applied along the rectangle diagonals. The geometry for human stimulation is depicted from the top view in Figure 6 with the nerve outlined and repeated here with extra clarification of the lines of interest:

Activating function maximum across a horizontal cross section at the nerve depth V/m^2

In this geometry the maximum of the activating function can be either directly below the electrodes or in-between the electrodes on the shorter sides of the rectangle depending on the nerve depth (this is what the discussion was aimed at answering originally). But the main point is that it will always be on the lines labeled lines of interest which are going through the electrode centers along the shorter side of the rectangle. We can show that numerically with a FEM model, or analytically in the simplified analytical approach (derived below). Temporally, the activating function maximum will occur when the phases of the stimulation currents I_1 and I_2 align such that both electrodes on each line of interest have the same polarity. The opposite also holds and the TIS is at its minimum when the current polarities on the line of interest are opposite.

For our analytical approach we focused on a single pair of electrodes along one of the lines of interest and disregarded the other two electrodes since they are much farther away from the region where stimulation will take place so their effect is significantly weaker. This enabled us to reduce the number of variables to only the most important ones to hopefully drive the main points easier, and we think it is a reasonable approach for the simplified model that we show for the purpose of better developing intuition on the TIS response.

Mainly, we think the main source of disagreement is the way we oriented the coordinate system without explicitly marking it on the supplementary figure. We wanted to keep the axes oriented the same way as in figure 6 of the main text, more specifically the same as in Figure 6D. Now we have revised the appendix figure with coordinate axes labeled showing the vertical cross section along one of the lines of interest depicted in the former figure:

The x direction is the direction of the nerve propagation, and it is going out of the depicted plane towards the reader. The x coordinate is the same for the entire plane and for simplicity, we can choose the x coordinate to be $x=0$ as the result will not depend on the exact value of the chosen x coordinate.

We know from FEM simulations and from the analytical approach that follows that the activating function maximum is on the plane of interest (with x coordinate 0). For that reason, we did the derivation for $x=0$:

$$A_x = \left. \frac{d^2 V}{dx^2} \right|_{x=0} = -\frac{I}{2\pi\sigma r^3}$$

And then compared the activating functions for the two configurations used in the experimental configurations. However, we fully agree with the reviewer that the exact location of the nerve activation along the nerve is not a trivial issue and we shall therefore derive it here explicitly and prove that the global maximum truly occurs in-between electrodes at $x=0$.

The full expression for A_x (without assuming any x coordinate) for a single point source is:

$$A_x = \frac{d^2 V}{dx^2} = \frac{d^2}{dx^2} \left(\frac{I}{2\pi\sigma \sqrt{x^2 + y^2 + z^2}} \right) = \frac{I}{2\pi\sigma} \frac{2x^2 - y^2 - z^2}{(x^2 + y^2 + z^2)^{5/2}}$$

Now to find the maximum along the nerve we should consider the contributions of both electrodes and plug in the nerve trajectory as seen from the electrodes which gives $z = -d$ and $y = l$ for the left electrode and $y = -l$ for the right electrode. The total activating function along the nerve is then:

$$A_{x,tot} = A_x(x, l, -d) + A_x(x, -l, -d) = \frac{I}{\pi\sigma} \frac{2x^2 - l^2 - d^2}{(x^2 + l^2 + d^2)^{5/2}}$$

We can find the extremes of the activating function by finding where the x derivative is zero:

$$\frac{dA_{x,tot}}{dx} = \frac{I}{\pi\sigma} \frac{3x(3d^2 + 3l^2 - 2x^2)}{(x^2 + l^2 + d^2)^{7/2}} = 0$$

Which gives the solutions:

$$x_1 = 0$$

$$x_{2,3} = \pm \sqrt{\frac{3}{2}(d^2 + l^2)}$$

By inserting these coordinates into $A_{x,tot}$ we obtain:

$$A_{x,max,1} = -\frac{I}{\pi\sigma(d^2 + l^2)^{\frac{3}{2}}} \approx -0.32 \frac{I}{\sigma(d^2 + l^2)^{\frac{3}{2}}}$$

$$A_{x,max,2,3} = \sqrt{\frac{2}{5}} \cdot \frac{8I}{25\pi\sigma(d^2 + l^2)^{\frac{3}{2}}} \approx 0.06 \frac{I}{\sigma(d^2 + l^2)^{\frac{3}{2}}}$$

By comparing the values, we find that **the global extreme of the activation function for the experimental configuration along the nerve does indeed occur at $x=0$.**

2- Regardless of the first issue, the equation the authors are using for the activating function of TIS in the point they are making their comparison is incorrect. The equation for the single source is:

$$V_{xx} = I/2\pi\sigma (2X^2 - Y^2)/(X^2 + Y^2)^{(5/2)}$$

,which simplifies to $I/2\pi\sigma 1/d^3$ for $x=0$ and that is correct. However, X is no longer 0 for the paired sources where they are doing their comparison (note that the equation above assumes a coordinate system with its origin at the source). Therefore, the equations they are setting equal to each other to find a relationship between l and d is incorrect. In fact, the equation for sources that are located at $+l$ and $-l$ would be completely different if the authors want to use a global coordinate system with the origin where the single source is located at.

We think that the misunderstanding here is again caused by our choice of coordinate axes labels. Y and Z are not 0 for the paired sources, but the X is still zero at the maximum location.

For $x=0$ plane we have:

$$A_x = \frac{d^2V}{dx^2} \Big|_{x=0} = \frac{d^2}{dx^2} \left(\frac{I}{2\pi\sigma\sqrt{x^2 + y^2 + z^2}} \right) \Big|_{x=0} = \frac{I}{2\pi\sigma} \frac{2x^2 - y^2 - z^2}{(x^2 + y^2 + z^2)^{5/2}} \Big|_{x=0} = -\frac{I}{2\pi\sigma(y^2 + z^2)^{3/2}}$$

And by inserting $r = \sqrt{y^2 + z^2}$ for $x=0$ plane we get our expression

$$A_x = -\frac{I}{2\pi\sigma r^3}$$

To calculate the total activating function of TIS setup, we need to sum the contributions from both electrodes at the nerve location. Since the nerve is positioned symmetrically between the electrodes, for both electrodes the distance to the nerve r is $r = \sqrt{l^2 + d^2}$ so the total activating function in the TIS configuration is:

$$A_{x,total} = A_{x1} + A_{x2} = \frac{I}{2\pi\sigma(l^2 + d^2)^{3/2}} + \frac{I}{2\pi\sigma(l^2 + d^2)^{3/2}} = \frac{I}{\pi\sigma(l^2 + d^2)^{3/2}}$$

Which is the same expression that we have in the appendix and the same as $A_{x,max,1}$

Rev.4

The authors have satisfactorily addressed my comments.

However, unless I am mistaken, I believe there are mathematical errors in the authors' response to the first comment from Reviewer #1. Specifically, with regards to both their equation for current density (J) and the corresponding expression for potential (V).

For a point-source electrode, the potential field in an infinite homogeneous medium should be expressed as: $V=I/(4\pi\sigma r)$

Our model assumed a point-source electrode on top of a hemispheric medium, not an infinite (and thus spherical) one. This is to reflect the situation that the electrode is on top of the tissue and on the upper side there is air, where we assume no current is going – all current enters into the hemispheric medium only. This makes the current density twice as large as compared to the spherical case and therefore, the correct equation is with a factor of 2 and not 4, i.e. $V=I/(2\pi\sigma r)$. We have explained this directly now in the revised text of the appendix.

Additionally, I believe there is an issue with their expression for the activating function, A_x . Since the potential V for a point-source electrode is derived in terms of r, and the calculation of A_x requires taking the second-order partial derivative of V with respect to x (along the axon), the authors must specifically account for how r varies as a function of x. Therefore, their expressions for A_x require correction, particularly given that this analysis is now included as part of the Appendix in the manuscript.

The derivation was done correctly, but some steps have been omitted for brevity. We will demonstrate here that the result is correct.

Starting from the expression for electric potential:

$$V = \frac{I}{2\pi\sigma r} = \frac{I}{2\pi\sigma\sqrt{x^2 + y^2 + z^2}}$$

2-electrode configuration
Sine, AM Sine, Burst

4-electrode configuration
TIS

We have for $x=0$ plane depicted above:

$$A_x = \left. \frac{d^2 V}{dx^2} \right|_{x=0} = \left. \frac{d^2}{dx^2} \left(\frac{I}{2\pi\sigma\sqrt{x^2 + y^2 + z^2}} \right) \right|_{x=0} = \left. \frac{I}{2\pi\sigma} \frac{2x^2 - y^2 - z^2}{(x^2 + y^2 + z^2)^{5/2}} \right|_{x=0} = -\frac{I}{2\pi\sigma(y^2 + z^2)^{3/2}}$$

And by inserting $r = \sqrt{y^2 + z^2}$ for $x=0$ plane we get our expression

$$A_x = -\frac{I}{2\pi\sigma r^3}$$

To calculate the total activating function of TIS setup, we need to sum the contributions from both electrodes at the nerve location. Since the nerve positioned symmetrically between the electrodes, for both electrodes the distance to the nerve r is $r = \sqrt{l^2 + d^2}$ so the total activating function in the TIS configuration is:

$$A_{x,total} = A_{x1} + A_{x2} = \frac{I}{2\pi\sigma(l^2 + d^2)^{3/2}} + \frac{I}{2\pi\sigma(l^2 + d^2)^{3/2}} = \frac{I}{\pi\sigma(l^2 + d^2)^{3/2}}$$